# Implementation and assessment of a carbonate system model (Eco3M-CarbOx v1.1) in a highly-dynamic Mediterranean coastal site (Bay of Marseille, France).

Katixa Lajaunie-Salla[1], Frédéric Diaz[1], Cathy Wimart-Rousseau[1], Thibaut Wagener[1], Dominique Lefèvre[1], Christophe Yohia[2], Irène Xueref-Remy[3], Brian Nathan[3], Alexandre Armengaud[4], Christel Pinazo[1]

[1]Aix Marseille Univ., Université de Toulon, CNRS, IRD, MIO, UM 110, 13288, Marseille, France
[2]Aix Marseille Univ., CNRS, IRD, OSU Institut Pythéas, 13288, Marseille, France

[3]Aix Marseille Univ., Université d'Avignon, CNRS, IRD, IMBE, Marseille, France
[4]AtmoSud : Observatoire de la qualité de l'air en région Sud Provence Alpes Côte d'Azur, le Noilly Paradis, 146 rue Paradis, 13294 Marseille, Cedex 06, France

*Correspondance to* : Katixa Lajaunie-Salla (katixa.lajaunie@gmail.com), Frédéric Diaz (frederic.diaz@univ-amu.fr)

**Abstract.** A carbonate chemistry balance module was implemented into a biogeochemical model of the planktonic food web. The model, named Eco3M-CarbOx includes 22 state variables that are dispatched into 5 compartments: phytoplankton, heterotrophic bacteria, detrital particulate organic matter, labile dissolved organic and inorganic matter. This model is applied to and evaluated in the Bay of Marseille (BoM, France) that is a coastal zone impacted by the urbanized and industrialized Aix-Marseille Metropolis, and subject to significant increases in anthropogenic emissions of $CO_2$.

The model was evaluated over the year 2017 for which *in situ* data of carbonate system are available in the study site. The biogeochemical state variables of the model only change with time, to represent the time evolution of a sea surface water cell in response to the implemented realistic forcing conditions. The model correctly simulates the values ranges and seasonal dynamics of most of variables of carbonate system except for the total alkalinity. Several numerical experiments were conducted to test the response of carbonate system to (i) a seawater temperature increase, (ii) wind events, (iii) Rhône River plume intrusions and (iv) levels of atmospheric $CO_2$ contents. This set of numerical experiments shows that the Eco3M-CarbOx model provides expected responses in the alteration of the marine carbonate balance regarding each of the considered perturbation. When the seawater temperature changes quickly, the behaviour of the BoM waters alters within a few days from a source of $CO_2$ to the atmosphere to a sink into the ocean. Moreover, the higher the wind speed is, the higher the air-sea $CO_2$ gas exchange fluxes are. The river intrusions with nitrate supplies lead to a decrease in the $p$CO_2$ value, favouring the conditions of a sink for atmospheric $CO_2$ into the BoM. A scenario of high atmospheric concentrations of $CO_2$ also favours the conditions of a sink for atmospheric $CO_2$ into the waters of the BoM. Thus the model results suggest that external forcings have an important impact on the carbonate equilibrium in this coastal area.

## 1. Introduction

Current climate change mostly originates from the carbon dioxide ($CO_2$) increase in the atmosphere at a high annual rate (+2.63 ppm from May 2018 to May 2019, https://www.esrl.noaa.gov/gmd/ccgg/trends/global.html). This atmospheric $CO_2$ increase impacts the carbonate chemistry equilibrium of the oceanic water column (Allen et al., 2009; Matthews et al., 2009). Oceans are known to act as a sink for anthropogenic $CO_2$, *i.e.* 30% of emissions, which leads to a marine acidification (Gruber et al., 2019; Orr et al., 2005; Le Quéré et al., 2018).

$CO_2$ is a key molecule in the biogeochemical functioning of the marine ecosystem. Photo-autotrophic organisms, mainly phytoplankton and macro-algae, fix this gas through photosynthesis in the euphotic zone and, in turn, produce organic matter and dissolved oxygen. Heterotrophic organisms, mainly heterotrophic protists and metazoans consume organic matter and dissolved oxygen by aerobic respiration and, in turn, produce $CO_2$. In the Ocean, the main processes regulating $CO_2$ exchanges between the atmosphere and sea are the solubility pump and the biological pump at different time-scales. Overall, the thermohaline gradients drive the solubility pump, while the metabolic processes of gross primary production and respiration set the intensity of the biological pump (Raven and Falkowski, 1999).

The coastal zones, despite their small surface area and volume compared to those of the open ocean, have a large influence upon carbon dynamics and represent 14 to 30% of the oceanic primary production (Gattuso et al., 1998). At the interface between open-ocean and continents, these zones receive large inputs of nutrients and organic matter from rivers, groundwater discharge, and from atmospheric depositions (Cloern et al., 2014; Gattuso et al., 1998). On coasts, shorelines are subject to an increasing density of population and associated urbanization (Small and Nicholls, 2003). This rapid alteration of shorelines all over the world accelerates the emissions of greenhouses gases near the coastal ocean, and it also involves large discharges of material into the seawater by wastewater runoff and/or rivers (Cloern, 2001). These anthropogenic forcing alter the biogeochemical functioning of these zones and could lead to a growing eutrophication (Cloern, 2001). Moreover, these forcing could affect the carbonate chemistry dynamics and amplify or attenuate the acidification in coastal zones. This alteration of the marine environment may provoke further changes in the structure of the plankton community, including *in fine* consequences on the populations with high trophic levels, such as teleosts (Esbaugh et al., 2012). At the global scale, coastal zones are considered to be a significant sink for atmospheric $CO_2$, with an estimated flux converging to 0.2 PgC $y^{-1}$ (Roobaert et al., 2019). However, some studies highlight that the status of these areas as a net sink or source still remains uncertain due to the complexity of the interactions between biological and physical processes, and also due to the lack of *in situ* measurements (Borges and Abril, 2011; Chen et al., 2013; Chen and Borges, 2009). Moreover, the capacity for coastal zones to absorb atmospheric $CO_2$ resulting from the increasing human pressure also remains poorly known. There are few works which highlight, under future atmospheric $CO_2$ levels, whether coastal zones will become a net sink or a reduced source of $CO_2$ (Andersson and Mackenzie, 2012; Cai, 2011).

The current increase in the atmospheric $CO_2$ partial pressure ($pCO_2$) is slowly shifting the marine carbonate chemistry equilibrium towards increases in the seawater $pCO_2$ and bicarbonate ions ($HCO_3^-$) and decreases in $pH$ and carbonate ions ($CO_3^{2-}$) (Hoegh-Guldberg et al., 2018). These trends were already described in several coastal and open-ocean locations worldwide (Cai et al., 2011). In a coastal Northwestern Mediterranean site, a 10-year time-series of *in situ* measurements highlights a trend of $pH$ decrease and $pCO_2$ increase (Kapsenberg et al., 2017). Low $pH$ values can inhibit the ability of many marine organisms to form the calcium carbonate ($CaCO_3$) used in the making of skeletons and shells (Gattuso et al., 2015). In an extreme case, this shift may promote dissolution of $CaCO_3$ because the water will become under-saturated with respect to $CaCO_3$ minerals (Doney et al., 2009).

The present study is dedicated to the implementation of a carbonate system module into a preexisting biogeochemical model of planktonic food web. This new model, named Eco3M-CarbOx (v1.1) is then evaluated in a highly-dynamic

coastal site, *i.e.* Bay of Marseille (BoM) in the Northwestern Mediterranean Sea. This evaluation is performed on the seasonal dynamics of biogeochemical and carbonate modeled variables against that of the corresponding *in situ* data available over the year 2017. This study is extended by a fine analysis of the variability of the marine carbonate system (stocks, fluxes) in relation to physical (*e.g.* wind events, river intrusions, temperature increases, changes in the atmospheric $pCO_2$ levels) and biogeochemical processes (gross primary production (GPP) and respiration (R)) in the study site. The BoM is suitable to this kind of study because this coastal area is subject to high emissions of atmospheric $CO_2$ from the nearby urban area, and it also receives effluents from the Aix-Marseille metropolis. In addition, strong wind events (Mistral) regularly occur, which could lead to (i) strong latent heat losses at the surface (Herrmann et al., 2011) and upwelling along the coast with a common consequence of cooling effect and (ii) Rhône River plume intrusion under specific wind conditions (Fraysse et al., 2013, 2014). In this regional context, many anthropogenic forcing can interact with the dynamics of the carbonate systems. Natural determinants of the composition of the marine planktonic community can also play a crucial role in these dynamics.

## 2. Materials & Methods

### 2.1 Numerical model description

The Eco3M-CarbOx biogeochemical model was developed to represent the dynamics of the seawater carbonate system and plankton food web in the BoM. The model was implemented using the Eco3M (Ecological Mechanistic and Modular Modelling) platform (Baklouti et al., 2006). The model structure used is based on an existing model of the plankton ecosystem (Fraysse et al., 2013), including a description of Carbon (C), Nitrogen (N) and Phosphorus (P) marine biogeochemical cycles. The Eco3M-CarbOx model includes 22 prognostic state variables that are split into 5 compartments: phytoplankton, heterotrophic bacteria, detrital particulate organic matter, labile and semi-labile dissolved organic matter, nutrients (ammonia, nitrate and phosphate), dissolved oxygen, and carbonate system variables (Fig. 1). In this study, the state variables of the Eco3M-CarbOx model only change along time (*i.e.* usually termed "model 0D"), they are representative of the time evolution of a sea surface water cell but this biogeochemical model is not coupled with a hydrodynamic model.

The model presented in this study includes a set of new developments and improvements in the realism of the plankton web structure and process formulations compared to the model of Fraysse et al. (2013). In order to improve the representation of chlorophyll concentration in the Bay of Marseille the phytoplankton is divided in two groups: one with some ecological and physiological traits of the *Synechococcus* cyanobacteria, which is one of the major constitutive members of pico-autotrophs in Mediterranean Sea (Mella-Flores et al., 2011), and another with traits of large diatoms, which are generally observed during spring blooms at mid-latitudes (Margalef, 1978). For both of the phytoplankton, there is a diagnostic chlorophyll-a variable related to the phytoplankton C-biomass, the phytoplankton N-to-C ratio, and the limiting internal ratio $f_Q^N$ (Faure et al., 2010; Smith and Tett, 2000; Tab. B2, Appendix B). The functional response of primary production was modified using another formulation of temperature limitation function which takes into account the optimal temperature of growth for each phytoplankton group. The exudation of phytoplankton was modified taking into account the intracellular phytoplankton ratio. For the uptake of matter by bacteria and the remineralization processes the dependence on intracellular bacteria ratio was modified. A temperature dependence of all biogeochemical processes was added to take into account the effects of rapid and strong variations of seawater temperature on plankton during episodes of upwelling for instance that are usually observed in the BoM. Also certain parameters in some formulations were modified owing to the alterations of some formulations (Tabs. B4 & B5, Appendix B).

Additionally, a carbonate system module was developed and three state variables were added: dissolved inorganic
carbon (DIC), total alkalinity (TA) and the calcium carbonate ($CaCO_3$) implicitly representing calcifying organisms.
The knowledge of DIC and TA allows the calculation of the $pCO_2$ and $pH$ (total $pH$ scale) diagnostic variables,
necessary for resolving all the equations of the carbonate system. These equations use apparent equilibrium constants,
which depend on temperature, pressure, and salinity (Dickson, 1990a, 1990b; Dickson and Riley, 1979; Lueker et al.,
2000; Millero, 1995; Morris and Riley, 1966; Mucci, 1983; Riley, 1965; Riley and Tongudai, 1967; Uppström, 1974;
Weiss, 1974). The details of the resolution of carbonate system module are given in the Appendix A. For this module
three processes were also added: the precipitation and dissolution of calcium carbonate and the gas exchange of $pCO_2$
with the atmosphere. Based on the review of Middelburg (2019), it is considered that: (i) TA decreases by 2 moles for
each mole of $CaCO_3$ precipitated, by 1 mole for each mole of ammonium nitrified, by 1 mole for each mole of
ammonium assimilated by phytoplankton, and TA increases by 2 moles for each mole of $CaCO_3$ dissolved, and by 1
mole for each mole of organic matter mineralized by bacteria in ammonium (Tab. B2, Appendix B). (ii) DIC is
consumed during the photosynthesis and calcification processes and is produced by respiration (of phytoplankton,
zooplankton, and bacteria) and the $CaCO_3$ dissolution processes. Moreover, the dynamics of DIC are altered by the
$CO_2$ exchanges with the atmosphere (Tab. B2, Appendix B). The air-sea $CO_2$ fluxes are calculated from the $pCO_2$
gradient across the air-sea interface and the gas transfer velocity (Tab. B3, Appendix B) estimated from the wind speed
and using the parametrization of Wanninkhof (1992).
In the Eco3M-CarbOx model, zooplankton is considered as an implicit variable. However, a closure term based on the
assumption that all of the matter grazed by the zooplankton and higher trophic levels returns as either organic or
inorganic matter by excretion, egestion and mortality processes is taken into account (Fraysse et al., 2013). The model
considers a "non-redfieldian" stoichiometry for phytoplankton and bacteria. All the biogeochemical model
formulations, equations and associated parameters values are detailed in the Appendix B.
**2.2    Study area**
The BoM is located in the eastern part of the Gulf of Lions, in the Northwestern Mediterranean Sea (Fig. 2). The city
of Marseille, located on the coast of the BoM, is the second largest city of France, with a population of *ca.* 1 million.
The Rhône River, which flows into the Gulf of Lions, is the greatest source of freshwater and nutrients for the
Mediterranean Sea, with a river mean flow of 1800 $m^3$ $s^{-1}$ (Pont et al., 2002). Several studies highlight the eastward
intrusion events from the Rhône River plume in the BoM under East and South-easterly wind conditions, which favor
biological productivity (Fraysse et al., 2014; Gatti et al., 2006; Para et al., 2010). The biogeochemistry of the BoM is
complex and highly driven by hydrodynamics. For example, North-Northwesterly winds induce upwelling events
which bring upward cold and nutrient-rich waters (Fraysse et al., 2013). Moreover, the oligotrophic Northern Current
occasionally intrudes into the BoM (Petrenko, 2003; Ross et al., 2016).
Despite the presence of several marine protected areas around the BoM (the Regional Park of Camargue, the Marine
protected area Côte Bleue and the National Park of Calanques), it is strongly impacted by diverse anthropogenic
forcing, because industrialized and urbanized areas are located all along the coast. From the land, the BoM receives
nutrients and organic matter from the urban area of the Aix-Marseille metropolis (Millet et al., 2018), the industrialized
area of Fos-sur-Mer (one of the biggest oil-based industry areas in Europe), and the Berre Lagoon, which is eutrophized
(Gouze et al., 2008; Fig. 2C). From the atmosphere, the BoM is subject to fine particles deposition and greenhouse gas
emissions (including $CO_2$) from the nearby urban area, and it also receives effluents from the Aix-Marseille metropolis.
**2.3    Dataset**

The modelled variables of the carbonate system (DIC, TA, $p$H and $p$CO$_2$) and chlorophyll-a are hereafter compared to observations collected at the SOLEMIO station (Figs. 2C & 3), which is a component of the French national monitoring network (Service d'Observation en Milieu Littoral - SOMLIT, http://somlit.epoc.u-bordeaux1.fr/fr/). Major biogeochemical parameters have been recorded since 1994. Carbonate chemistry variables ($p$H, $p$CO$_2$, DIC and TA) have been available since 2016, every two weeks.

## 2.4    Design of numerical experiments

In the present work, the Eco3M-CarbOx model was run for the whole year of 2017. This year was chosen because *in situ* data of carbonate systems (DIC, TA, $p$H and $p$CO$_2$) are available for the whole year at the SOLEMIO station (Fig. 2C). The biogeochemical variables were initialized using *in situ* data from winter conditions (Tab. B1, Appendix B). The model was forced by time-series of sea surface temperature and salinity, wind (at 10 m), light, and atmospheric CO$_2$ concentrations. The sea temperature time-series is from *in situ* hourly data recorded at the Planier station (Fig. 2C). For salinity, hourly *in situ* data from the SOLEMIO station and from the CARRY buoy were used (Fig. 2C). Wind and light hourly time-series were extracted from the WRF meteorological model at the SOLEMIO station (Yohia, 2017). Finally, we used hourly atmospheric CO$_2$ values from *in situ* measurements recorded at the *Cinq Avenues* station (CAV station, Fig. 2B) by the AtmoSud Regional Atmospheric Survey Network, France (https://www.atmosud.org). This simulation is the reference simulation (noted S0). As highlighted previously, Rhône River plume intrusions (due to wind-specific conditions) have an impact on the dynamics of primary production (Fraysse et al., 2014; Ross et al., 2016) and then on the seawater carbonate system. Moreover, the seawater temperature and atmospheric CO$_2$ variations control the seawater CO$_2$ dynamics *via* the solubility equilibrium and gas exchange with the atmosphere (Middelburg, 2019). In order to quantify the impact of different forcing, several simulations (hereafter noted S), which are summarized in Table 1, were conducted:

● Impact of seawater temperature increase, S1: the forcing time-series of *in situ* temperatures was shifted by +1.5°C (Cocco et al., 2013).

● Impact of wind events: a first simulation S2 was run with a constant wind intensity of 7 m s$^{-1}$ (2017 annual average wind speed) throughout the year and a second one (S3) with two three-day periods of strong wind speed (20 m s$^{-1}$) representative of short bursts of Mistral (data not shown) starting on May 15$^{th}$ and August 15$^{th}$, and a constant value of 7 m s$^{-1}$ the rest of the year.

● Impact of nutrient supply (nitrate) during a Rhône River plume intrusion (S4). A threshold of 37 has been chosen to identify the presence of low-salinity waters from the Rhône River plume in the forcing file of salinity. Here, the contents of nitrate supplied by the river depends on the salinity. A relationship between these two variables was then established for the SOLEMIO point from the MARS3D-RHOMA coupled physical and biogeochemical model (Fraysse et al., 2013; Pairaud et al., 2011). This relationship has already been used successfully to reproduce realistic observed conditions in the studies of Fraysse et al. (2014) and Ross et al. (2016): NO$_{3intrusion}$ (mmol m$^{-3}$) = -1.70×S + 65.

● Non-urban atmospheric CO$_2$ concentrations (S5): this simulation takes into account the forcing of atmospheric CO$_2$ values measured at the *Observatoire de Haute Provence* station (OHP, Fig. 2B) located outside of the Aix-Marseille metropolis from the ICOS National Network, France (http://www.obs-hp.fr/ICOS/Plaquette-ICOS-201407_lite.pdf).

In this work, we calculated the daily mean values of state variables, statistical parameters and mean fluxes of modeled processes throughout the year and over two main hydrological periods: the stratified and mixed water column periods. The stratified water column (SWC) is defined with a temperature difference between the surface and bottom of more than 0.5°C (Monterey and Levitus, 1997). For the simulated year (2017), the SWC period lasts from May 10th to October 20th. The mixed water column (MWC) period corresponds to the rest of the year.

## 3. Results

### 3.1 Model skills

Following the recommendations of Rykiel (1996), three criteria were considered to evaluate the performance of our model:

- Does the model reproduce the timing of the observed variations of carbonate system at the seasonal time scale?
- Does the model reproduce the observed $pCO_2$ and $pH$ ranges at the seasonal time scale?
- Analysis of the Willmott Skill Score (WSS): this index is an objective measurement of the degree of agreement between the modeled results and the observed data. A correct representation of observations by the model is achieved when this index is higher than 0.70 (Willmott, 1982).

Over most of the studied period, the model simulates lower chlorophyll-a concentrations than the *in situ* observations, especially during the MWC period (Fig. 3A). Two maxima of chlorophyll-a concentrations are observed *in situ:* the first one at *ca.* 1.71 mg m$^{-3}$ in March and the second one at *ca.* 0.68 mg m$^{-3}$ in May. They are both linked to Rhône River plume intrusions. Several *in situ* maxima between 0.50 and 0.70 mg m$^{-3}$ are observed between March and April (at the end of the MWC period), and they signaled the spring bloom event (Tab. 2 & Fig. 3A). The biogeochemical model quantitatively reproduces the spring bloom observed at the end of the MWC period (Fig. 3A) with a maximum value of *ca.* 0.69 mg m$^{-3}$. The model does not catch the two aforementioned maxima of chlorophyll, and it contains a low WSS and a strong bias (0.37 and +0.22 mg m$^{-3}$, respectively - Tab. 2).

On the whole, the seasonal variations of the seawater $pCO_2$ are correctly simulated by the biogeochemical model (Fig. 3B), even if the values are rather overestimated during the MWC period. From January to February, the model reproduces the slight decrease in the observed $pCO_2$ and from February to March the increase in $pCO_2$ even if the latter modelled remains smaller. In mid-April, during the simulated spring bloom period, the observed drop in $pCO_2$ and increase in $pH$ are also spotted in the model (Fig. 3B & 3C). The model especially succeeds in reproducing the observed increase in relation to high temperatures during the SWC period. The reduction of the $CO_2$ solubility due to thermal effects mostly explains the increase in $pCO_2$ during the SWC period. The strong standard deviation of modeled values during the SWC period can be explained by the rapid changes in temperature probably due to upwelling usually occurring at this time of the year (Millot, 1990). The range of modeled $pCO_2$ values (345 - 503 µatm) encompasses the range of observed values (358 - 471 µatm; Tab. 2). The statistical analysis provides a mean bias of +23 µatm, and a WSS of 0.69 (Tab. 2).

The seasonal dynamic of $pH$ is mostly reproduced by the model, and in particular, the decrease during the SWC period (Fig. 3C). However, the modelled $pH$ is generally underestimated throughout the year, except during the SWC period, with a mean bias of -0.015 (Tab. 2). The seasonal range is captured by the model with a minimum value during the SWC period (7.994 *vs.* 8.014 for observations; Tab.2) and a maximum one during the MWC period (8.137 *vs.* 8.114 for observations; Tab.2). The statistical analysis highlights an index of agreement between the *in situ* data and the model outputs higher than 0.70 (Tab. 2).

The seasonal variations of DIC show the highest values during the MWC period and a decrease (resp. increase) during the beginning (resp. the end) of the SWC period (Fig. 3D). The lowest values are observed during September. The Eco3M-CarbOx model closely matches the seasonal dynamic by reproducing the range of extreme observed values (Tab. 2). The mean bias is also small (-8.48 µmol kg$^{-1}$, Tab. 2). More than 70% (0.73, Tab. 2) of modeled DIC concentrations are in statistical agreement with the corresponding observations.

The seasonal cycle of measured TA does not show a clear pattern (Fig. 3E). Large variations of values ranging between 2561 and 2624 µmol kg$^{-1}$ (Tab. 2) are observed, whatever the hydrological season is that is considered. The biogeochemical model provides almost constant values around 2570 µmol kg$^{-1}$ all along the year, which is lower than *in situ* data. With a low WSS index of agreement and a large mean bias (Tab. 2), the model is not able to confidently reproduce the observed variations of TA (Fig. 3E & Tab. 2).

### 3.2 Carbon fluxes and budgets

For the year 2017, the values of temperature vary between 13.3°C and 25.9°C (Fig. 4A). The DIC variations closely match those of temperature (correlation coef. -0.75). For example, the spring increase in temperature leads to a decrease in DIC concentrations (Figs. 4A & 4C), and the minimum values are reached at the end of SWC period. Over the simulated period, the air-sea $CO_2$ fluxes ($F_{aera}$) vary between -14 and 17 mmol m$^{-3}$ d$^{-1}$, with a weakly positive annual budget of +6 mmol m$^{-3}$ y$^{-1}$ (or +0.017 mmol m$^{-3}$ d$^{-1}$, Tab. 3). Then, the BoM waters would act as a net source of $CO_2$ to the atmosphere on an annual basis. However, on a seasonal basis, the BoM waters would change from a net sink during the MWC period ($F_{aera}$ <0; Tab. 3) to a net source during the SWC one ($F_{aera}$ >0; Tab. 3).

On an annual basis, the gross primary production (GPP) and total respiration (R) are balanced, leading to a null average net ecosystem production (NEP, NEP=GPP-R) (Fig. 4F & Tab. 3). The intensity of autotroph respiration ($R_a$) is lower than that of primary production (annual mean of 0.065 *vs.* -0.413 mmol m$^{-3}$ d$^{-1}$, respectively - Tab. 3). While the zooplankton and bacterial respiration account for an average of 0.348 mmol m$^{-3}$ d$^{-1}$ (Tab. 3). On a seasonal basis, the model highlights an ecosystem dominated by autotrophy during the MWC period (NEP>0; Tab. 3) and heterotrophy during the SWC period with higher fluxes values (NEP<0; Tab. 3). The biogeochemical fluxes show the strongest variations along the SWC period, following those of temperature (Fig. 4F). The maximum GPP occurs in April and is correlated with the maximum chlorophyll concentration. At this time, the ecosystem is autotroph (NEP>0; Figs. 4B & 4F), and is a net sink for atmospheric $CO_2$, which explains the DIC and seawater $p$CO$_2$ decreases during the bloom period (Figs. 4C, 4D & 4E)

When looking in details at the temperature and salinity 2017 time-series (Fig. 4A), several crucial events can be seen occurring, including freshwater intrusions (*e.g.* March 15[th] and May 6[th]) into the BoM and large variations of temperature in relation with upwelling events or latent heat losses due to wind bursts. The largest freshwater intrusion from the Rhône River plume occurs in mid-March, with a minimum observed salinity of *ca.* 32.5 at the SOLEMIO station (Fig. 4A). During this event, the seawater $p$CO$_2$ decreases and $p$H increases concomitantly (Figs. 4C & 4D). Then, seawater appears to be temporarily under-saturated in $CO_2$ and the BoM waters thus acts as a sink for atmospheric $CO_2$ at the time of intrusion (Fig. 4E).

During the SWC period, upwelling events quickly cool the surface seawater. In two days, from July 25[th] to 27[th], the water temperature drops from 24.7°C to 16.9°C (Fig. 4G). The decrease in temperature corresponds to the increase in DIC concentrations (Fig. 4I). Concomitantly, the values of seawater $p$CO$_2$ decrease from 497 to 352 µatm and $p$H increase from 7.99 to 8.12 (Figs. 4I & 4J). This event quickly changes the BoM waters from a source to a sink for atmospheric $CO_2$ (from +17 to -14 mmol m$^{-3}$ d$^{-1}$, Fig. 4K), and also from a net heterotroph to a net autotroph ecosystem (Fig. 4L).

### 3.3 Impact of external forcing on the dynamics of carbonate system

#### 3.3.1 Temperature increase

Here we compare the reference simulation S0 with the S1 simulation (seawater temperature elevation of 1.5°C - Fig. 5). During the year, there are few changes on the carbonate system variables such as the $p$CO$_2$ and $p$H (data not shown). The main alterations occur during the blooms of phytoplankton. The simulated bloom of phytoplankton occurs later, at beginning of May, for both diatoms and picophytoplankton, with a maximum value of chlorophyll at 1.4 and 0.4 mg m$^{-3}$, respectively (Figs. 5A & 5F).

As both the limitations due to light and nutrients remain about the same during the simulations S0 and S1, this counterintuitive occurrence of bloom relative to changes in temperature is mainly explained by the temperature limiting function, which depends on the optimal temperature of growth ($T_{opt}$). For the picophytoplankton, from January to April, the increase of 1.5°C drastically reduces the limitation by temperature (Fig. 5C), because the temperature is closer to the optimal temperature ($T_{opt}$=16°C, Tab. A4) during S1 than S0. In the S0 simulation, the temperature reaches $T_{opt}$ *ca.* April 15$^{th}$ and it induces the bloom, while at the same time in S1 the temperature moves slightly away from $T_{opt}$ and it does not enable the triggering of a bloom. At the time of the bloom in S1, the opposite configuration occurs. In S0, the ambient temperature is again far from $T_{opt}$, explaining the absence of bloom, while in the S1 the ambient temperature is closer to $T_{opt}$ enabling the occurrence of bloom. The picophytoplankton bloom then occurs later in the warm simulation S1 than in the reference simulation S0 (Fig. 5A). The duration and termination of bloom is controlled both by the nutrients availability and the temperature (Figs. 5C & 5D). Inversely, from January to April, the diatoms' growth limitation by temperature is strengthened in the warm simulation S1 (Fig. 5H), because the resulting ambient temperature is further from the optimum temperature ($T_{opt}$=13°C, Tab. A4) than that in the reference simulation S0. This induces a slower growth of diatoms and a delay of the maximum concentration (Fig. 5F). Afterwards the photosynthesis is mainly limited by temperature (Fig. 5H).

The ecosystem is net autotroph at the time of blooms whatever the simulation considered (NEP>0; Fig. 5E) and the quantity of DIC (not shown) fixed through autotroph processes is larger than that released by heterotroph processes. During the short period of bloom, the seawater $p$CO$_2$ decreases, leading to some negative air-sea fluxes (*i.e.* an oceanic sink for atmospheric CO$_2$). In the warm simulation, the later occurrence of bloom enables the period of the spring sink to extend by *ca.* three weeks over May relative to the reference simulation (Fig. 5J).

#### 3.3.2 Wind speed

The Bay of Marseille is periodically under the influence of strong wind events (Millot, 1990). Here we compare two simulations: one with a constant wind value (S2) and the other one with two wind events that occur in May and August (S3) (Figs. 6A & D). The result of this numerical experiment shows that the stronger the wind speed is, the higher the air-sea fluxes are, mainly owing to the increase in gas transfer velocity. Depending on the gradient of CO$_2$ between seawater and the atmosphere, strong wind speeds will favor either the emission or uptake of CO$_2$ (Figs. 6B & E). In May, with the air-sea CO$_2$ flux being positive, the outgassing of CO$_2$ to the atmosphere is enhanced leading to a decrease in seawater $p$CO$_2$ (Fig. 6C). On the contrary, in August the oceanic sink of atmospheric CO$_2$ is amplified which leads to an increase in the seawater $p$CO$_2$ value (Fig. 6F).

#### 3.3.3 Supply in nitrate by river inputs

According to the model results (Fig. 7), the occasional inputs of nitrate (S4) that are linked to Rhône River plume intrusions favor primary production and they led to increased chlorophyll concentrations (Figs. 7B & 7C) five times

during the SWC period. These blooms, as seen previously, lead to a decrease (resp. increase) in the seawater $pCO_2$ (resp. $p$H) (Figs. 7E & 7F). It can be noted that with the strongest river supply at mid-March (Figs. 7A & 7B) the occurrence of the spring bloom is earlier (Fig. 7C) than that occurring in the reference simulation (S0). The time lag between river nutrient supply and bloom is due to the temperature limitation (Fig. 4C). During blooms occurring within the SWC period following intrusions, the DIC concentrations are generally lower than those of the reference simulation, as in the case of the bloom of mid-May (decrease by $ca.$ 15 µmol kg$^{-1}$, Fig. 7J), due to the autotroph processes dominating the heterotroph ones. In turn, the seawater $pCO_2$ drops by $ca.$ 30 µatm (Fig. 7K) and $p$H increases by $ca.$ 0.030 (Fig. 7L). Nitrate inputs, favoring primary production, reduce the source of $CO_2$ to the atmosphere and intensify the sink of atmospheric $CO_2$ into the waters of BoM (Figs. 7E & 7K).

### 3.3.4    Urban air $CO_2$ concentrations

The Aix-Marseille metropolis is strongly subject to urban emissions to the atmosphere (Xueref-Remy et al., 2018a). The seasonal variability of atmospheric $CO_2$ concentrations at the urban site (CAV station, Fig. 2) is much higher than that observed in a non-urban area (OHP station, Fig. 2), especially during the MWC period (Fig. 8A): $CO_2$ concentrations vary between 379 and 547 µatm at the CAV station and between 381 and 429 µatm at the OHP station. Moreover, in winter the atmospheric $pCO_2$ is higher in the urban area than non-urban area, whereas in summer those of both areas are quite close. These differences in the seasonal pattern and between areas are usually explained by (i) the thinner atmospheric boundary layer, (ii) the decreased fixation of $CO_2$ by terrestrial vegetation, and (iii) the greater influence of anthropogenic activities by emissions from heating (Xueref-Remy et al., 2018b). Forcing the model by atmospheric pCO$_2$ values from urban or non-urban site can lead to significant differences in the values of the seawater $pCO_2$ during the MWC period especially. The air-sea gradient of $pCO_2$ is higher when using a forcing derived from the $CO_2$ concentrations originating from an urban area than from non-urban area, which strengthens the sink of atmospheric $CO_2$ into the waters of BoM. The seawater $pCO_2$ is then lower with non-urban area pressure (S5) than with urban area pressure (S0), because of lower $CO_2$ solubility in the BoM (Fig. 8B).

### 4.    Discussion

### 4.1    Model performance

The evaluation of model skill $vs.$ $in$ $situ$ data highlights that the modeled $p$H, $pCO_2$, DIC are in acceptable agreement with observations (Fig. 3). The seasonal variations observed for the different variables are captured by the model, including for example the seasonal decrease in DIC and $p$H during the SWC period, in relation to the increase in $pCO_2$, and the inverse scenario during the MWC period. The chlorophyll content variability is not well reproduced, especially during spring (Fig. 3A), even taking into account the nitrate supply from the Rhône River plume intrusion (Fig. 7C). This is due to the multiple origins of chlorophyll, organic matter, and nutrients in the BoM that are not accounted for in the Eco3M-CarbOx model: autochthonous marine production, and allochthonous origins from the Rhône and Huveaune River plumes (Fraysse et al., 2013). The observed variations and levels of TA are not correctly simulated by the model (Fig. 3F). The study of Soetaert et al. (2007) highlights that the main variations of TA in the marine coastal zones are linked to freshwater supplies and marine sediments. The present study does not take into account the inputs of TA from the Rhône River and the water-sediment interface, and it may explain why the TA variable is not correctly predicted by our model.

**4.2    Contribution of physical and biogeochemical processes to the variability of carbonate system**

The contribution of each biogeochemical process to the DIC variability can be assessed using the presented model: the aeration process contributes to 78% of the DIC variations and biogeochemical processes together to 22% (Tab. 3). As mentioned by Wimart-Rousseau et al. (2020), the model suggests that the seawater $pCO_2$ variations and associated fluxes would be mostly driven by the seawater temperature dynamics. Moreover, the seasonal variations of the air-sea $CO_2$ flux are in agreement with some previous field studies (De Carlo et al., 2013; Wimart-Rousseau et al., 2020), which measured a weak oceanic sink for atmospheric $CO_2$ during winter and a weak source to the atmosphere during summer.

The model results reveal that temperature would play a crucial role in controlling two counterbalanced processes: (1) the carbonate system equilibrium and (2) the phytoplankton growth. The increase in temperature during SWC leads to a higher $pCO_2$ in seawater due to the decrease in the $CO_2$ solubility (Middelburg, 2019) and, at the same time, the fixation of DIC by phytoplankton is favored, leading to a decrease in the $pCO_2$ level. The imbalance between the latter two processes leads to a change in the ecosystem status (autotrophic or heterotrophic) and the corresponding behavior as a sink or source to the atmosphere. In case of a 1.5°C rise over the whole year, the temperature variation has a very small impact on the carbonate system dynamics. However, it favors the autotrophic processes and strengthens the oceanic sink of atmospheric $CO_2$ during the bloom of phytoplankton (Figs. 5E & 5J).

**4.3    Contribution of the external forcing to the variability of carbonate system**

In line with several previous works on the Northwestern Mediterranean Sea (De Carlo et al., 2013; Copin-Montégut et al., 2004; Wimart-Rousseau et al., 2020), the model also suggests that the status of the Bay of Marseille regarding sink or source for $CO_2$ could change at high temporal frequency (*i.e.* hours to days). Bursts of North, Northwestern winds, lead to sudden and sharp decreases in seawater temperature (<2 days, Fig. 4G) either directly by latent heat loss through evaporation at the surface (Herrmann et al., 2011) or indirectly by creating upwelling (Millot, 1990), with the consequences of decrease in the seawater $pCO_2$ values (Fig. 4J) and *in fine*, an alteration of the $CO_2$ air-sea fluxes. Model results suggest that the fast variations of temperature could lead to rapid changes of the sink *vs.* source status in this coastal zone (Fig. 4K). Moreover, Fraysse et al. (2013) highlight that upwelling in the BoM favors ephemeral blooms of phytoplankton by nutrient supplies up to euphotic layer and would, in turn, contribute to the seawater $pCO_2$ decrease. North, and Northwestern winds through latent heat losses and/or upwelling events could then enhance the sink for atmospheric $CO_2$ due to the temperature drop and nutrients inputs. However, these results remain to be preliminary because in our experimental design only the cooling effect of upwelling on the carbonate balance is taken into account. But concomitantly, upwelling usually bring nutrients and DIC at the surface and these supplies could also perturb the balance of the carbonate system. A next coupling of the Eco3M-CarbOx model with a tridimensional hydrodynamic model would enable to certainly embrace the multiple effects of upwelling on the dynamics of the carbonate system in this area and refine the results presented in this study.

High wind speeds (>7 m s$^{-1}$) amplified considerably the gaseous exchange of $CO_2$ (De Carlo et al., 2013; Copin-Montégut et al., 2004; Wimart-Rousseau et al., 2020). The model highlights that a strong wind event of 3 days has a significant impact on the seawater $pCO_2$ values during a longer period of *ca.* 15 days (Fig. 6). A combination of high atmospheric $pCO_2$ value and high wind speed would then favor the sink for $CO_2$ into the waters of the BoM. The aeration process depends also on the choice of the formulation of the gas transfer velocity ($k_{600}$). In this study, the formulation of Wanninkhof (1992) is used and depends of the wind speed at 10 m above the water surface. However, the current velocity could favor the gas exchange and suspended matter concentration could limit the gas exchange (Abril et al., 2009; Upstill-Goddard, 2006; Zappa et al., 2003). Due to the important heterogeneity of physical and

biogeochemical forcings in coastal zones, other factors that control the air-sea gas exchange should certainly be taken
into account.
The simulation with intrusions of the Rhône River plume shows that inputs of nitrate cause a drop of seawater $p$CO$_2$
owing to nutrients supply favoring the phytoplankton development (Fig. 7). In this scenario, the oceanic sink of
atmospheric CO$_2$ is enhanced. But rivers also supply TA (*e.g.* Gemayel et al., 2015; Schneider et al., 2007) and DIC
(*e.g.* Sempéré et al., 2000) that shift the carbonate system equilibrium toward a $p$CO$_2$ decrease and a DIC increase
(Middelburg, 2019). Taking into account these further supplies may sensibly modify the modeled carbonate balance
in the BoM. A next step to the present work will be to design more realistic numerical experiments to refine the results
obtained in this preliminary study. The intrusions of Rhône River plume also induce a salinity decrease of the BoM
waters, which leads to drop the $p$CO$_2$ levels in the model. This drop of $p$CO$_2$ is due to the decrease in the CO$_2$ solubility
when salinity decreases (Middelburg, 2019).
In the scenario of forcing the model by using urban atmospheric $p$CO$_2$ time-series, the air-sea gradient increases and
then, it enhances the status of the BoM as a sink for atmospheric CO$_2$. As suggested by the *in situ* study of Wimart-
Rousseau et al. (2020), the Eco3M-Carbox model highlights the crucial role of the coastal ocean in urbanized area,
with an increase in atmospheric CO$_2$, the CO$_2$ uptake by the costal ocean may increase. This results is in line with
studies of Andersson and Mackenzie (2004) and Cai (2011) that predict an increase in the intensity of CO$_2$ sink and a
potential threat to coastal marine biodiversity in coastal areas owing to high atmospheric CO$_2$ levels.
**5. Conclusion**
A marine carbonate chemistry module was implemented in the Eco3M-CarbOx biogeochemical model and evaluated
against *in situ* data available in the Bay of Marseille (Northwestern Med. Sea) over the year 2017. The model correctly
simulates the values ranges and seasonal dynamics of most of variables of the carbonate system except for the total
alkalinity. Several numerical experiments were also conducted to test the sensitivity of carbon balance to physical
processes (temperature and salinity), biogeochemical processes (GPP and respiration processes) and external forcing
(wind, river intrusion and atmospheric CO$_2$). This set of numerical experiments shows that the Eco3M-CarbOx model
provides expected responses in the alteration of the marine carbonate balance regarding each of the considered
perturbation.
On the whole, the model results suggest that the carbonate system is mainly driven by the seawater temperature
dynamics. At a seasonal scale, the BoM marine waters appear to be a net sink of atmospheric CO$_2$ and a dominant
autotroph ecosystem during the MWC period, and a net source of CO$_2$ to the atmosphere during the SWC period,
which is mainly characterized by a dominance of heterotroph processes. However, the model results highlight that
sharp seawater cooling observed within the SWC period and probably owing to upwelling events, cause the CO$_2$ status
of the BoM marine waters to change from a source to the atmosphere to a sink into the ocean within a few days.
External forcing as the temperature increases leads to a delay in the bloom of phytoplankton. Strong wind events
enhance the gas exchange of CO$_2$ with the atmosphere. A Rhône River plume intrusion with input of nitrate favors
$p$CO$_2$ decreases, and the sink of atmospheric CO$_2$ into the BoM waters is enhanced. The higher atmospheric $p$CO$_2$
values from the urban area intensify the oceanic sink of atmospheric CO$_2$.
The BoM biogeochemical functioning is mainly forced by wind-driven hydrodynamics (upwelling, downwelling),
urban rivers, wastewater treatment plants, and atmospheric deposition (Fraysse et al., 2013). In addition, Northern
Current and Rhône River plume intrusions frequently occurred (Fraysse et al., 2014; Ross et al., 2016). Moreover, the
BoM harbors the second bigger metropolis of France (Marseille) that is impacted by many harbor activities. The next

step of this study will be to couple the Eco3M-CarbOx biogeochemical model to a 3D hydrodynamic model that will mirror the complexity of the BoM functioning. In this way, the contributions of hydrodynamic, atmospheric, anthropic, and biogeochemical processes to the DIC variability will be able to be determined with higher refinement and realism, and an overview of the air-sea $CO_2$ exchange could be made at the scale of the Bay of Marseille. The main results of our study could be transposed to other coastal sites that are also impacted by urban and anthropic pressures. Moreover, in this paper we highlighted that fast and strong variations of $pCO_2$ values occur, so thus it is essential to acquire more *in situ* values at high frequency (at least with an hourly resolution) to understand the rapid variations of the marine carbon system at these short spatial and temporal scales.

**Appendix A: Details of resolution of carbonate system module**

    **A.1. Calculation of carbonate systems constants:**

- Total Fluoride ($TF$) concentrations from Riley (1965) in mol kg$^{-1}$:

$$TF = \frac{0.000067}{18.998} \cdot \frac{S}{1.80655}$$

- Total Sulphate ($TS$) concentration from Morris and Riley (1966) in mol kg$^{-1}$:

$$TS = \frac{0.14}{96.062} \cdot \frac{S}{1.80655}$$

- Calcium ion concentration from Riley and Tongudai (1967) in mol kg$^{-1}$:

$$Ca^{2+} = \frac{0.02128}{40.087} \cdot \frac{S}{1.80655}$$

- Total Boron (TB) concentration from Uppström (1974) in mol kg$^{-1}$:

$$TB = \frac{0.000416 \cdot S}{35}$$

- Ionic Strength (IonS) from Millero (1982):

$$IonS = \frac{19.924 \cdot S}{1000 - 1.005 \cdot S}$$

The constants are calculated on the total $p$H scale except for $K_S$ on free $p$H scale

- If necessary, pH scale conversion factors are following:

From Seawater $p$H Scale (SWS) to total $p$H scale: $\text{SWStoTOT} = \dfrac{1+\frac{TS}{K_S}}{1+\frac{TS}{K_S}+\frac{TF}{K_F}}$

From Free $p$H Scale to Total $p$H scale: $FREEtoTOT = 1 + \dfrac{TS}{K_S}$

- $K_S$ equilibrium constant of dissociation of HSO$_4^-$ from Dickson (1990a) in mol kg$^{-1}$:

$$K_S = \frac{-4276.1}{T_{(K)}} + 141.328 - 23.093 \cdot \log(T_{(K)}) + \left(324.57 - 47.986 \cdot \log(T_{(K)}) - \frac{13856}{T_{(K)}}\right) \cdot Ions^2$$

$$K_S = K_S + \left(-771.54 + 114.723 \cdot \log(T_{(K)}) + \frac{35474}{T_{(K)}}\right) \cdot Ions + \frac{-2698}{T_{(K)}} \cdot Ions^{\frac{3}{2}} + \frac{1776}{T_{(K)}} \cdot Ions^2$$

$$K_S = e^{K_S} \cdot (1 - 0.001005 \cdot S)$$

- $K_F$ equilibrium constant of dissociation of hydrogen fluoride (HF) formation from Dickson and Riley (1979) in mol kg$^{-1}$:

$$K_F = e^{\frac{1590.2}{T_{(K)}} - 12.641 + 1.525 \cdot Ions^{\frac{1}{2}}} \cdot (1 - 0.001005 \cdot S)$$

- $K_B$ equilibrium constant of dissociation of boric acid from Dickson (1990b) in mol kg$^{-1}$

$$K_B = (-8966.9 - 2890.53 \cdot S^{\frac{1}{2}} - 77.942 \cdot S + 1.728 \cdot S^{\frac{3}{2}} - 0.0996 \cdot S^2)/T_{(K)}$$

$$K_B = K_B + 148.0248 + 137.1942 \cdot S^{\frac{1}{2}} + 1.62142 \cdot S + (-24.4344 - 25.085 \cdot S^{\frac{1}{2}} - 0.2474 \cdot S)$$
$$\cdot \log(T) + 0.053105 \cdot S^{\frac{1}{2}} \cdot T$$

- $K_0$ constant of CO$_2$ solubility from Weiss (1974) in mol kg$^{-1}$ atm$^{-1}$:

$K_0 = \exp\left(-60.2409 + 93.4517 \cdot \frac{100}{T_{(K)}} + 23.3585 \cdot \log\left(\frac{T_{(K)}}{100}\right) + S \cdot \left(0.023517 - 0.023656 \cdot \frac{T_{(K)}}{100} + 0.0047036 \cdot\right.\right.$
$\left.\left.\left(\frac{T_{(K)}}{100}\right)^2\right)\right)$
•    $K_e$: Dissociation constant of water from Millero (1995) in (mol kg$^{-1}$)$^2$ :
$K_e = \exp(\frac{-13847.26}{T_{(K)}} + 148.9802 - 23.6521 \cdot \log(T_{(K)}) + \left(-5.977 + \frac{118.67}{T_{(K)}} + 1.0495 \cdot \log(T_{(K)})\right) \cdot S^{\frac{1}{2}} -$
$0.01615 \cdot S)$
$K_e = K_e \cdot SWStoTOT$, on total $p$H scale in mol kg$^{-1}$
•    $K_1$ and $K_2$ from Lueker et al. (2000) in mol kg$^{-1}$:
$K_1 = 10^{\frac{-3633.86}{T_{(K)}} + 61.2172 - 9.6777 \cdot \log(T_{(K)}) + 0.011555 \cdot S - 0.0001152 \cdot S^2}$
$K_2 = 10^{\frac{-471.78}{T_{(K)}} + 251.929 - 3.16967 \cdot \log(T_{(K)}) + 0.01781 \cdot S - 0.0001122 \cdot S^2}$
•    $K_{ca}$ for calcite from Mucci (1983) in (mol kg$^{-1}$)$^2$:
$K_{ca}$
$= 10^{-171.9065 - 0.077993 \cdot T_{(K)} + \frac{2839.319}{T_{(K)}} + 71.595 \cdot log10(T_{(K)}) + (-0.77712 + 0.0028426 \cdot T_{(K)} + \frac{178.34}{T_{(K)}} S^{\frac{1}{2}} - 0.07711 \cdot S + 0.0041249 \cdot S^{\frac{3}{2}}}$
•    All the constants are corrected by the effect of hydrostatic pressure:
$R = 83.1451$ in ml bar$^{-1}$ K$^{-1}$ mol$^{-1}$
$lnK_1fac = \frac{\left(25.5 - 0.1271 \cdot T_{(°C)} + 0.5 \cdot \left(\frac{-3.08 + 0.0877 \cdot T_{(°C)}}{1000}\right) \cdot P_{bar}\right) \cdot P_{bar}}{R*T_{(K)}}$; $K_1 = K_1 \cdot e^{lnK_1fac}$
$lnK_2fac = \frac{\left(15.82 - 0.0219 \cdot T_{(°C)} + 0.5 \cdot \left(\frac{1.13 + 0.1475 \cdot T_{(°C)}}{1000}\right) \cdot P_{bar}\right) \cdot P_{bar}}{R \cdot T_{(K)}}$: $K_2 = K_2 \cdot e^{lnK_2fac}$
$lnK_Bfac = \frac{\left(29.48 - 0.1622 \cdot T_{(°C)} + 0.002608 \cdot T_{(°C)}^2 + 0.5 \cdot \left(\frac{-2.84}{1000}\right) \cdot P_{bar}\right) \cdot P_{bar}}{R \cdot T_{(K)}}$; $K_B = K_B \cdot e^{lnK_Bfac}$
$lnK_efac = \frac{\left(20.02 - 0.1119 \cdot T_{(°C)} + 0.001409 \cdot T_{(°C)}^2 + 0.5 \cdot \left(\frac{-5.13 + 0.0794 \cdot T_{(°C)}}{1000}\right) \cdot P_{bar}\right) \cdot P_{bar}}{R \cdot T_{(K)}}$; $K_e = K_e \cdot e^{lnK_efac}$
$lnK_Ffac = \frac{\left(9.78 - 0.009 \cdot T_{(°C)} + 0.0009429 \cdot T_{(°C)}^2 + 0.5 \cdot \left(\frac{-3.91 + 0.054 \cdot T_{(°C)}}{1000}\right) \cdot P_{bar}\right) \cdot P_{bar}}{R \cdot T_{(K)}}$; $K_F = K_F \cdot e^{lnK_Ffac}$
$lnK_Sfac = \frac{\left(18.03 - 0.0466 \cdot T_{(°C)} + 0.000316 \cdot T_{(°C)}^2 + 0.5 \cdot \left(\frac{-4.53 + 0.009 \cdot T_{(°C)}}{1000}\right) \cdot P_{bar}\right) \cdot P_{bar}}{R \cdot T_{(K)}}$; $K_S = K_S \cdot e^{lnK_Sfac}$
$lnK_cafac = \frac{\left(48.76 - 0.5304 \cdot T_{(°C)} + 0.5 \cdot \left(\frac{-11.76 + 0.3692 \cdot T_{(°C)}}{1000}\right) \cdot P_{bar}\right) \cdot P_{bar}}{R \cdot T_{(K)}}$; $K_{ca} = K_{ca} \cdot e^{lnK_cafac}$
•    Calculation of the Fugacity factor:
We suppose that the pressure is at one atmosphere or close to it (Weiss, 1974):
$P_{atm} = 1.01325 \; bar$
$delta = 57.7 - 0.118 \cdot T$ in cm$^3$ mol$^{-1}$
$b = -1636.75 + 12.0408 \cdot T - 0.0327957 \cdot T^2 + 3.16528 \cdot 0.00001 \cdot T^3$ in cm$^3$ mol$^{-1}$
$FugFac = exp^{\frac{(b + 2 \cdot delta) \cdot P_{atm}}{R \cdot T}}$

**A.2. Resolution of carbonate system**

To resolve the carbonate system, we calculate the $deltapH$, which is the difference of $pH$ between two iterations of the model. We initialize the run by imposing a $pH$ value of 8.

$if\ (nbiter < 1)\ pH = 8$

$pHtol = 0.001$ ! tolerance for iterations end

$deltapH = pHtol + 1$

$do\ while\ (abs(deltapH) > 0.0001)$

$\quad H = 10^{-pH}$

$\quad Denom = H^2 + K_1 \cdot H + K_1 \cdot K_2$

$\quad CAlk = DIC \cdot K_1 \cdot \left(\dfrac{H + 2 \cdot K_2}{Denom}\right)$

$\quad BAlk = \dfrac{TB \cdot K_B}{K_B + H}$

$\quad OH = \dfrac{K_e}{H}$

$\quad FreetoTot = 1 + \dfrac{TS}{K_S}$

$\quad Hfree = \dfrac{H}{FreetoTot}$

$\quad HSO_4 = \dfrac{TS}{1 + \dfrac{K_S}{Hfree}}$

$\quad HF = \dfrac{TF}{1 + \dfrac{K_F}{Hfree}}$

$\quad Residual = TA - CAlk - BAlk - OH + Hfree + HSO_4 + HF$

$\quad Slope = DIC \cdot H \cdot K_1 \cdot (H^2 + K_1 \cdot K_2 + 4 \cdot H \cdot K_2)$

$\quad Slope = \dfrac{Slope}{Denom^2} + OH + H + \dfrac{BAlk \cdot H}{K_B + H}$

$\quad Slope = \log10 \cdot Slope$

$\quad deltapH = Residual/Slope$ ! this is Newton's method

$\quad do\ while\ (abs(deltapH) > 1)\ deltapH = \dfrac{deltapH}{2}$ ! to keep the jump from being too big

$enddo$

$pH = pH + deltapH$ ! Is on the same scale as $K_1$ and $K_2$ were calculated, *i.e.* total $pH$ scale

$pCO_2 = \left(\dfrac{DIC \cdot H^2}{H^2 + K_1 \cdot H + K_1 \cdot K_2}\right) \cdot \dfrac{10^6}{K_0 \cdot FugFac}$ ! in µatm

$CO_2 = \dfrac{DIC \cdot 10^6}{1 + \dfrac{K_1}{H} + \dfrac{K_1 \cdot K_2}{H^2}}$

$HCO_3 = \dfrac{K_1 \cdot CO_2}{H}$

$CO_3 = \dfrac{K_2 \cdot HCO_3}{H}$

$Omega = \dfrac{Ca \cdot CO_3 \cdot 10^{-6}}{K_{ca}}$

**Appendix B: Biogeochemical model variables and parameters**

**Table B1: Initial conditions of the state variables of Eco3M-CarbOx model (*diagnostic variables)**

| Variables | Name | Unit | values |
|---|---|---|---|
| **Picophytoplankton** | *PicoC* | mmolC m$^{-3}$ | 0.0480 |
| | *PicoN* | mmolN m$^{-3}$ | 0.0092 |
| | *PicoP* | mmolP m$^{-3}$ | 0.0003 |
| **Diatom** | *DiaC* | mmolC m$^{-3}$ | 0.0571 |
| | *DiaN* | mmolN m$^{-3}$ | 0.0089 |
| | *DiaP* | mmolP m$^{-3}$ | 0.0007 |
| **Bacteria** | *BacC* | mmolC m$^{-3}$ | 0.1083 |
| | *BacN* | mmolN m$^{-3}$ | 0.0379 |
| | *BacP* | mmolP m$^{-3}$ | 0.0039 |
| **DPOM** <br> **Detrital Particulate organic matter** | *DPOC* | mmolC m$^{-3}$ | 0.1252 |
| | *DPON* | mmolN m$^{-3}$ | 0.0307 |
| | *DPOP* | mmolP m$^{-3}$ | 0.0021 |
| **LDOM** <br> **Labile Dissolved organic matter** | *LDOC* | mmolC m$^{-3}$ | 1.0990 |
| | *LDON* | mmolN m$^{-3}$ | 8.7980 |
| | *LDOP* | mmolP m$^{-3}$ | 0.0018 |
| **DIM** <br> **Dissolved inorganic matter** | $NH_4$ | mmolN m$^{-3}$ | 0.3375 |
| | $NO_3$ | mmolN m$^{-3}$ | 0.6723 |
| | $PO_4$ | mmolP m$^{-3}$ | 0.7150 |
| | DO | mmolO m$^{-3}$ | 257.00 |
| | DIC | µmolC kg$^{-1}$ | 2358.4 |
| **Total alkalinity** | TA | µmolC kg$^{-1}$ | 2660.5 |
| **Sea water partial pressure of CO₂** | *p*CO₂ | µatm | 371.28 |
| ***p*H** | *p*H | - | 8.1099 |
| **calcium carbonate** | $CaCO_3$ | mmol m$^{-3}$ | 1.0000 |
| ***Picophytoplankton chlorophyll*** | *PicoChl* | mgChl m$^{-3}$ | 0.0193 |
| ***Diatom chlorophyll*** | *DiaChl* | mgChl m$^{-3}$ | 0.0229 |
| **Number of bacteria*** | NBA | $10^{12}$ cell m$^{-3}$ | 0.2000 |



**Table B2: Balance equations of Eco3M-CarbOx model**

| Variables | Balance equation |
|---|---|
| **Pico-phytoplankton** | $\frac{\partial PicoC}{\partial t} = R_{PP}^{PicoC} - R_{resp}^{PicoC} - R_{exu}^{PicoC} - R_{Gr}$ <br><br> $\frac{\partial PicoN}{\partial t} = R_{uptPicoN}^{NH_4} + R_{uptPicoN}^{NO_3} - R_{exu}^{PicoN} - R_{Gr}$ <br><br> $\frac{\partial PicoP}{\partial t} = R_{uptPicoP}^{PO_4} - R_{exu}^{PicoP} - R_{Gr}$ <br><br> $PicoChl = Q_C^N \cdot \left( Q_{N,min}^{Chla} + f_Q \cdot \left( Q_{N,max}^{Chla} - Q_{N,min}^{Chla} \right) \right) \cdot PicoC$ |
| **Diatom** | $\frac{\partial DiaC}{\partial t} = R_{PP}^{DiaC} - R_{resp}^{DiaC} - R_{exu}^{DiaC} - R_{Gr}$ <br><br> $\frac{\partial DiaN}{\partial t} = R_{uptDiaN}^{NH_4} + R_{uptDiaN}^{NO_3} - R_{exu}^{DiaN} - R_{Gr}$ <br><br> $\frac{\partial DiaP}{\partial t} = R_{uptDiaP}^{PO_4} - R_{exu}^{DiaP} - R_{Gr}$ <br><br> $DiaChl = Q_C^N \cdot \left( Q_{N,min}^{Chla} + f_Q \cdot \left( Q_{N,max}^{Chla} - Q_{N,min}^{Chla} \right) \right) \cdot DiaC$ |
| **Bacteria** | $\frac{\partial BacC}{\partial t} = R_{uptBac}^{DPOC} + R_{uptBac}^{LDOC} - R_{BR} - R_{Gr}^{BacC}$ <br><br> $\frac{\partial BacN}{\partial t} = R_{uptBac}^{DPON} + R_{uptBac}^{LDON} + R_{uptBac}^{NH4} - R_{miner}^{NH4} - R_{Gr}^{BacN}$ <br><br> $\frac{\partial BacP}{\partial t} = R_{uptBac}^{DPOP} + R_{uptBac}^{LDOP} + R_{uptBac}^{PO4} - R_{miner}^{PO4} - R_{Gr}^{BacP}$ |
| **DPOM** | $\frac{\partial DPOC}{\partial t} = R_{pf} + R_m - R_{Gr} - R_{uptBac}^{DPOC}$ <br><br> $\frac{\partial DPON}{\partial t} = R_{pf} + R_m - R_{Gr} - R_{uptBac}^{DPON}$ <br><br> $\frac{\partial DPOP}{\partial t} = R_{pf} + R_m - R_{Gr} - R_{uptBac}^{DPOP}$ |
| **LDOM** | $\frac{\partial LDOC}{\partial t} = R_{exu}^{PicoC} + R_{exu}^{DiaC} + R_{exu}^{LDOC} - R_{uptBac}^{LDOC}$ <br><br> $\frac{\partial LDON}{\partial t} = R_{exu}^{PicoN} + R_{exu}^{DiaN} + R_{exu}^{LDON} - R_{uptBac}^{LDON}$ <br><br> $\frac{\partial LDOP}{\partial t} = R_{exu}^{PicoP} + R_{exu}^{DiaP} + R_{exu}^{LDOP} - R_{uptBac}^{LDOP}$ |
| **NH₄** | $\frac{\partial NH_4}{\partial t} = R_{excr}^{NH_4} + R_{miner}^{NH_4} - R_{nit} - \sum R_{uptPhyN}^{NH_4} - R_{uptBac}^{NH_4}$ |
| **NO₃** | $\frac{\partial NO_3}{\partial t} = R_{nit} - \sum R_{uptPhyN}^{NO_3}$ |
| **PO₄** | $\frac{\partial PO_4}{\partial t} = R_{excr}^{PO_4} + R_{miner}^{PO_4} - \sum R_{uptPhyP}^{PO_4} - R_{uptBac}^{PO_4}$ |
| **DO** | $\frac{\partial DO}{\partial t} = R_{aera} + \left(\frac{O}{C}\right) \cdot R_{PP}^{PhyC} + \left(\frac{O}{N}\right) \cdot R_{uptPhyN}^{NO_3} - \left(\frac{O}{C}\right) \cdot R_{resp}^{PhyC} - \left(\frac{O}{C}\right) \cdot R_{excr}^{DIC} - \left(\frac{O}{C}\right) \cdot R_{BR} - \left(\frac{O}{N}\right) \cdot R_{nit}$ |
| **DIC** | $\frac{\partial DIC}{\partial t} = R_{aera} + R_{resp}^{PhyC} + R_{BR} + R_{excr}^{DIC} - R_{PP}^{PhyC} - R_{precip} + R_{diss}$ |
| **TA** | $\frac{\partial TA}{\partial t} = 2 \cdot R_{diss} + \left( R_{uptPhyN}^{NO_3} + R_{uptPhyP}^{PO_4} - R_{uptPhyN}^{NH_4} \right) + R_{miner}^{NH_4} - 2 \cdot R_{precip} - 2. R_{nit}$ |


     **Table B3: Biogeochemical processes simulated by the Eco3M-CarbOx model**

| Notation | Biogeochemical processes | Unit | Formulation | |
|---|---|---|---|---|
| $R_{PP}^{Phy}$ | Primary production | molC m$^{-3}$ s$^{-1}$ | $R_{PP}^{PhyC} = P_{max} \cdot f_T^{PP} \cdot f_I \cdot PhyC$ | $f_Q = min[f_Q^N, f_Q^P]; f_Q^X = \frac{Q_C^X - Q_{C,min}^X}{Q_C^X - Q_{C,min}^X + \beta_X}$ $f_T^{PP} = max\left(\frac{2 \cdot (1-b)\frac{(T-T_{let})}{(T_{opt}-T_{let})}}{\left(\frac{(T-T_{let})}{(T_{opt}-T_{let})}\right)^2 - 2 \cdot b \cdot \frac{(T-T_{let})}{(T_{opt}-T_{let})} + 1}; 0\right)$ $f_I = \left[1 - exp\left(\frac{-\alpha_{Chla} \cdot E_{PAR} \cdot Q_C^{Chla}}{P_{max} \cdot f_Q \cdot f_T^{PP}}\right)\right]$ |
| $R_{resp}^{Phy}$ | Phytoplankton respiration | molC m$^{-3}$ s$^{-1}$ | $R_{resp}^{PhyC} = k_r^{PhyC} \cdot PhyC$ | |
| $R_{uptPhy}^{NH_4}$ | NH$_4$ uptake by phytoplankton | molX m$^{-3}$ s$^{-1}$ | $R_{uptPhyN}^{NH_4} = V_{N,max} \cdot \frac{NH_4}{NH_4 + K_{NH_4}}$ | $V_{N,max} = Q_{C,max}^N \cdot R_{PP}^{Phy}$ |
| $R_{uptPhy}^{NO_3}$ | NO$_3$ uptake by phytoplankton | molN m$^{-3}$ s$^{-1}$ | $R_{uptPhyN}^{NO_3} = V_{N,max} \cdot \frac{NO_3}{NO_3 + K_{NO_3}} \cdot \left(1 - \frac{I_{in} \cdot NH_4}{NH_4 + K_{in}}\right)$ | |
| $R_{uptPhy}^{PO_4}$ | PO$_4$ uptake by phytoplankton | molP m$^{-3}$ s$^{-1}$ | $R_{uptPhyP}^{PO_4} = V_{P,max} \cdot \frac{PO_4}{PO_4 + K_{PO_4}} \cdot$ | $V_{P,max} = Q_{C,max}^P \cdot R_{PP}^{Phy}$ |
| $R_{exu}^{PhyC}$ | Phytoplankton exudation as LDOC | molC m$^{-3}$ s$^{-1}$ | $R_{exu}^{PhyC} = \left(1 - f_Q\right) \cdot R_{PP}^{Phy}$ | |
| $R_{exr}^{PhyX}$ | Phytoplankton exudation as LDON or LDOP | molX m$^{-3}$ s$^{-1}$ | $R_{exu}^{PhyX} = \left(1 - h_Q^X\right) \cdot R_{uptX}^{Phy}$ | $h_Q^X = \frac{Q_{C,max}^X - Q_C^X}{Q_{C,max}^X - Q_{C,min}^X}$ |
| $R_{BP}$ | Bacterial production | cell m$^{-3}$ s$^{-1}$ | $R_{BP} = \mu_{max}^{Ba} \cdot f_Q^{Ba} \cdot f_T^{Ba} \cdot NBA$ | $f_T^{Ba} = Q_{10}^{\frac{(T-T_{rem})}{10}}$ $f_Q^{BA} = min\left[1 - \frac{Q_{C,min}^{BA}}{Q_C^{BA}}, 1 - \frac{Q_{N,min}^{BA}}{Q_N^{BA}}, 1 - \frac{Q_{P,min}^{BA}}{Q_P^{BA}}\right]$ |
| $R_{BR}$ | Bacterial respiration | molC m$^{-3}$ s$^{-1}$ | $R_{BR} = \rho_g^{Ba} \cdot Q_C^{Ba} \cdot R_{BP} + \rho_r^{Ba} \cdot \left(Q_C^{Ba} - Q_{C,min}^{Ba}\right) \cdot NBA$ | |
| $R_{uptBac}^X$ | X uptake by bacteria | molX m$^{-3}$ s$^{-1}$ | $R_{uptBac}^X = V_{max}^{BA} \cdot \frac{X}{X + K_X^{Ba}} \cdot f_T^{Ba} \cdot NBA$ | |
| $R_{Gr}^{Phy}$ | Phytoplankton grazing by zooplankton | molX m$^{-3}$ s$^{-1}$ | $R_{Gr}^{Phy} = g_{Phy} \cdot f_{Gr} \cdot Phy$ | $f_{Gr} = \frac{Phy}{Phy + DPOM}$ |
| $R_{Gr}^{DPOM}$ | DPOM grazing by zooplankton | molX m$^{-3}$ s$^{-1}$ | $R_{Gr}^{DPOM} = g_{DPOM} \cdot f_{Gr} \cdot DPOM$ | $g_{DPOM} = \frac{g_{Pico} \cdot Pico + g_{Dia} \cdot Dia}{Pico + Dia};$ $f_{Gr} = \frac{DPOM}{Phy + DPOM}$ |

| | | | | |
|---|---|---|---|---|
| $R_{Gr}^{Bac}$ | Bacterial grazing by zooplankton | molX m$^{-3}$ s$^{-1}$ | $R_{Gr}^{Ba} = R_{BP} \cdot \frac{Bac}{NBA}$ | |
| $R_{excr}^{DIM}$ | Zooplankton excretion as DIC, NH$_4$ and PO$_4$ | molX m$^{-3}$ s$^{-1}$ | $R_{excr}^{DIM} = \varepsilon_{DIM} \cdot d_X \cdot (1 - k_{X,zoo}) \cdot (R_{Gr}^{Phy} + R_{Gr}^{DPOM} + R_{Gr}^{Ba})$ | |
| $R_{exu}^{LDOM}$ | Zooplankton exudation as LDOM | molX m$^{-3}$ s$^{-1}$ | $R_{exu}^{LDOM} = (1 - \varepsilon_{DIM}) \cdot d_X \cdot (1 - k_{X,zoo}) \cdot (R_{Gr}^{Phy} + R_{Gr}^{DPOM} + R_{Gr}^{Ba})$ | |
| $R_{pf}$ | Zooplankton egestion | molX m$^{-3}$ s$^{-1}$ | $R_{pf} = (1 - d_X) \cdot (R_{Gr}^{Phy} + R_{Gr}^{DPOM} + R_{Gr}^{Ba})$ | |
| $R_m$ | Zooplankton mortality | molX m$^{-3}$ s$^{-1}$ | $R_m = d_X \cdot k_{X,zoo} \cdot (R_{Gr}^{Phy} + R_{Gr}^{DPOM} + R_{Gr}^{Ba})$ | |
| $R_{miner}$ | Mineralization of organic matter by bacteria | molX m$^{-3}$ s$^{-1}$ | $R_{miner}^{X} = (1 - h_Q^{Ba}) \cdot (R_{uptBac}^{LDOM} + R_{uptBac}^{DPOM} + R_{uptBac}^{DIM})$ | |
| $R_{nit}$ | Nitrification | molX m$^{-3}$ s$^{-1}$ | $R_{nit} = k_{nit} \cdot f_T^{Ba} \cdot \frac{DO}{DO + K_{DO}} \cdot NH_4$ | |
| $R_{diss}$ | Carbonate dissolution | molC m$^{-3}$ s$^{-1}$ | $R_{diss} = (1 - \Omega_C) \cdot k_{diss} \cdot [CaCO_3]$ | $\Omega_C$ = aragonite saturation |
| $R_{precip}$ | Carbonate precipitation | molC m$^{-3}$ s$^{-1}$ | $R_{precip} = k_{precip} \cdot \frac{(\Omega_C - 1)}{K_C + (\Omega_C - 1)} \cdot (R_{PP}^{Phy} - R_{resp}^{Phy})$ | |
| $R_{aera}$ | Gas exchange with atmosphere of DO or CO$_2$ | molX m$^{-3}$ s$^{-1}$ | $R_{aera} = \frac{k_{ex}}{H} \cdot ([DO]_{sea} - [DO]_{sat})$ <br> $R_{aera} = \frac{k_{ex}}{H} \cdot \alpha \cdot (pCO_{2,sea} - pCO_{2,atm})$ | $k_{ex} = 0.31 \cdot U_{10}^2 \cdot \frac{660^{0.5}}{Sc}$ <br> H (depth), $U_{10}$ (wind velocity) <br> $\alpha$ (solubility), $Sc$ (Schmidt number) and $[DO]_{sat}$ are function of T and S |


 **Table B4: Value of parameters**

| Parameters | | Pico | Dia | Unit | Reference |
|---|---|---|---|---|---|
| $P_m^C$ | Maximal production | 1.815 | 1.057 | d$^{-1}$ | Sarthou et al. (2005) |
| $m_1$ | Fraction of the solar energy flux photosynthetically available | 0.43 | 0.43 | - | Tett (1987) |
| $m_2$ | Sea surface reflection | 0.95 | 0.95 | - | Tett (1987) |
| $m_3$ | More rapid attenuation of polychromatic light near the sea surface | 1.0 | 1.0 | - | Tett (1987) |
| $\alpha_{Chla}$ | Chlorophyll-specific light absorption coefficient | 8 10$^{-6}$ | 5 10$^{-6}$ | m² molC (gChla J)$^{-1}$ | Leblanc et al. (2018) |
| $T_{opt}$ | Temperature optimal of growth | 16.0 | 13.0 | °C | - |
| $T_{let}$ | Lethal temperature | 11.0 | 9.0 | °C | - |
| b | Shape factor for temperature curve | 0.5 | 0.8 | - | Lacroix and Grégoire (2002) |
| $\beta_N$ | Coefficient in the quota function | 0.0072 | 0.002 | molN molC$^{-1}$ | Leblanc et al. (2018) |
| $\beta_P$ | Coefficient in the quota function | 0.0002 | 0.0005 | molP molC$^{-1}$ | Leblanc et al. (2018) |
| $Q_{C,min}^N$ | Minimum phytoplankton N:C ratio | 0.115 | 0.07 | molN molC$^{-1}$ | Leblanc et al. (2018) |
| $Q_{C,max}^N$ | Maximum phytoplankton N:C ratio | 0.229 | 0.18 | molN molC$^{-1}$ | Leblanc et al. (2018) |
| $Q_{C,min}^P$ | Minimum phytoplankton P:C ratio | 0.0015 | 0.006 | molP molC$^{-1}$ | Auger et al. (2011); Campbell et al. (2013) |
| $Q_{C,max}^P$ | Maximum phytoplankton P:C ratio | 0.0068 | 0.016 | molP molC$^{-1}$ | Auger et al. (2011); Campbell et al. (2013) |
| $Q_{N,min}^{Chla}$ | Minimum phytoplankton Chl:N ratio | 1.0 | 1.0 | gChl molN$^{-1}$ | Leblanc et al. (2018)** |
| $Q_{N,max}^{Chla}$ | Maximum phytoplankton Chl:N ratio | 2.2 | 2.7 | gChl molN$^{-1}$ | Leblanc et al. (2018) |
| $k_r^{PhyC}$ | Phytoplankton respiration rate | 0.099 | 0.099 | d$^{-1}$ | Faure et al. (2010) |
| $K_{NO_3}$ | Half saturation constant for NO$_3$ | 0.73 | 1.0 | mmolN m$^{-3}$ | Leblanc et al. (2018) |
| $K_{NH_4}$ | Half saturation constant for NH$_4$ | 0.07 | 0.015 | mmolN m$^{-3}$ | Leblanc et al. (2018) |
| $K_{PO_4}$ | Half saturation constant for PO$_4$ | 0.008 | 0.01 | mmolP m$^{-3}$ | Leblanc et al. (2018)** |
| $I_{in}$ | Factor of inhibition | 0.82 | 0.82 | - | Harrison et al. (1996) |
| $K_{in}$ | Amount of NH$_4$ from which assimilation by NO$_3$ is reduced. | 0.578 | 0.578 | mmolN m$^{-3}$ | Harrison et al. (1996) |
| g | Grazing rate | 1.452 | 0.846 | d$^{-1}$ | Gutiérrez-Rodríguez et al. (2011) |

** calibrated from parameter used in the cited article

**Table B5: Value of parameters (continue)**

| Parameters | | Value | Unit | Reference |
|---|---|---|---|---|
| $NBA$ | Number of bacteria | 0.20 | $10^{12}$ cell m$^{-3}$ | Moran (2015) |
| $\mu_{max}^{Ba}$ | Bacterial production rate | 8.36 | d$^{-1}$ | Fraysse et al. (2013) |
| $Q_{C,min}^{BA}$ | Minimum bacteria C:cell ratio | 0.49 | mmolC ($10^{12}$ cell)$^{-1}$ | Fukuda et al. (1998) |
| $Q_{N,min}^{BA}$ | Minimum bacteria N:cell ratio | 0.09 | mmolN ($10^{12}$ cell)$^{-1}$ | Fukuda et al. (1998) |
| $Q_{N,max}^{BA}$ | Maximum bacteria N:cell ratio | 0.23 | mmolN ($10^{12}$ cell)$^{-1}$ | Fukuda et al. (1998) |
| $Q_{P,min}^{BA}$ | Minimum bacteria P:cell ratio | 0.005 | mmolP ($10^{12}$ cell)$^{-1}$ | Fraysse et al. (2013) |
| $Q_{P,max}^{BA}$ | Maximum bacteria P:cell ratio | 0.02 | mmolP ($10^{12}$ cell)$^{-1}$ | Fraysse et al. (2013) |
| $\rho_g^{Ba}$ | Factor of carbon respired by bacteria | 0.60 | - | Thingstad (1987) |
| $\rho_r^{Ba}$ | Respiration rate of bacteria | 0.01 | d$^{-1}$ | Thingstad (1987) |
| $V_{DPOC,max}^{BA}$ | Maximum DPOC uptake by bacteria | 0.029 | mmolC ($10^{12}$ cell)$^{-1}$ d$^{-1}$ | Campbell et al. (2013) |
| $V_{LDOC,max}^{BA}$ | Maximum LDOC uptake by bacteria | 16.33 | mmolC ($10^{12}$ cell)$^{-1}$ d$^{-1}$ | Campbell et al. (2013) |
| $V_{DPON,max}^{BA}$ | Maximum DPON uptake by bacteria | 0.05 | mmolN ($10^{12}$ cell)$^{-1}$ d$^{-1}$ | Faure et al. (2010) |
| $V_{LDON,max}^{BA}$ | Maximum LDON uptake by bacteria | 0.32 | mmolN ($10^{12}$ cell)$^{-1}$ d$^{-1}$ | Faure et al. (2010) |
| $V_{NH_4,max}^{BA}$ | Maximum NH$_4$ uptake by bacteria | 0.32 | mmolN ($10^{12}$ cell)$^{-1}$ d$^{-1}$ | Faure et al. (2010) |
| $V_{DPOP,max}^{BA}$ | Maximum DPOP uptake by bacteria | 0.01 | mmolP ($10^{12}$ cell)$^{-1}$ d$^{-1}$ | Thingstad (1987) |
| $V_{LDOP,max}^{BA}$ | Maximum LDOP uptake by bacteria | 0.48 | mmolP ($10^{12}$ cell)$^{-1}$ d$^{-1}$ | Thingstad (1987) |
| $V_{PO_4,max}^{BA}$ | Maximum PO$_4$ uptake by bacteria | 0.48 | mmolP ($10^{12}$ cell)$^{-1}$ d$^{-1}$ | Thingstad (1987) |
| $K_{DPOC}^{BA}$ | Half-saturation constant for DPOC | 10.0 | mmolC m$^{-3}$ | Faure et al. (2010) |
| $K_{LDOC}^{BA}$ | Half-saturation constant for LDOC | 25.0 | mmolC m$^{-3}$ | - |
| $K_{DPON}^{BA}$ | Half-saturation constant for DPON | 0.50 | mmolN m$^{-3}$ | - |
| $K_{LDON}^{BA}$ | Half-saturation constant for LDON | 0.50 | mmolN m$^{-3}$ | - |
| $K_{NH_4}^{BA}$ | Half-saturation constant for NH$_4$ | 0.15 | mmolN m$^{-3}$ | Leblanc et al. (2018) |
| $K_{DPOP}^{BA}$ | Half-saturation constant for DPOP | 0.08 | mmolP m$^{-3}$ | - |
| $K_{LDOP}^{BA}$ | Half-saturation constant for LDOP | 0.08 | mmolP m$^{-3}$ | Leblanc et al. (2018) |
| $K_{PO_4}^{BA}$ | Half-saturation constant for PO$_4$ | 0.02 | mmolP m$^{-3}$ | Campbell et al. (2013) |
| $\varepsilon_{DIC}$ | fraction excretion of DIC | 0.31 | - | Faure et al. (2010) |
| $\varepsilon_{NH_4}$ | fraction excretion of NH$_4$ | 0.50 | - | Faure et al. (2010) |
| $\varepsilon_{PO_4}$ | Fraction excretion of PO$_4$ | 0.50 | - | Fraysse et al. (2013) |
| $d_C$ | Fraction of C assimilated | 0.92 | - | Gerber and Gerber (1979) |
| $d_N$ | Fraction of N assimilated | 0.95 | - | Faure et al. (2010) |
| $d_P$ | Fraction of P assimilated | 0.95 | - | Fraysse et al. (2013) |
| $k_{C,zoo}$ | Net C growth efficiency | 0.40 | - | Gerber and Gerber (1979) |
| $k_{N,zoo}$ | Net N growth efficiency | 0.44 | - | Le Borgne and Rodier (1997) |
| $k_{P,zoo}$ | Net P growth efficiency | 0.37 | - | Le Borgne (1982) |
| Q$_{10}$ | Temperature coefficient | 2.0 | - | - |
| T$_{rem}$ | Reference temperature for mineralization | 20.0 | °C | - |
| $k_{nit}$ | Nitrification rate | 0.05 | d$^{-1}$ | Lacroix and Grégoire (2002) |
| T$_{nit}$ | Reference temperature for nitrification | 10.0 | °C | - |
| $K_{DO}$ | Half-saturation constant DO | 30.0 | mmolO$_2$ m$^{-3}$ | Tett (1990) |

| | | | | | |
|---|---|---|---|---|---|
| $k_{diss}$ | Dissolution rate | 10.9 | $d^{-1}$ | Gehlen et al. (2007) | |
| $k_{precip}$ | Fraction of PIC to LPOC | 0.02 | - | Marty et al. (2002) | |
| $K_C$ | Half-saturation constant of CaCO$_3$ precipitation | 0.40 | $(\mu mol\ kg^{-1})^2$ | | |
| $\left(\dfrac{O}{C}\right)$ | Ratio O:C | 1.0 | - | - | |
| $\left(\dfrac{O}{N}\right)$ | Ratio O:N | 2.0 | - | - | |


**Appendix C: Short User Manual**

After uploading the whole archive on the zenodo site (ref. doi: 10.5281/zenodo.3757677), the exact version of the Eco3M-CarbOx code used in this study can be run as following:

make !two executable files will be created : eco3M_ini.exe and eco3M.exe

- the file config.ini allows to define: the time, time step, and save time of simulation variables biogeochemical process
- Results files are stocked in "SORTIES" directory
- Boundary conditions and forcings data are stocked in "DATA" directory
- All subroutines of biogeochemical processes are stocked in "F_PROCESS" directory

For further information, please contact Dr. Frédéric DIAZ (frederic.diaz@univ-amu.fr) or Dr. Christel PINAZO (christel.pinazo@univ-amu.fr)

**Code availability**

Eco3M is freely available under CeCILL license agreement (a French equivalent to the L-GPL license; http://cecill.info/licences/Licence_CeCILL_V1.1-US.html; last access: 10 February 2020). The Eco3M-CarbOx model is written in Fortran-90/95 and the plotting code is written in Matlab®. The exact version of the model used to produce the results presented in this paper is archived on Zenodo (DOI: 10.5281/zenodo.3757677) (last access: 24 August 2020). A short User Manual is given in the Appendix C of this study.

**Author contributions**

KL-S: conceptualization, formal analysis, methodology, software, visualisation, writing- original draft preparation, writing- review & editing. FD: conceptualization, supervision, writing- original draft preparation, writing- review & editing. CW-R: data curation, writing review and editing, TW: data curation, writing review and editing, DL: Funding acquisition, Project administration, data curation, CY: Software, resources, IX-R: Funding acquisition, Project administration, data curation, BN: writing- review & editing, AA: data curation, CP: conceptualization, supervision, writing- original draft preparation, writing- review & editing.

**Competing interests**

The authors declare that they have no conflict of interest.

**Financial support**

This study is part of the AMC project (Aix-Marseille Carbon Pilot Study, 2016-2019) funded and performed in the framework of the Labex OT-MED (ANR-11-LABEX-0061, part of the "Investissement d'Avenir" program through the A*MIDEX project ANR-11-IDEX-0001-02), funded by the French National Research Agency (ANR). The project leading to this publication has received funding from the European FEDER Fund under project 1166-39417.

**Acknowledgements**

We thank the National Service d'Observation en MILieu Littoral (SOMLIT) for its permission to use SOLEMIO data. We wish to thank the crewmembers of the R.V. 'Antedon II', operated by the DT-INSU, for making these samplings

possible. We wish to acknowledge the team of the SAM platform (Service Atmosphère Mer) of MIO institute for their
helping in field work. For the collection and analyses of the seawater sample, we thank Michel Lafont and Véronique
Lagadec of the PACEM (Plateforme Analytique de Chimie des Environnenments Marins) platform of MIO institute
and also the SNAPO-CO2 at LOCEAN, Paris. The SNAPO-CO2 service at LOCEAN is supported by CNRS-INSU
and OSU Ecce-Terra.
We acknowledge the staff of the "Cluster de calcul intensif HPC" Platform of the OSU Institut PYTHEAS (Aix-
Marseille Université, INSU-CNRS) for providing the computing facilities. We gratefully acknowledge Julien Lecubin
from the Service Informatique de OSU Institut PYTHEAS (SIP) for their technical assistance. Moreover, we thank
Camille Mazoyer and Claire Seceh for their contribution on the Eco3M-CarbOx model development.

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

**Tables**

|  | Temperature | Wind | River input | Atmospheric CO$_2$ |
|---|---|---|---|---|
| **S0 – Reference** | In situ data of 2017 | WRF model 2017 | No | CAV station 2017 |
| **S1 - T increases** | In situ data of 2017 +1.5°C | WRF model 2017 | No | CAV station 2017 |
| **S2 - Wind constant** | In situ data of 2017 | 7 m s$^{-1}$ | No | CAV station 2017 |
| **S3 - Wind events** | In situ data of 2017 | 3 days at 20 m s$^{-1}$ | No | CAV station 2017 |
| **S4 - NO$_3$** | In situ data of 2017 | WRF model 2017 | Yes, NO$_3$ | CAV station 2017 |
| **S5 - Non-urban** | In situ data of 2017 | WRF model 2017 | No | OHP station 2017 |

**Table 1: Forcing of the different scenarios (S) simulated with the model. See section 2.4 for details of scenarios.**

|  | Chl | seawater pCO$_2$ | pH | DIC | TA |
|---|---|---|---|---|---|
| **Obs min-max** | [0.10– 1.71] | [358 – 471] | [8.014 – 8.114] | [2260 – 2348] | [2561 – 2624] |
| **Mod min-max** | [0.03 – 0.73] | [331 – 522] | [7.979 – 8.171] | [2220 – 2323] | [2560– 2572] |
| **Bias** | -0.22 | 22.47 | -0.016 | -8.48 | -24.91 |
| **WSS** | 0.36 | 0.69* | 0.75* | 0.71* | 0.43 |
| **N** | 22 | 20 | 21 | 20 | 20 |

**Table 2: Statistical evaluation of observations *vs*. model for 2017 year: observed and simulated minimum and maximum**
**values, WSS = Wilmott Skill Score, N = number of measurements. Units of bias are those of modeled variables: chlorophyll**
***a* (Chl, mg m$^{-3}$), seawater pressure of CO$_2$ (seawater *p*CO$_2$, µatm), *p*H, dissolved inorganic carbon (DIC, µmol kg$^{-1}$) and total**
**alkalinity (TA, µmol kg$^{-1}$). *significant value of WSS (> 0.70).**

|  |  | Aeration | GPP | R$_A$ | R$_H$ | R | NEP |
|---|---|---|---|---|---|---|---|
|  | **Year** | 0.017 | -0.413 | 0.065 | 0.348 | 0.413 | 0 |
| **Mean flux** | **MWC** | -0.245 | -0.314 | 0.052 | 0.176 | 0.228 | 0.086 |
|  | **SWC** | 0.405 | -0.521 | 0.079 | 0.555 | 0.634 | -0.113 |
| **Contribution** | **Year** | 78% | 11% | 2% | 9% | 11% | - |

**Table 3: Mean flux values (mmol m$^{-3}$ d$^{-1}$) and the contribution of each process to the DIC variations for the reference**
**simulation over the year and SWC/MWC periods. GPP: Gross primary production, R$_A$: Autotroph respiration, R$_H$:**
**heterotroph respiration, NEP: Net Ecosystem Production**

**Figures**

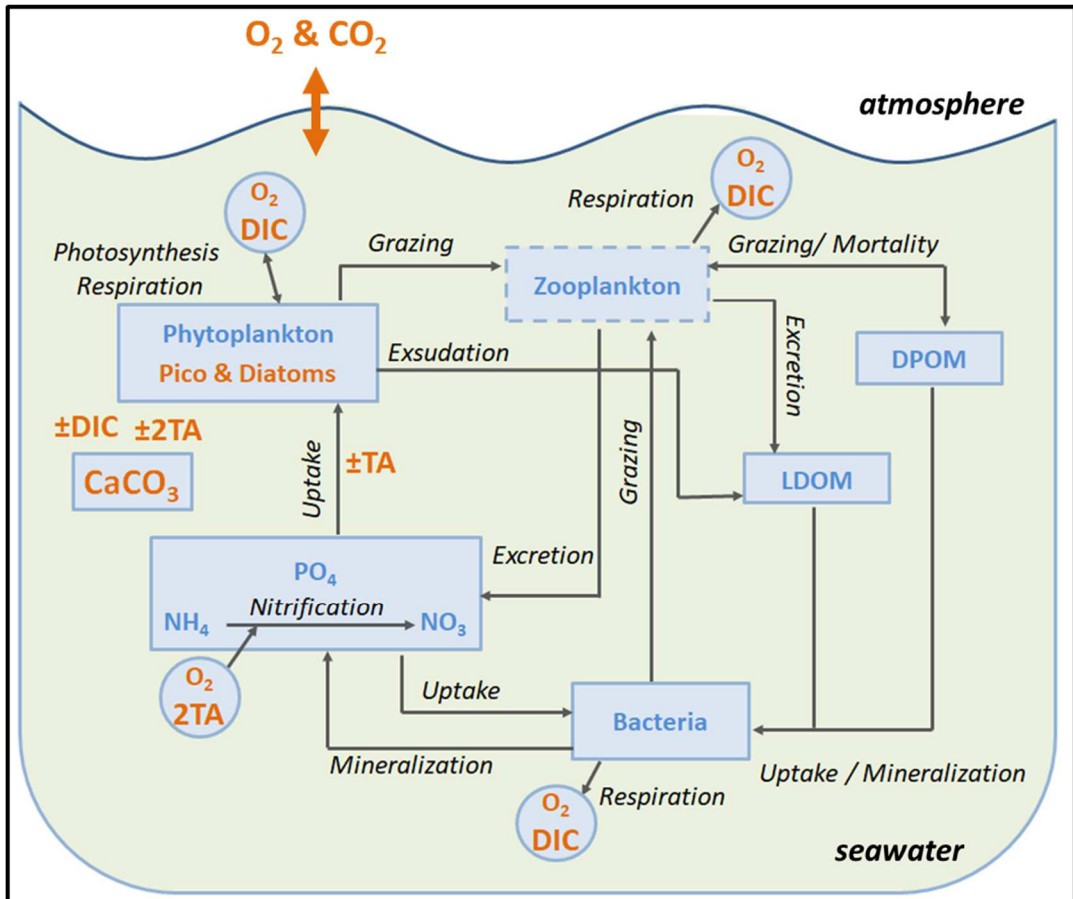

**Figure 1: Schematic diagram of the biogeochemical model Eco3M-CarbOx. Explicit state variables of the model are**
**indicated in continuous-line box or circles except the implicit variable for zooplankton (dotted line box). Orange-written**
**state variables are added variables compared to the preexisting biogeochemical model of Fraysse et al. (2013). Arrows**
**represent processes between two state variables. TA: Total Alkalinity. DIC: Dissolved Inorganic Carbon, CO₂: carbon**
**dioxide, O₂: Oxygen, CaCO₃: calcium carbonate.**

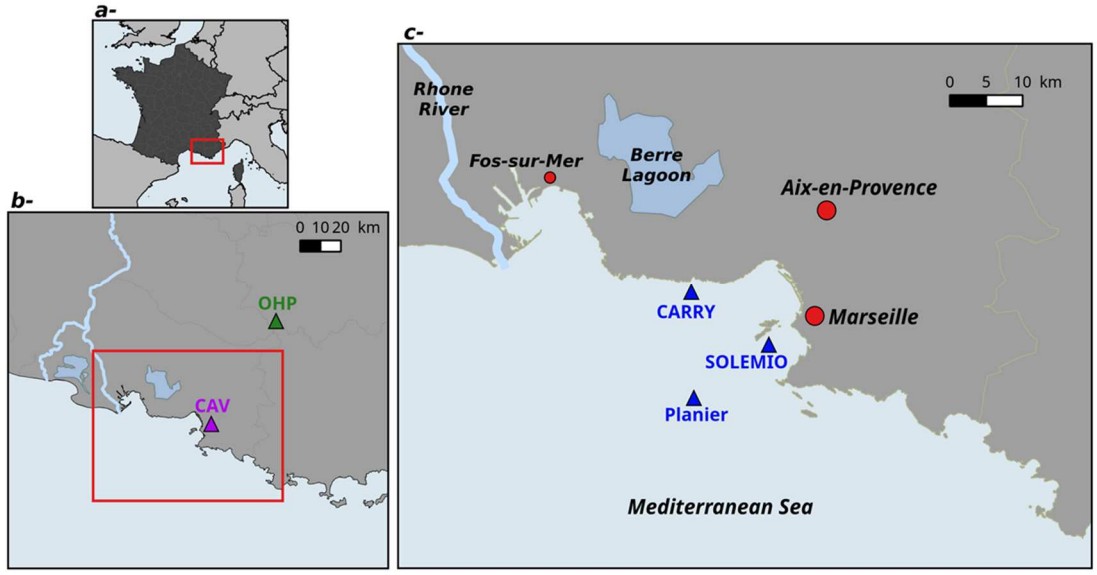

**Figure 2: Map of study area: The Region Sud (A), Aix-Marseille Metropolis (B), the Bay of Marseille (C). CAV= Cinq**
**AVenues Station (urban site), OHP: Observatoire de Haute Provence station (non-urban site), Carry, Solemio, Planier:**
**Marine study sites in the Bay of Marseille.**

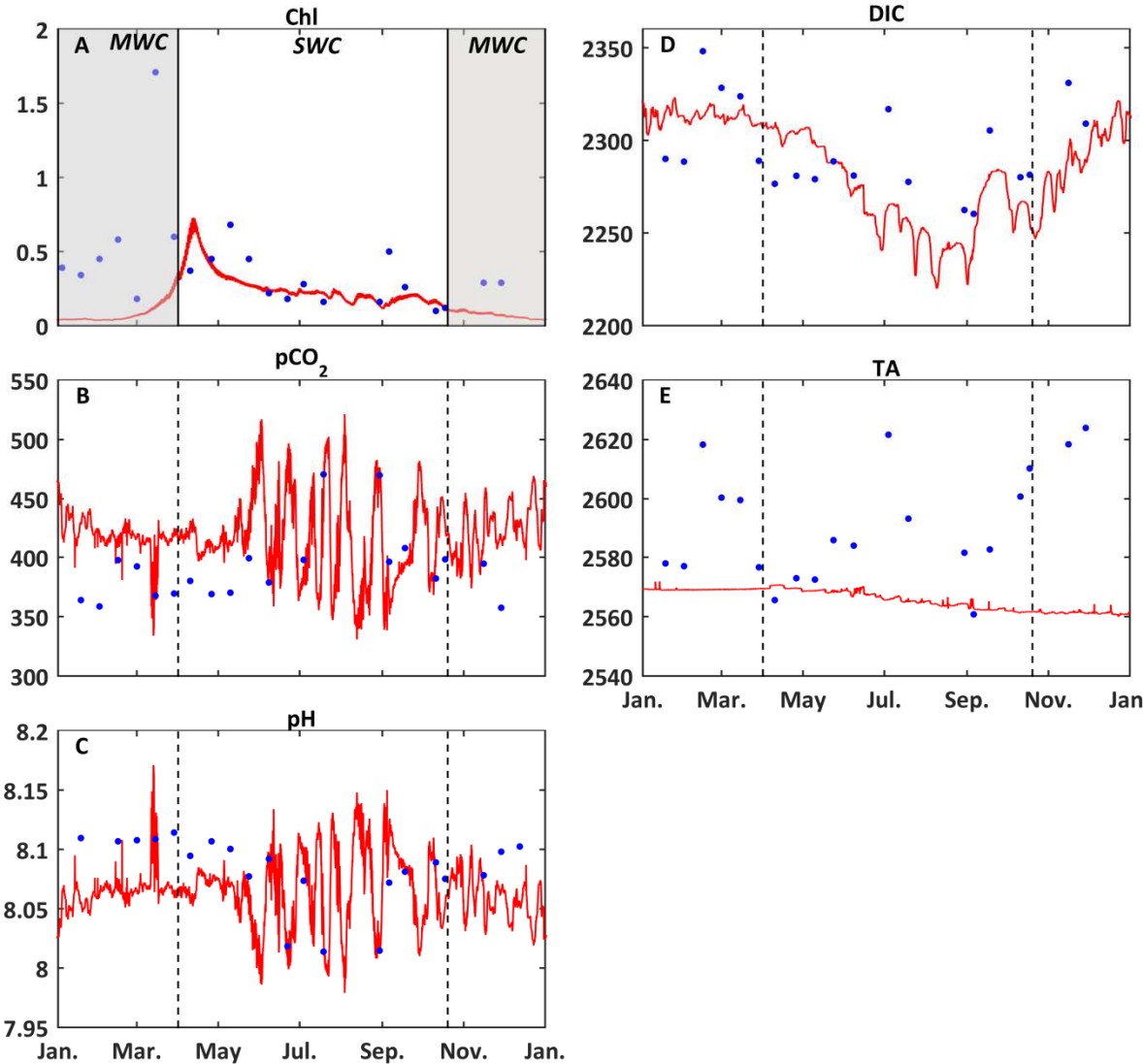

**Figure 3: Comparison of model results (red) and *in situ* data (blue) at the surface of the SOLEMIO station. (A) Chlorophyll-**
***a* concentrations (mg m⁻³), (B) *p*CO₂ (μatm), (C) *p*H, (D) DIC (μmol kg⁻¹), (E) TA (μmol kg⁻¹). The value of each state variable**
**represents the mean around ±5 days of the sampling date. The shaded area and dotted black line delimit the SWC and**
**MWC periods.**

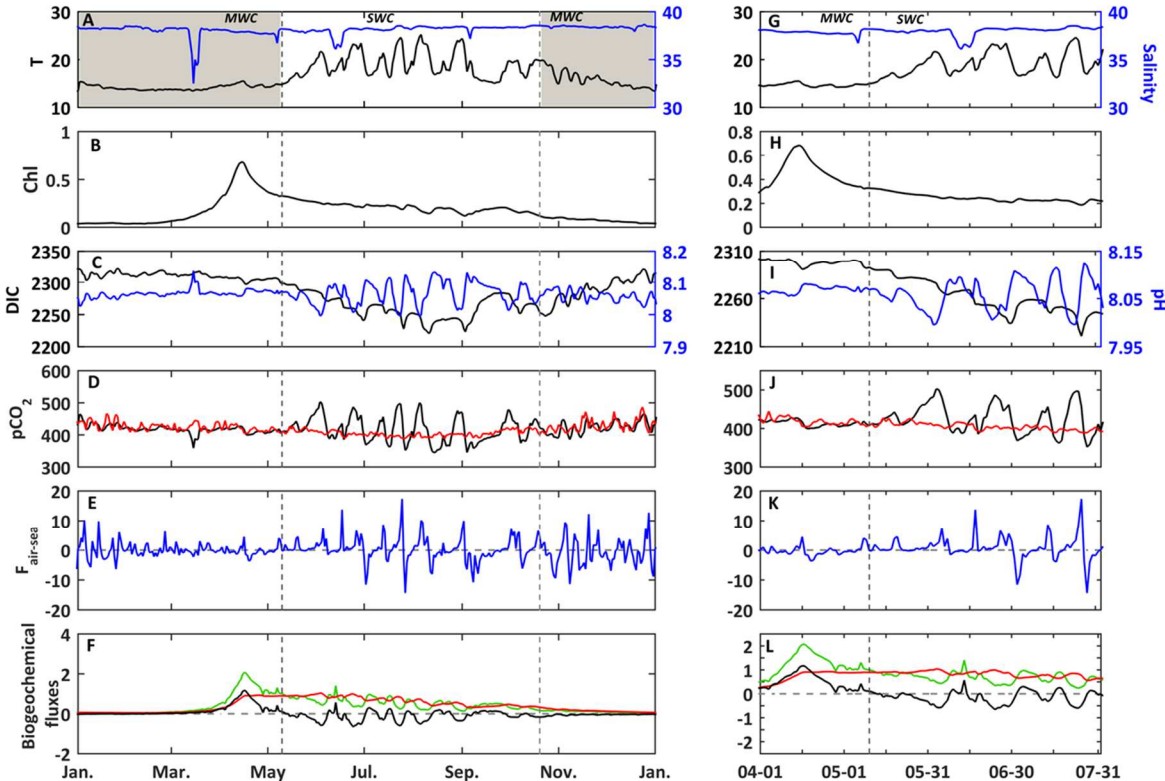

Figure 4: In the left panels: year 2017, right panels: temporal focus between April 1st and July 31th, 2017. *In situ* daily average of (A, G) temperature (°C, black line) and salinity (blue line) at the SOLEMIO station (at the surface). Modeled daily average (B, H) chlorophyll *a* concentrations (mg m-3, black line) (C, I) DIC (µmol kg-1, black line) and *p*H (blue line), (D, J) seawater *p*CO2 (µatm, black line) and atmosphere *p*CO2 from OHP (µatm, red line), (E, K) air-sea CO2 fluxes mmol m-3 d-1), (F, L) Gross Primary Production (mmol m-3 d-1, green line), total respiration (mmol m-3 d-1, red line) and Net Ecosystem Production (mmol m-3 d-1, black line). The shaded areas and dotted black lines delimit the SWC and MWC periods.

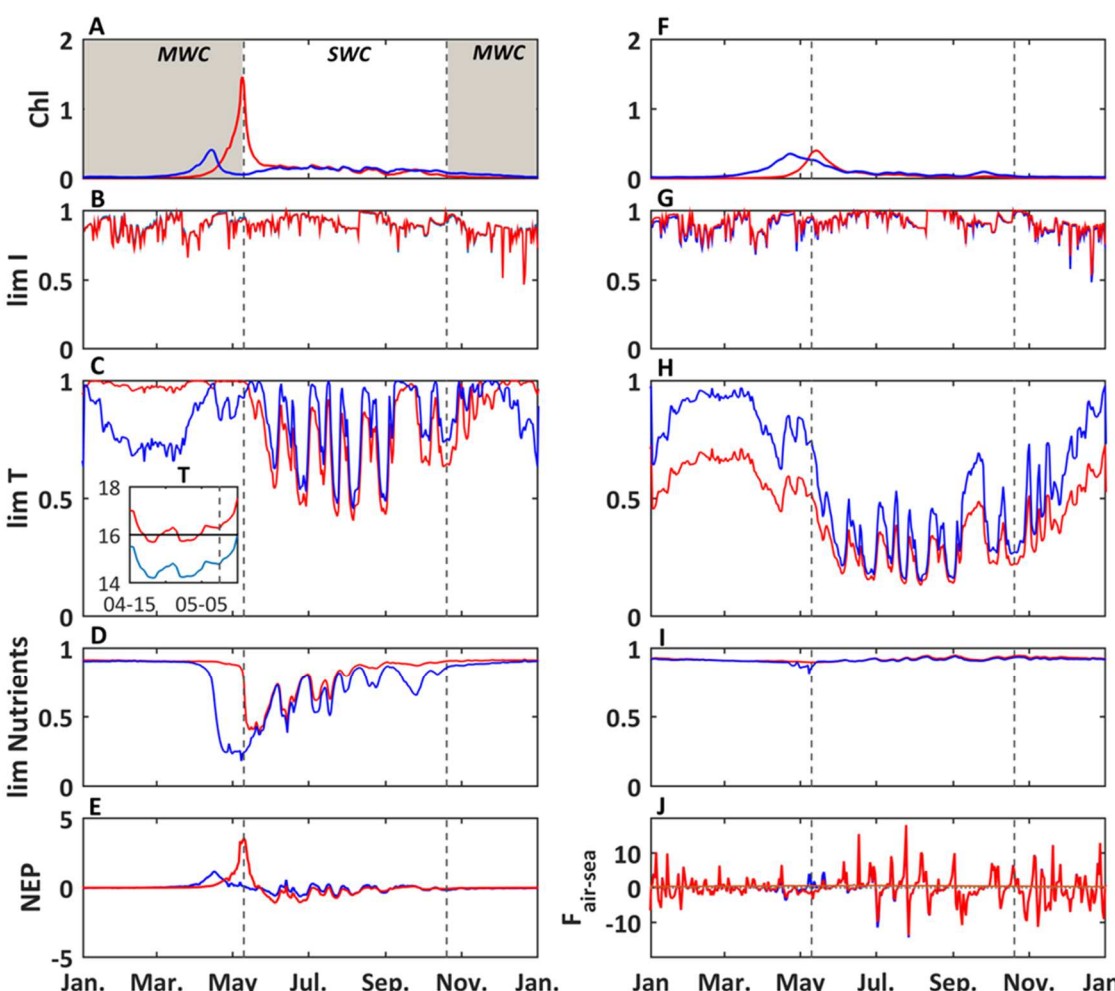

**Figure 5: Modeled daily average chlorophyll *a* concentrations (mg m$^{-3}$) (A), light limitation (B), temperature limitation, and a zoom from April 15$^{th}$ to May 5$^{th}$ of temperature (C) and nutrient limitation (D) for picophytoplankton and the same set for diatoms (F, G, H and I). Modeled daily average NEP (mmol m$^{-3}$ d$^{-1}$, E) and air-sea CO$_2$ flux (mmol m$^{-3}$ d$^{-1}$, J). Reference simulation (S0, blue line) and temperature-shifted simulation by 1.5°C (S2, red line). The shaded area and dotted black lines delimit the SWC and MWC periods.**

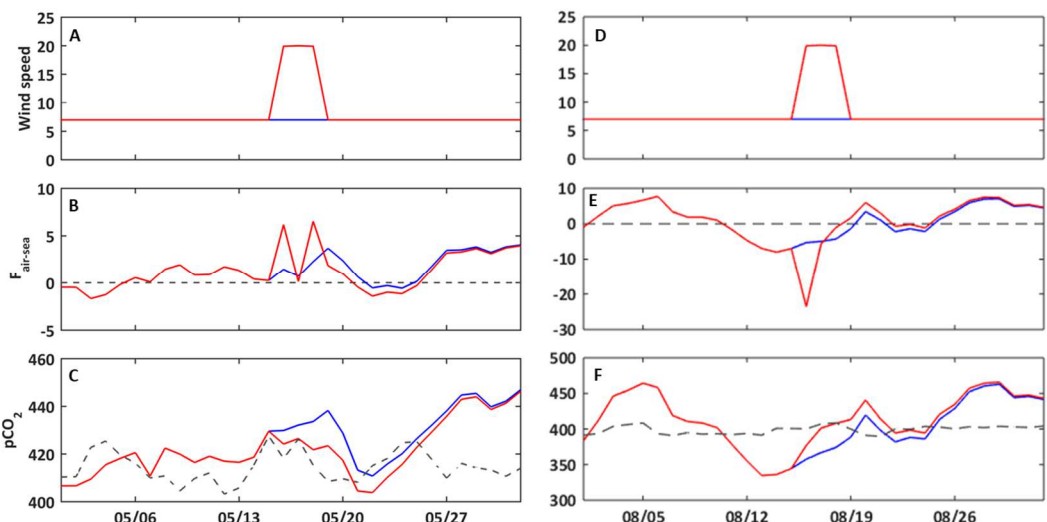

**Figure 6: Temporal evolution for May (left panels) and August (right panels) 2017 of the wind speed (m s$^{-1}$, A, D); air-sea CO$_2$ fluxes (mmol m$^{-3}$ d$^{-1}$, B, E); seawater partial pressure of CO$_2$ (μatm, C, F). Constant wind scenario (S2, blue line) and wind event scenario (S3, red line). On panels C and F, the dashed line represents the atmosphere partial pressure of CO$_2$ (μatm) at the CAV station.**

885

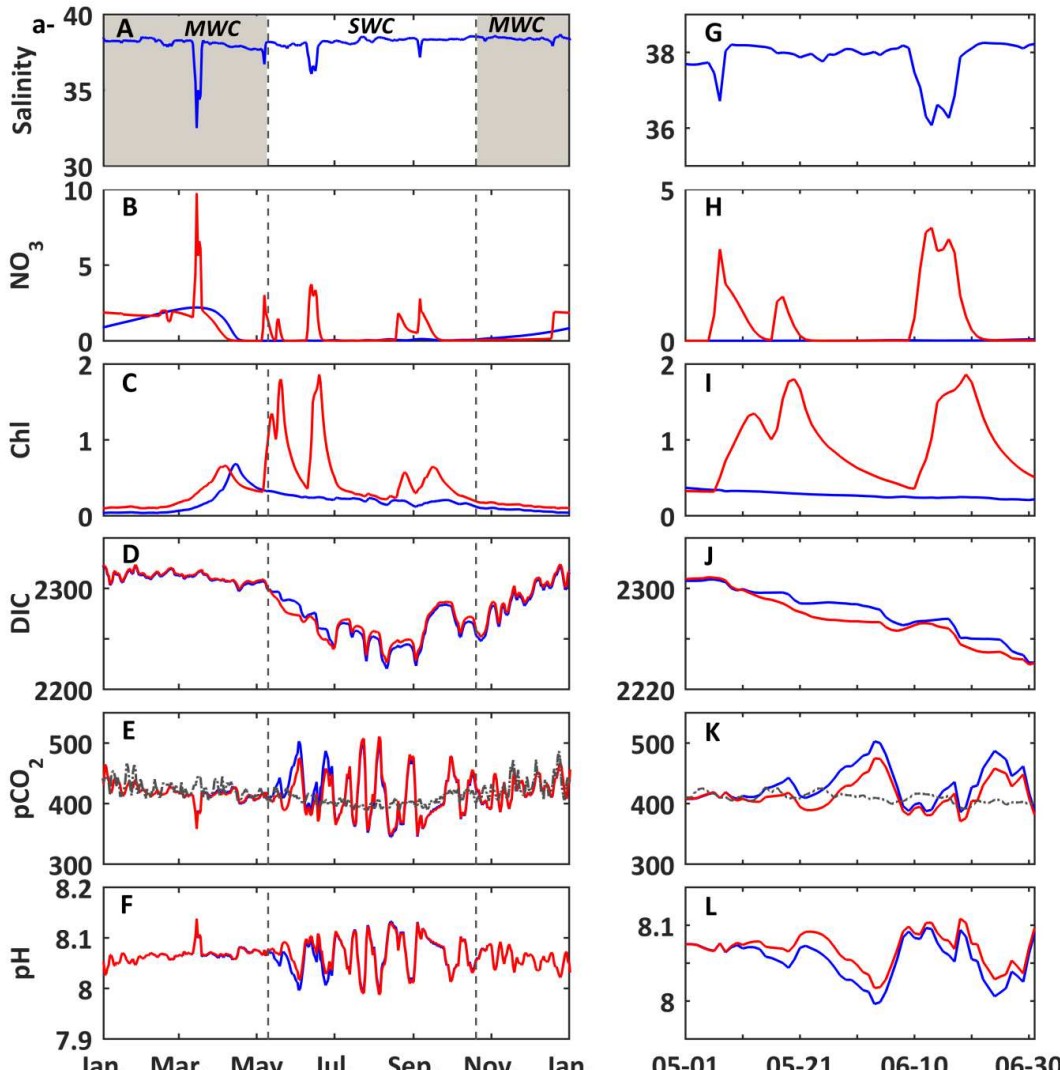

**Figure 7: In the left panels: year 2017 and right panels: temporal focus between May 1st and July 1st, 2017. (A, G)** *In situ* **daily average of salinity. Modeled daily average (B, H) nitrate concentrations (mmol m⁻³); (C, I) chlorophyll** *a* **concentrations (mg m⁻³); (D, J) DIC (µmol kg⁻¹); (E, K) seawater** *p*CO₂ **(µatm), and (F, L)** *p*H. **Reference simulation (S0, blue line) and nitrate supply simulation (S4, red line). On panels E and K, the dashed line represents the atmosphere partial pressure of CO₂ (µatm) at the CAV station. The shaded area and dotted black lines delimit the SWC and MWC periods.**

890

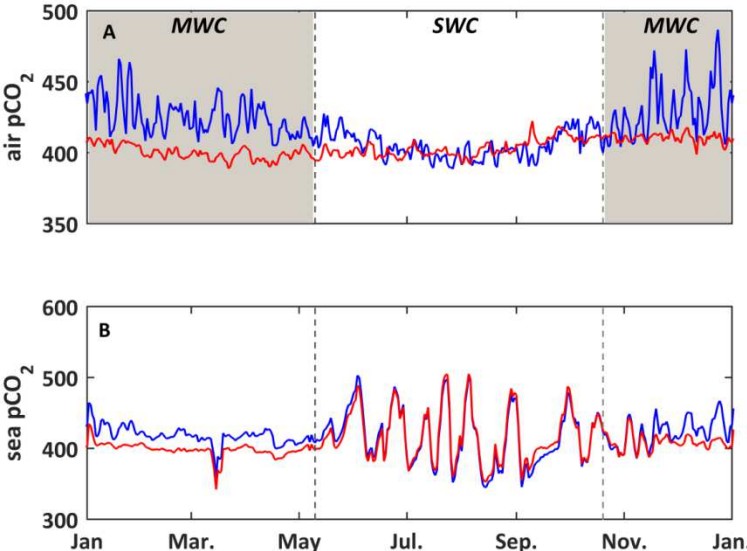

**Figure 8:** **(A) Temporal evolution for the year 2017 of the observed partial pressure of $CO_2$ (µatm) in the atmosphere at the CAV station, called the "urban scenario" (S0, blue line), and at the OHP station, called the "non-urban scenario", and in seawater (S6, red line). (B) Temporal evolution for the year 2017 of the modeled seawater partial pressure of $CO_2$ (µatm) with forcings from the urban (S0, blue line) and non-urban (S6, red line) scenarios. The shaded area and dotted black lines delimit the SWC and MWC periods.**