# Peer review of "Implementation and assessment of a carbonate system model (Eco3M-CarbOx v1.1) in a highly-dynamic Mediterranean coastal site (Bay of Marseille, France)."

_Geoscientific Model Development, 2020_

## Referee Comment (RC1) · Sebastiaan van de Velde (Referee) · 14 Jun 2020

Lajaunie-Salla and coworkers present an extension of an existing food-web model with a carbonate chemistry balance. They subsequently use this model to look at the carbonate dynamics in the Bay of Marseille, and use sensitivity test to find under what circumstances the the coastal waters of the BoM could be a source or sink for CO2. Overall I found this an interesting study, even though its focus is very local. Depending on the flexibility, this model could likely also be used on different coastal sites. I do have a few issues with the manuscript in its current state that I feel need to be resolved

before it can be published. First and foremost, I had troubles to understand the model set-up. Given that this is a model development journal, the model should be clearly articulated in the main text, and this is not the case. Secondly, I am not convinced by the authors discussion of the disagreement between model and data. Because they are using the model to look at dynamics in carbonate chemistry, they should either be able to reproduce the data in a better way, or at the very least discuss in more detail why there is a disagreement, and why that is not a problem.

I do however think these issues can be dealt with in a revised version, and find the study itself valuable.

Comments:

1. Model Description: I do not think the model development or set-up has been well described in the text. All equations and parameters are to be found in a number of tables of the appendix, and the readers are expected to either know the plankton model used, or go to other papers to find it. This might be fine if it was an established model and in a different journal, but I do not think it is good for GMD. The reasoning behind model set-up and parametrisation is not explaining in the text, so it is difficult to understand why the model was set-up as it was. After reading the methods I still had a number of basic questions; (i) what are the dimensions of the model (1 box, 3D, . . .)? (ii) It is stated at L153 that the variables were initialized at winter conditions and forced with measured temperature etc. Does it not require a spin-up for the circulation – that is, presuming it has circulation? (iii) Why do you choose two three-day periods of wind speed? Why 7 m s-1? What are the boundary conditions? How does the BoM connect to the rest of the Mediterranean? . . .

Furthermore, there seem to be a number of inconsistencies between tables and between tables and text, for example; (i) In Table A3 you use 'POM' for the bacterial grazing term, but that does not show up in your state variable list (it is likely detritus, but then it should be called that, otherwise it causes confusion), (ii) (O/N)nit in Table

**[GMDD](https://example.org)**
A2 -> (O/C)nit in Table A5, (O/N)uptNO3 in Table A2 -> (O/C)uptNO3 in Table A5 . . .

It is stated at L138 there is closure term, but I did not find where the grazing term closes the balance?

2. Model-data agreement: You discuss the model-data comparison in section 3.1, but seem to brush over some of the misfits quite easily. For example, the alkalinity in Fig.3 – you say the model results are within the range of the data, but they are only barely within the range, and most of the data plots above the model values. At L208 you say that the model shows the same trends for pH and pCO2, but you have a consistent offset in the first half of your pCO2 graph? And the trends seem to be inverse in the first half (blue dots going up and orange down) and last part (orange up and blue down) of the graph. Same for the pH (which is what you would expect as they are coupled) I would think that if you want to use the model to investigate carbonate dynamics under future climate change scenarios, you would want to be able to reproduce (or explain) what happens with the alkalinity and pH? Those model-data comparison does not give much confidence to the model (or parametrization) if you specifically want to address carbonate related questions.

3. General readability: In general, the text reads a bit awkward, and seems to have a strong French influence (I mean no offense, but that is just the way it felt when I was reading it). In particular oddly placed articles, and plural forms where it should be singular. The manuscript could probably benefit from proofreading by a native speaker. Then again, I might be wrong as I am also not a native speaker, and this is merely a suggestion.

Minor comments:

L29: 'strong atmospheric CO2' -> do you mean high concentrations?

L41: are you considering the biological pump to be a physical process?

L47: 'organic matters' -> organic matter

L143: just 'dataset' suffices

L149: Why did you not plot the temporal trend versus the datapoints?

L236: Why is the flux not expressed in a mol per unit area value?

L248: There is not really a decrease of seawater pCO2, it just becomes much more variable

L282: 'farer' -> further ?

L304: how can it affect the Spring bloom before the nutrients are supplied?

L359: It sounds contradictory to say the 1.5°C rise affects the carbonate system little, but at L350 that the system is mostly driven by temperature variations

L363: double set of citations

Figures: give the legend in the figure panels instead of in the caption

Fig. 6: what are those weird spikes in the air-sea gas exchange curve?

---

## Referee Comment (RC2) · Guy Munhoven (Referee) · 27 Jul 2020

**1 General comments**

**1.1 Appreciation of the manuscript**

In this paper, K. Lajaunie-Salla and co-authors present Eco3M-CarbOx, a biogeochemical model of the carbonate system and the plankton food web. It is integrated into

the *Ecological, Mechanistic and Modular Modelling* framework, Eco3M (Baklouti et al., 2006b). In the paper, it is applied to the Bay of Marseille (Mediterranean Sea) for which there are comprehensive data time series available.

I found the study interesting, but the paper has, unfortunately, a number of weaknesses. The paper is well-readable, but there remain numerous English errors.

This manuscript has been submitted as a "model description paper." The model description in the main text reduces to thirty-two lines of text only (plus a schematic and a few equations in the appendix), of which fifteen deal with the new carbonate system module. These fifteen lines contain only a few commonplace statements followed by a sequence of eleven reference citations. This is far from what I expect to read in a "model description paper" in *Geoscientific Model Development*. It is also far from what is expected for that type of publication (see http://www.geoscientific-model-development.net/about/manuscript_types.html#item1). Accordingly, I do not even think that this manuscript fits the scope of the journal in its current form. Details about the approximations, numerical methods adopted and algorithms used, their applicability and their limitations are completely missing.

Furthermore, some of the model experiments also have critical shortcomings: it is not realistic to assume that river water intrusions only impact the Total Alkalinity, TA, budget but not that of Dissolved Inorganic Carbon, DIC (riverine TA is mainly carried as $HCO_3^-$ which impacts the DIC and the TA budgets alike).

For upwelling events, only a temperature effect is considered. However, upwelling events also bring nutrients, DIC and TA to the surface. The effects of these latter are not considered in the paper, possible effects not even discussed.

I think the authors will first have to make up their mind whether they want to consider their manuscript as a "model description paper" in *Geoscientific Model Development*,
or whether they would prefer to focus on the data analysis and interpretation. In both cases, they will have to extend the model description and revise the experimental design; in the second case, it would be recommended to submit this paper in other journal, such as, e.g., the sister journal *Biogeosciences*.

I am nevertheless convinced that this paper could make a valuable contribution to *Geoscientific Model Development*: from what I have been able to grasp from the paper and the code, the model approach looks solid and it could certainly be applied to other regions of the World as well. I therefore encourage the authors to go for the first option and prepare a **major revision** of their manuscript that includes a comprehensive model description and a sound experimental design.

**2   Specific comments**

**2.1   Abstract**

The abstract is not well focused and the hesitation between a model description and a data analysis approach is strongly visible.

**2.2   Model description**

The model description is completely insufficient. For the underlying model, upon which Eco3M-CarbOx is built, only the ecological structure is summarized. Nothing is said about the spatial extension adopted: is it a point model? does it have some spatial extensions? 1D, 2D, 3D? In the code, one can see that state variable arrays are three-dimensional, but it is not clear if the three spatial dimensions are actually used: the applied pressure, e.g., is set to 1 bar throughout, as if the model was applied for a

water depth of about 10 m only. If the model has some spatial extension, how are the lateral and bottom boundaries treated? The physical processes, although mentioned from time to time, are not at all dealt with here. How are they (e.g., transport processes) represented? This lack of description is rather incomprehensible as the authors themselves emphasize in their description of the study area that the biogeochemistry in the Bay of Marseille is "highly driven by hydrodynamics" (p. 3, l. 99).

Any carbonate system speciation calculation procedure rests upon a TA approximation and a pH solver. Here, we do not read anything about these two elements:

- What TA approximation is adopted, i.e., which acid-base systems are taken into account?

- What pH-scale is adopted?

- Which numerical method is used to solve the resulting pH equation? What are the limitations of the adopted method (some methods fail to converge for low salinity water samples, e.g.)?

- Which parametrisations have been used for the stoichiometric constants?

I have been able to find answers to some of these questions by browsing though the code (although I am still not sure which pH scale is actually used in the end—probably $pH_{tot}$). These informations must nevertheless be given in the main paper. It should not be necessary to inspect the code to find such basic informations.

As the model description stands, there is no way to reproduce the model results, a main requirement of model descriptions in *Geoscientific Model Development*.

**2.3  Experimental design**

As mentioned above, I find that there are inconsistencies in the design of the model experiments.

In the model upwelling events, only a temperature effect is taken into account. However, as stated on p. 3, l. 100, such events also bring nutrient rich waters to the surface. Accordingly, they should also perturb the nitrate, DIC and TA balances. This is not what the model results reflect: they witness of cooling events only.

Similarly, only part of the effects of Rhône river plume intrusions on the carbonate system are taken into account: the experiments only consider the resulting TA perturbation, but not the DIC perturbation. To my best knowledge, rivers mainly carry TA in the form of $HCO_3^-$ which impacts the DIC balance as strongly as the TA budget. I am even wondering—but could not find any decent data—if the River Rhône water does not also have high $p\mathrm{CO}_2$, in which case it would even carry more DIC than TA.

**2.4  Code**

The code is provided on Zenodo and is easy to download. No reference to this manuscript is given on the model's entry page on Zenodo though. I have not tried to compile the code but only browsed through it as I was interested in getting at least some basic information about the new carbonate system model announced in the title.

The code is commented, but most of the comments are unfortunately in French. This is especially annoying for the `Makefile` and the initialisation file `BIO/config.ini` which cannot be understood without a good proficiency in French.

No user manual is provided, neither on the Zenodo page nor as a Supplement to the paper.

**3   Technical comments**

Throughout the paper: please always specify which pH scale is used for reporting the data and model results (in tables, on graphs, etc.)

p. 1, l. 5: is co-author "Irène Remy-Xueref" not actually "Irène Xueref-Remy"? (in the reference section at least the name is spelled that way and that is also the name registered in the submission system)

p. 1, l. 18: "22 states variables" should read "22 state variables"

p. 1, l. 33: "2018, May to 2019, May" should read "May 2018 to May 2019"

p. 2, ll. 42–43: this is unclear. I would not range the biological pump among the physical ones. But it is not sure what is meant here by "physical" pumps.

p. 2, l. 54: "dynamic" should read "dynamics"

p. 2, l. 54: Why only "amplify"? A priori, the forcings could just as well attenuate or reduce acidification.

p. 2, l. 56: "At a global scale" should read "At the global scale"

p. 2, l. 58: "as a net sink or source" may be more appropriate.

p. 2, l. 63: 'MacKenzie" should read "Mackenzie" (also misspelled in the bibliography)

p. 2, ll. 75–76: "strong winds events" should read "strong wind events"

p. 3, l. 86: "implemented within" should read "implemented into"

p. 3, l. 95: Please delete "inhabitants" ("a population of ca. 1 million" is sufficient).

p. 3, l. 97: "winds conditions" should read "wind conditions"

p. 3, l. 101: "intrudes in the BoM" should read "intrudes the BoM" or "intrudes into the BoM"

p. 3, ll. 103–104: "diverse anthropogenic forcing" should read "diverse anthropogenic forcings"

p. 3, l. 106: delete "city"

p. 3, l. 113: "modeling platform" should read "platform" (there is no need to repeat the "modeling")

p. 3, l. 113: The paper cited (Baklouti et al., 2006a) is not adequate, as far as I can see. The companion paper Baklouti et al. (2006b), which describes the platform would be more appropriate. Baklouti et al. (2006a) review mechanistic formulations for key processes that control phytoplankton dynamics and present a generic model, less so the platform. Please check this.

p. 4, ll. 129–134: The TA production and consumption rates are stated in a very imprecise way here. As such these statement do not make much sense. It should be

specified what are the references for the stated TA changes (e.g. TA decreases by two moles *for each mole of* CaCO$_3$ precipitated, and by $x$ moles *for each mole of* XY assimilated by phytoplankton, etc.)

p. 4, l. 132: "when bacteria mineralized" should read "when bacteria mineralize"

p. 4, l. 147: please delete "However" which does not make sense here.

p. 4, l. 148: "model results" can be compared with the observations, not the model itself.

p. 5, l. 161: "winds specific conditions" should read "wind-specific conditions"

p. 6, l. 206: "contains a low value of WSS" – not sure what this could possibly mean. "a low value of WSS" should anyway read "a low WSS" as the last 'S' stands for 'score,' which is a value.

p. 6, l. 214: "calculates a WSS value of 0.69" better had to read "yields a WSS of 0.69"

p. 6, l. 223: "seasonal dynamic" should read "seasonal dynamics"

p. 7, ll. 257–262: I am quite surprised about this. I would expect that upwelling events not only bring up cold water, but also nutrient, DIC and TA rich waters. Unfortunately, the model description does not explain how the upwelling events are represented. Could you please elaborate on this.

p. 8, l. 281: "diatoms" should read " diatoms' " (genitive)

p. 8, l. 295: "in-gassing" should read "absorption" or "uptake"

p. 8, l. 295: "variability" at which time scales?

p. 9, l. 329: "weaker" should read "lower"

p. 9, l. 354: I think that "counteracting" is more appropriate than "counterbalanced" at this point

p. 12, l. 424: "Environnenemnts" should read "Environnements"

p. 12, l. 432: "takes part" should read "is part"

p. 12, l. 434: "Agence" should read "Agency"

p. 12, l. 434: "from European" should read "from the European"

p. 12, l. 434: "used in this paper" should read "presented in this paper"

**References**

Baklouti, M., Diaz, F., Pinazo, C., Faure, V., and Quéguiner, B.: Investigation of mechanistic formulations depicting phytoplankton dynamics for models of marine pelagic ecosystems and description of a new model, Progress in Oceanography, 71, 1–33, https://doi.org/10.1016/j.pocean.2006.05.002, 2006a.

Baklouti, M., Faure, V., Pawlowski, L., and Sciandra, A.: Investigation and sensitivity analysis of a mechanistic phytoplankton model implemented in a new modular numerical tool (Eco3M) dedicated to biogeochemical modelling, Progress in Oceanography, 71, 34–58, https://doi.org/10.1016/j.pocean.2006.05.003, 2006b.
* * *

---

## Author Comment (AC1) · 27 Jul 2020

Sebastiaan van de Velde (Referee)
sebastiv@ucr.edu

Lajaunie-Salla and coworkers present an extension of an existing food-web model with a carbonate chemistry balance. They subsequently use this model to look at the carbonate dynamics in the Bay of Marseille, and use sensitivity test to find under what circumstances the coastal waters of the BoM could be a source or sink for $CO_2$. Overall I found this an interesting study, even though its focus is very local. Depending on the flexibility, this model could likely also be used on different coastal sites.

I do have a few issues with the manuscript in its current state that I feel need to be resolved before it can be published. First and foremost, I had troubles to understand the model set-up. Given that this is a model development journal, the model should be clearly articulated in the main text, and this is not the case.

Secondly, I am not convinced by the authors discussion of the disagreement between model and data. Because they are using the model to look at dynamics in carbonate chemistry, they should either be able to reproduce the data in a better way, or at the very least discuss in more detail why there is a disagreement, and why that is not a problem. I do however think these issues can be dealt with in a revised version, and find the study itself valuable.

Reply: We thank the reviewer for the positive evaluation of our work to be published in GMD journal. We thank too him for the detailed and useful comments that contributed to greatly improve the manuscript.

This study is focused on the Bay of Marseille (BoM) that harbors the big metropolis of the aforementioned town hosting more than 800 000 inhabitants, with in summer an increasing of tourism activities. Moreover, the BoM is impacted by many harbor activities. In this context, we think so that the main results of our study could be transposed to other coastal sites that are also impacted by urban and anthropic pressures.

We modify the conclusion section of manuscript L429-441 as following:

"*The BoM biogeochemical functioning is mainly forced by wind-driven hydrodynamics (upwelling and downwelling), urban rivers, wastewater treatment plants, and atmospheric deposition (Fraysse et al., 2013). In addition, Northern Current and Rhone River plume intrusions frequently occurred (Fraysse et al., 2014; Ross et al., 2016). **Moreover the Bay of Marseille harbors the second bigger metropolis of France (Marseille) that is impacted by many harbor activities. The next step of this study will be to couple the Eco3M-CarbOx biogeochemical model with a 3D hydrodynamic model that** will mirror the complexity of the BoM functioning. In this way, the contributions of hydrodynamic, atmospheric, anthropic, and biogeochemical processes to the DIC variability could be determined, and an overview of the air-sea $CO_2$ exchange could be made at the scale of the Bay of Marseille. **The main results of our study could be transposed to other coastal sites that are also impacted by urban and anthropic pressures.** Moreover, in this paper we highlighted that fast and strong variations of $pCO_2$ values occur, so thus it is essential to acquire more in situ values at high frequency (at least with an hourly resolution) to understand the rapid variations of the marine carbon system at these short spatial and temporal scales.*

As mentioned by the reviewer we give more details about the model set-up and the modifications made from the previous version. Moreover, we propose to change the figure 3 of the previous manuscript by the figure that compares model and *in situ* values without any mean. This figure shows better the good reproduction of the in situ data by the model.

**Comments:**

1. Model Description:

I do not think the model development or set-up has been well described in the text. All equations and parameters are to be found in a number of tables of the appendix, and the readers are expected to either know the plankton model used, or go to other papers to find it. This might be fine if it was an established model and in a different journal, but I do not think it is good for GMD. The reasoning behind model set-up and parameterization is not explaining in the text, so it is difficult to understand why the model was set-up as it was. After reading the methods I still had a number of basic questions;

Reply: The Eco3m-CarbOx model is based on a pre-existing model of the plankton ecosystem developed by Fraysse et al. (2013). The model presented in our study includes a set of new developments and improvements in the realism of the plankton web structure and process formulations. In order to improve the representation of chlorophyll concentration in the Bay of Marseille, two types of phytoplankton were added: the *Synechococcus* cyanobacteria, which is one of the major constitutive members of picophytoplankton in Mediterranean Sea (Mella-Flores et al., 2011), and the large diatoms, which are generally observed during spring blooms at mid-latitudes (Margalef, 1978). The functional response of primary production was modified using another formulation of temperature limitation function which takes into account the optimal temperature of growth for each phytoplankton (diatoms and picoplankton). The exudation of phytoplankton was modified taking into account the intracellular phytoplankton ratio of C, N and P. For the assimilation of matter (inorganic and organic) by bacteria and the remineralization processes the dependence on intracellular bacteria ratio was added. A temperature dependence of all biogeochemical processes was added. Also certain parameters in some formulations were modified. The Eco3m-CarbOx model also hosts now a carbonate system module. For this three processes were added (i) the precipitation, (ii) dissolution of carbonate and (iii) the gas exchange with the atmosphere. We agree with the reviewer on this point and we added some important elements on the approach of alteration of the pre-existing model. However, we think that, for a sake of clarity, it is important to keep in the appendix section the tables including source-sink equations and the long lists of model parameters.

We modify a part of the "Numerical model description" section L121-143 as following:
*"The model presented in this study includes a set of new developments and improvements in the realism of the plankton web structure and process formulations. In order to improve the representation of chlorophyll concentration in the Bay of Marseille The phytoplankton is divided in two groups: one with traits of the Synechococcus cyanobacteria, which is one of the major constitutive members of pico-autotrophs in Mediterranean Sea (Mella-Flores et al., 2011), and another with traits of large diatoms, which are generally observed during spring blooms at mid-latitudes (Margalef, 1978). For both of the phytoplankton, there is a diagnostic chlorophyll-a variable related to the phytoplankton biomass in carbon, the phytoplankton N-to-C ratio, and the limiting internal ratio $f_Q^N$ (Faure et al., 2010; Smith and Tett, 2000; Tab. A2). The functional response of primary production was modified using another formulation of temperature limitation function which takes into account the optimal temperature of growth for each phytoplankton. The exudation of phytoplankton was*

*modified taking into account the intracellular phytoplankton ratio. For the uptake of matter by bacteria and remineralization processes the dependence on intracellular bacteria ratio was added. A temperature dependence of all biogeochemical processes was added to take into account the effects of rapid and strong variations of seawater temperatures on plankton during episodes of upwelling for instance that are usually often observed in the BoM. Also certain parameters in some formulations were modified owing to the alterations of some formulations.*

*Additionally**, a carbonate system module was developed and** three state variables have been added: dissolved inorganic carbon (DIC), total alkalinity (TA) and the calcium carbonate (CaCO₃) implicitly representing calcifying organisms. The knowledge of DIC and TA allows the calculation of the pCO₂ and pH diagnostic variables, necessary for resolving all the equations of the carbonate system. These equations use apparent equilibrium constants, which depend on temperature, pressure, and salinity (Dickson, 1990a, 1990b; Dickson and Riley, 1979; Lueker et al., 2000; Millero, 1995; Morris and Riley, 1966; Mucci, 1983; Riley, 1965; Riley and Tongudai, 1967; Uppström, 1974; Weiss, 1974). **For this module three processes were also added: the precipitation and dissolution of calcium carbonate and the gas exchange of pCO₂ with the atmosphere.*"

(i) what are the dimensions of the model (1 box, 3D, : : :)?

Reply: The spatial dimension of the model is 0 in this study. The state variables of the model only change along time. We add this information in L113-115, as following:
"*In this study, the state variables of the Eco3M-CarbOx model only change along time (i.e. usually termed "model 0D"), are representative of the time evolution of a sea surface water cell but this biogeochemical plankton model is not coupled with a hydrodynamic model.*"

(ii) It is stated at L153 that the variables were initialized at winter conditions and forced with measured temperature etc. Does it not require a spin-up for the circulation – that is, presuming it has circulation?

Reply: We agree that this information was not clear in our previous version of the MS. As in this work the biogeochemical model is not coupled with a hydrodynamic model, the circulation is not taking into account. We add this information in L113-115, as following:
"*In this study, the state variables of the Eco3M-CarbOx model only change along time (i.e. usually termed "model 0D"), are representative of the time evolution of a sea surface water cell but this biogeochemical plankton model is not coupled with a hydrodynamic model.*"

(iii) Why do you choose two three-day periods of wind speed? Why 7 m s$^{-1}$? What are the boundary conditions? How does the BoM connect to the rest of the Mediterranean?

Reply: As we can see in the figure below, which represents the time series of wind speed at the SOLEMIO station located in the BoM, the studied area is highly impacted by windy periods with speed values above 20 m s$^{-1}$. In 2017 the average wind speed was 7 m s$^{-1}$. In order to analyze the impact of these windy periods on the carbonate system we run the model with wind speed of 20 m s$^{-1}$ during three days. We choose three-day periods because the BoM is often impacted by short bursts of Mistral. This wind usually blows for a few days with high speeds ranging from 14 to 28 m s$^{-1}$.

[Figure]

*Figure A: Time series of wind velocity (m s⁻¹) at SOMLIT station between July 1st and August 1st, 2017 (gray line) and the average of wind velocity (red line).*

We modify the manuscript L182-185 as following:

*"Impact of wind events: a first simulation S2 was run with a constant wind intensity of 7 m s⁻¹ (2017 annual average wind speed) throughout the year and a second one (S3) with two three-day periods of strong wind speed (20 m s⁻¹) representative of short bursts of Mistral (data not shown) starting on May 15ᵗʰ and August 15ᵗʰ, and a constant value of 7 m s⁻¹ the rest of the year."*

Furthermore, there seems to be a number of inconsistencies between tables and between tables and text, for example;

(i) In Table A3 you use 'POM' for the bacterial grazing term, but that does not show up in your state variable list (it is likely detritus, but then it should be called that, otherwise it causes confusion).

Reply: We agreed with the reviewers to modify in the manuscript the term "detritus" by "DPOM" to define the detrital particulate organic matter. In same way we modify the term dissolved organic matter "DOM", by the labile and semi labile dissolved organic carbon "LDOM". In the manuscript these changes were made in table A1-A2 and A3 and in L19 and– L118.

(ii) (O/N)nit in Table A2 -> (O/C)nit in Table A5, (O/N)uptNO3 in Table A2 -> (O/C)uptNO3 in Table A5 :

Reply: We agreed with the reviewer that there are several mistakes on the tables A2 and A5. In this study, the ratio $\frac{O}{C}$ and $\frac{O}{N}$ are equal to 1 and 2, respectively. There is not need to differentiate the name of this ratio for each process. We choose then to use only the ratio $\frac{O}{C}$ and $\frac{O}{N}$ without sub-index, we then modify the tables A2 and A5 considering these changes:

$$\left(\frac{O}{C}\right) = \left(\frac{O}{C}\right)_{PP} = \left(\frac{O}{C}\right)_{resp} = \left(\frac{O}{C}\right)_{respZ} = \left(\frac{O}{C}\right)_{respBa} = 1$$

$$\left(\frac{O}{N}\right) = \left(\frac{O}{N}\right)_{uptNO_3} = \left(\frac{O}{N}\right)_{nit} = 2$$

It is stated at L138 there is closure term, but I did not find where the grazing term closes the balance?

Reply: In the model, all the matter grazed by the zooplankton, (i.e. bacteria, phytoplankton, and DPOM) return as either organic or inorganic matter by excretion, egestion and mortality processes (Fig. B).
- The excretion of zooplankton returns dissolved inorganic matter with a flux of :
$$\boldsymbol{R_{excr}^{DIM}} = \varepsilon_{DIM} \cdot d_X \cdot \left(1 - k_{X,zoo}\right) \cdot \left(R_{Gr}^{Phy} + R_{Gr}^{DPOM} + R_{Gr}^{Ba}\right)$$
- The excretion of zooplankton returns labile dissolved organic matter with a flux of :
$$\boldsymbol{R_{excr}^{LDOM}} = (1 - \varepsilon_{DIM}) \cdot d_X \cdot \left(1 - k_{X,zoo}\right) \cdot \left(R_{Gr}^{Phy} + R_{Gr}^{DPOM} + R_{Gr}^{Ba}\right)$$
- The egestion of zooplankton returns detrital particulate organic matter with a flux of :
$$\boldsymbol{R_{pf}} = (1 - d_X) \cdot \left(R_{Gr}^{Phy} + R_{Gr}^{DPOM} + R_{Gr}^{Ba}\right)$$
- The mortality of zooplankton returns detrital particulate organic matter with a flux of :
$$\boldsymbol{R_m} = d_X \cdot k_{X,zoo} \cdot \left(R_{Gr}^{Phy} + R_{Gr}^{DPOM} + R_{Gr}^{Ba}\right)$$

[Figure]

*Figure B: Repartition of matter grazed by zooplankton*

The sum of these four processes is equal to the flux of matter grazed by zooplankton:
$$\boldsymbol{R_{excr}^{DIM}} + \boldsymbol{R_{excr}^{LDOM}} + \boldsymbol{R_{pf}} + \boldsymbol{R_m} = R_{Gr}^{Phy} + R_{Gr}^{DPOM} + R_{Gr}^{Ba}$$
This is the way whom the biogeochemical model is closed in this study.

2. Model-data agreement:

You discuss the model-data comparison in section 3.1, but seem to brush over some of the misfits quite easily. For example, the alkalinity in Fig.3
- you say the model results are within the range of the data, but they are only barely within the range, and most of the data plots above the model values.

Reply: We agreed with the reviewers and modify the sentence L246-248 as following:
*"The biogeochemical model provides almost constant values around 2570 µmol kg$^{-1}$ all along the year, which is lower than in situ data."*

- At L208 you say that the model shows the same trends for pH and pCO$_2$, but you have a consistent offset in the first half of your pCO$_2$ graph? And the trends seem to be inverse in the first half (blue dots going up and orange down) and last part (orange up and blue down) of the graph. Same for the pH (which is what you would expect as they are coupled) I would think that if you want to use the model to investigate carbonate dynamics under future climate change scenarios, you would want to be able to reproduce (or explain) what happens with the alkalinity and pH? Those model-data comparison does not give much confidence to the model (or parametrization) if you specifically want to address carbonate related questions.

Reply: We agree with the reviewer that in the figure 3 we cannot see well the variability of pCO$_2$ and pH, because the plotted data from model is a mean value at ±5 days around the

sampling date. Looking the figure C below, that represents the simulated data and *in situ* data, we can see that the model reproduces the dynamic of seasonal variations of seawater $pCO_2$ and pH.

[Figure]

*Figure C: Comparison of model results (red) and in situ data (blue) at the SOLEMIO station. (a) pCO$_2$ (µatm) and (b) pH*

From January to February, the model reproduces the observed decrease of $pCO_2$ and from February to March the increase of $pCO_2$. In mid-April the observed drop of $pCO_2$ and increase of pH are also spotted in the model (Fig.C green box). During summer, the dynamics linked to the temperature variations are also well reproduced by the model. We propose to change the Figure 3 of the previous manuscript by the figure that compares model and *in situ* values without any mean.

We modify the manuscript L223-233 as following:
*"On the whole, the seasonal **variations of the seawater pCO$_2$ are correctly simulated by the biogeochemical model (Fig. 3B), even if the values are slightly overestimated during MWC period. From January to February, the model reproduces the observed decrease of pCO$_2$ and from February to March the increase of pCO$_2$. In mid-April, during the simulated spring bloom period, the observed drop of pCO$_2$ and increase of pH are also spotted in the model (Fig.3C & E).** The model especially succeeds in reproducing the observed increase in relation to high temperatures during the SWC period. The reduction of the CO$_2$ solubility due to thermal effects mostly explains the increase in pCO$_2$ during the SWC period. The strong standard deviation of modeled values during the SWC period can be explained by the rapid changes in temperature due to upwelling occurring at this time of the year. The range of modeled pCO$_2$ values (345 - 503 µatm) encompasses the range of observation values (358 - 471 µatm; Tab. 2). **The statistical analysis calculated** a mean bias of +23 µatm, and a WSS value of 0.69 (Tab. 2)."*

3. General readability:

In general, the text reads a bit awkward, and seems to have a strong French influence (I mean no offense, but that is just the way it felt when I was reading it). In particular oddly placed articles, and plural forms where it should be singular. The manuscript could probably benefit from proofreading by a native speaker. Then again, I might be wrong as I am also not a native speaker, and this is merely a suggestion.

Reply: Before submitting the new revised version, we will give it to a native speaker for proofreading and English improvements.

**Minor comments:**

L29: 'strong atmospheric CO2' -> do you mean high concentrations?

Reply: We agree with the reviewer and changed "strong" by "high" in L29.

L41: are you considering the biological pump to be a physical process?

Reply: In fact the sentence no was clear, we modified as following L41:
"In the ocean, the main processes regulating $CO_2$ exchanges between the atmosphere and sea are the solubility pump and the biological pump at different time scales"

L47: 'organic matters' -> organic matter

Reply: We corrected as suggested L47.

L143: just 'dataset' suffices

Reply: We corrected as suggested L158.

L149: Why did you not plot the temporal trend versus the datapoints?

Reply: As the *in situ* data are recorded every 15 days, and in order to do a statistical analysis and have the same number of data we used the mean value at ±5 days around the sampling date. In the figure D below, we plot the temporal trend versus the datapoints without any mean.

[Figure]

*Figure D: Comparison of model results (orange) and in situ data (blue) at the SOLEMIO station. (a) Chlorophyll-a concentrations (mg m$^{-3}$), (b) pCO$_2$ (µatm), (c) pH, (d) DIC (µmol kg$^{-1}$) and (e) TA (µmol kg$^{-1}$)*

As mentioned before, we propose to change the Figure 3 of the previous manuscript by this figure that compares model and *in situ* values without any mean.

L236: Why is the flux not expressed in a mol per unit area value?

Reply: In our study as the model run only along time there is not an area dimension, then the flux is expressed in mmol m$^{-3}$ d$^{-1}$. All biogeochemical processes are expressed in mmol m$^{-3}$ d$^{-1}$.

L248: There is not really a decrease of seawater pCO$_2$, it just becomes much more variable

Reply: Here we wanted to highlight the impact of primary production on the seawater pCO$_2$. Looking at the figure E below (Fig. EC, black line), we can see net decreases of DIC (-10 mmol kg$^{-1}$) and seawater pCO$_2$ (-20 µatm) correlated with the increase in chlorophyll concentration. Then we prefer to do not change the sentence and to keep as written in the previous version of the manuscript.

[Figure]

*Figure E: Temporal focus between April 1st and July 1st, 2017. In situ daily average of (a) temperature (°C, black line) and salinity (blue line) at the SOLEMIO station. Modeled daily average, (b) chlorophyll concentrations (mg m⁻³, black line), (c) DIC (μmol kg⁻¹, black line) and pH (orange line),(d) seawater $pCO_2$ (μatm, black line) and atmosphere $pCO_2$ from OHP (μatm, red line), (e) air-sea $CO_2$ fluxes mmol m⁻³ d⁻¹, (f) Gross Primary Production (mmol m⁻³ d⁻¹, green line), total respiration (mmol m⁻³ d⁻¹, red line) and Net Ecosystem Production (mmol m⁻³ d⁻¹, black line).*

For sake of clarity, this figure could be added in the annex of the manuscript or in Figure 4 (on the right) of the manuscript.

L282: 'farer' -> further ?

Reply: Yes, we agree with the reviewer we modified the sentence as suggested L301.

L304: how can it affect the spring bloom before the nutrients are supplied?

Reply: In fact, this sentence needs to be corrected. Here we wanted to highlight, that the input of nutrients from river favors the primary production, and then the bloom of phytoplankton will occurs earlier than the reference simulation, which does not take into account rivers inputs. We modified the sentence as following L322-324:

*"It can be noted that with the strongest river supply at mid-March (Figs. 7A & 7B), the occurrence of the spring bloom is earlier (Fig. 7C) than that occurring in the reference simulation (S0)."*

L359: It sounds contradictory to say the 1.5°C rise affects the carbonate system little, but at L350 that the system is mostly driven by temperature variations

Reply: In winter the seawater temperature is around 13°C and in summer is around 20°C. However, in summer the upwelling events drop temperature by more than 5°C which have a strong impact on the carbonate system. In case of the scenario of a temperature rise of 1.5°C, we increase the time series of temperature by 1.5°C all along the year. This increase being smaller than the variations of temperature that occurs during upwelling events, the impact of +1.5°C over the year on carbonate system is less significant.

L363: double set of citations

Reply: We corrected this mistake.

Figures: give the legend in the figure panels instead of in the caption

Reply: We think that to insert the legend in the caption will overload the figure and will be unreadable. We decide then to keep the legend figures in the caption.

Fig. 6: what are those weird spikes in the air-sea gas exchange curve?

In figure 6, the two spikes in the air-sea gas exchange occur when the wind reaches the maximum velocity (20 m s$^{-1}$). Between these two spikes, the pCO$_2$ of seawater and air are balanced then the air-sea gas exchange is null.

---

## Author Comment (AC2) · 3 Aug 2020

Guy Munhoven (Referee)
guy.munhoven@uliege.be

**1 General comments**

**1.1 Appreciation of the manuscript**

In this paper, K. Lajaunie-Salla and co-authors present Eco3M-CarbOx, a biogeochemical model of the carbonate system and the plankton food web. It is integrated into the Ecological, Mechanistic and Modular Modelling framework, Eco3M (Baklouti et al., 2006b). In the paper, it is applied to the Bay of Marseille (Mediterranean Sea) for which there are comprehensive data time series available. I found the study interesting, but the paper has, unfortunately, a number of weaknesses. The paper is well-readable, but there remain numerous English errors. This manuscript has been submitted as a "model description paper." The model description in the main text reduces to thirty-two lines of text only (plus a schematic and a few equations in the appendix), of which fifteen deal with the new carbonate system module. These fifteen lines contain only a few commonplace statements followed by a sequence of eleven reference citations. This is far from what I expect to read in a "model description paper" in Geoscientific Model Development. It is also far from what is expected for that type of publication (see http://www.geoscientific-model-development.net/about/manuscript_types.html#item1). Accordingly, I do not even think that this manuscript fits the scope of the journal in its current form. Details about the approximations, numerical methods adopted and algorithms used, their applicability and their limitations are completely missing. Furthermore, some of the model experiments also have critical shortcomings: it is not realistic to assume that river water intrusions only impact the Total Alkalinity, TA, budget but not that of Dissolved Inorganic Carbon, DIC (riverine TA is mainly carried as $HCO_3$ which impacts the DIC and the TA budgets alike). For upwelling events, only a temperature effect is considered. However, upwelling events also bring nutrients, DIC and TA to the surface. The effects of these latter are not considered in the paper, possible effects not even discussed. I think the authors will first have to make up their mind whether they want to consider their manuscript as a "model description paper" in Geoscientific Model Development, or whether they would prefer to focus on the data analysis and interpretation. In both cases, they will have to extend the model description and revise the experimental design; in the second case, it would be recommended to submit this paper in other journal, such as, e.g., the sister journal Biogeosciences.

I am nevertheless convinced that this paper could make a valuable contribution to Geoscientific Model Development: from what I have been able to grasp from the paper and the code, the model approach looks solid and it could certainly be applied to other regions of the World as well. I therefore encourage the authors to go for the first option and prepare a **major revision** of their manuscript that includes a comprehensive model description and a sound experimental design.

Reply: We thank the reviewer for the evaluation of our work to be published in GMD journal. We thank too him for the detailed and useful comments that contributed to greatly improve

the manuscript. We consider his comments to improve the manuscript. We agree that is missing information about the new carbonate system module. In this way, we propose to add all the details of the carbonate system module in appendix (see the appendix is at the end of this document). Concerning the riverine inputs scenarii, we decide to focus on nitrate and alkalinity supply. In fact the model simulates the DIC increases, as is observed, which highlight that the carbonate system module is well resolved.

In this study we did not experiments the impact of an upwelling events (with a decrease of temperature). During upwelling events the seawater temperature could decrease by 5°C, which explain the strong temperature variations observed during summer (Fig.4A). Observations show strong variations of carbonates system variables during summer, which reproduce the model. These results are explained in the section 3.2 L280-285 of the MS.

**2 Specific comments**

**2.1 Abstract**

The abstract is not well focused and the hesitation between a model description and a data analysis approach is strongly visible.

Reply: Here, we think that the reviewer was confused with the windy periods, upwelling and temperature changes. Looking the temperature time series along the year 2017, we observe strong temperature variation in summer. These variations are due to upwelling events that bring bottom cold water to the surface. These upwelling events occur under specific wind direction and when the velocity is high, as during Mistral wind periods. To be clearer we modify the L24-26 as following:
*"Upwelling events change seawater temperature quickly, which alter the behavior of the BoM waters within a few days from a source of $CO_2$ to the atmosphere to a sink into the ocean."*

**2.2 Model description**

The model description is completely insufficient. For the underlying model, upon which Eco3M-CarbOx is built, only the ecological structure is summarized. Nothing is said about the spatial extension adopted: is it a point model? does it have some spatial extensions? 1D, 2D, 3D? In the code, one can see that state variable arrays are threedimensional, but it is not clear if the three spatial dimensions are actually used: the applied pressure, e.g., is set to 1 bar throughout, as if the model was applied for a water depth of about 10m only. If the model has some spatial extension, how are the lateral and bottom boundaries treated? The physical processes, although mentioned from time to time, are not at all dealt with here. How are they (e.g., transport processes) represented? This lack of description is rather incomprehensible as the authors themselves emphasize in their description of the study area that the biogeochemistry in the Bay of Marseille is "highly driven by hydrodynamics" (p. 3, l. 99).

Reply: We agree that this information was not clear in our previous version of the MS: The spatial dimension of the model is 0 in this study. The state variables of the model only change along time. As in this work the biogeochemical model is not coupled with a hydrodynamic model, the circulation is not taking into account. We add this information in L114-116, as following:
*"In this study, the state variables of the Eco3M-CarbOx model only change along time (i.e. usually termed "model 0D"), are representative of the time evolution of a sea surface water cell but this biogeochemical plankton model is not coupled with a hydrodynamic model."*

Any carbonate system speciation calculation procedure rests upon a TA approximation and a pH solver. Here, we do not read anything about these two elements:

• What TA approximation is adopted, i.e., which acid-base systems are taken into account?

• What pH-scale is adopted?

• Which numerical method is used to solve the resulting pH equation? What are the limitations of the adopted method (some methods fail to converge for low salinity water samples, e.g.)?

• Which parametrisations have been used for the stoichiometric constants?

I have been able to find answers to some of these questions by browsing though the code (although I am still not sure which pH scale is actually used in the end—probably pHtot). These informations must nevertheless be given in the main paper. It should not be necessary to inspect the code to find such basic informations.

As the model description stands, there is no way to reproduce the model results, a main requirement of model descriptions in Geoscientific Model Development.

Reply: We agree with the reviewer that information about the carbonates system speciation is not clearly specified. To resolve the carbonate system we use the value of DIC and TA to determine the value of pH and pCO2 and we use the total pH scale. DIC and TA are states variables of the Eco3M-CarbOx model.

The biogeochemical processes that impact DIC dynamics are: photosynthesis and respiration of phytoplankton, respiration of bacteria and zooplankton, precipitation and dissolution of CaCO3, and the CO2 exchange with the atmosphere.

The biogeochemical processes that impact TA dynamics are: uptake of nutrients by phytoplankton, mineralization of nitrogen organic matter by bacteria, nitrification and, precipitation and dissolution of CaCO3.

Taking into account the comments of the reviewer we propose to add an Appendix that gives all details about the resolution of the carbonate system. The appendix is at the end of this document. We add this following sentence L143-145:

*"The details of the resolution of carbonate system module are given at the appendix B. For this module three processes were also added: the precipitation and dissolution of calcium carbonate and the gas exchange of pCO2 with the atmosphere."*

**2.3 Experimental design**

As mentioned above, I find that there are inconsistencies in the design of the model experiments. In the model upwelling events, only a temperature effect is taken into account. However, as stated on p. 3, l. 100, such events also bring nutrient rich waters to the surface. Accordingly, they should also perturb the nitrate, DIC and TA balances. This is not what the model results reflect: they witness of cooling events only. Similarly, only part of the effects of Rhône river plume intrusions on the carbonate system are taken into account: the experiments only consider the resulting TA perturbation, but not the DIC perturbation. To my best knowledge, rivers mainly carry TA in the form of HCO₃ which impacts the DIC balance as strongly as the TA budget. I am even wondering—but could not find any decent data—if the River Rhône water does not also have high pCO2, in which case it would even carry more DIC than TA.

Reply: In this study we did not experiments the impact of an upwelling events (with a decrease of temperature). We evaluate the impact of different forçings: the temperature, wind, intrusion of nutrients, alkalinity, and atmospheric CO2.

In the BoM, upwelling events occurs during wind-specific conditions, as Mistral wind. During these days the seawater temperature could decrease by 5°C, which explain the strong temperature variations observed during summer (Fig.4A). Observations show strong variations of carbonates system variables during summer, which reproduce the model. These results are explained in the section 3.2 L280-285 of the MS.

As we did not experiments upwelling events with model, we did not take into account the surface water enrichments by nutrients and TA.

Concerning the riverine inputs scenarii, we decide to focus on nitrate and alkalinity supply. In fact the model simulates the DIC increases, as is observed, which highlight that the carbonate system module is well resolved.

**2.4 Code**

The code is provided on Zenodo and is easy to download. No reference to this manuscript is given on the model's entry page on Zenodo though. I have not tried to compile the code but only browsed through it as I was interested in getting at least some basic information about the new carbonate system model announced in the title. The code is commented, but most of the comments are unfortunately in French. This is especially annoying for the Makefile and the initialisation file BIO/config.ini which cannot be understood without a good proficiency in French. No user manual is provided, neither on the Zenodo page nor as a Supplement to the paper.

Reply: Yes, we agree to put comments of the "Makefile" and "config.ini" in English and to add an user manual as a Supplement.

**3 Technical comments**

Throughout the paper: please always specify which pH scale is used for reporting the data and model results (in tables, on graphs, etc.)

Reply: As suggested by the reviewer at the section of "numerical model description" we add the information that we use the total pH scale (L140). In this way we do not have to add "pH total scale" in all tables and figures.

p. 1, l. 5: is co-author "Irène Remy-Xueref" not actually "Irène Xueref-Remy"? (in the reference section at least the name is spelled that way and that is also the name registered in the submission system)

Reply: We corrected this mistake.

p. 1, l. 18: "22 states variables" should read "22 state variables"

Reply: We corrected as suggested.

p. 1, l. 33: "2018, May to 2019, May" should read "May 2018 to May 2019"

Reply: We corrected as suggested.

p. 2, ll. 42–43: this is unclear. I would not range the biological pump among the physical ones. But it is not sure what is meant here by "physical" pumps.

Reply: We agree with the reviewer we corrected as following L40-42:
*"In the ocean, the main processes regulating $CO_2$ exchanges between the atmosphere and sea are the solubility pump and the biological pump":*

p. 2, l. 54: "dynamic" should read "dynamics"

Reply: We corrected as suggested.

p. 2, l. 54: Why only "amplify"? A priori, the forcings could just as well attenuate or reduce acidification.

Reply: We agree with the reviewer we corrected as following L53-54:
*"Moreover, these forcings could affect the carbonate chemistry dynamics and amplify or reduce the acidification in coastal zones"*

p. 2, l. 56: "At a global scale" should read "At the global scale"

Reply: We corrected as suggested L56.

p. 2, l. 58: "as a net sink or source" may be more appropriate.

Reply: We corrected as suggested L58.

p. 2, l. 63: 'MacKenzie" should read "Mackenzie" (also misspelled in the bibliography)

Reply: We corrected as suggested.

p.2, ll. 75–76: "strong winds events" should read "strong wind events"

Reply: We corrected as suggested.

p. 3, l. 86: "implemented within" should read "implemented into"

Reply: We corrected as suggested.

p. 3, l. 95: Please delete "inhabitants" ("a population of ca. 1 million" is sufficient).

Reply: We corrected as suggested.

p. 3, l. 97: "winds conditions" should read "wind conditions"

Reply: We corrected as suggested.

p. 3, l. 101: "intrudes in the BoM" should read "intrudes the BoM" or "intrudes into the BoM"

Reply: We corrected as suggested.

p. 3, ll. 103–104: "diverse anthropogenic forcing" should read "diverse anthropogenic forcings"

Reply: We corrected as suggested.

p. 3, l. 106: delete "city"

Reply: We corrected as suggested.

p. 3, l. 113: "modeling platform" should read "platform" (there is no need to repeat the "modeling")

Reply: We corrected as suggested.

p. 3, l. 113: The paper cited (Baklouti et al., 2006a) is not adequate, as far as I can see. The companion paper Baklouti et al. (2006b), which describes the platform would be more appropriate. Baklouti et al. (2006a) review mechanistic formulations for key processes that control phytoplankton dynamics and present a generic model, less so the platform. Please check this.

Reply: We corrected as suggested.

p. 4, ll. 129–134: The TA production and consumption rates are stated in a very imprecise way here. As such these statement do not make much sense. It should be specified what are the references for the stated TA changes (e.g. TA decreases by two moles for each mole of $CaCO_3$ precipitated, and by x moles for each mole of XY assimilated by phytoplankton, etc.)

Reply: We corrected as following L145-151:
"Based on the review of Middelburg (2019), it is considered that: (i) TA decreases by 2 moles for each mole of $CaCO_3$ precipited, by 1 mole for each mole of ammonium nitrified, by 1 mole for each mole of ammonium assimilated by phytoplankton, and TA increases by 2 moles

for each mole of CaCO$_3$ dissolveld, and by 1 mole for each mole of organic matter mineralized by bacteria in ammonium (See Appendix A Tab. A2)"

p. 4, l. 132: "when bacteria mineralized" should read "when bacteria mineralize"

Reply: We corrected as suggested.

p. 4, l. 147: please delete "However" which does not make sense here.

Reply: We removed as suggested.

p. 4, l. 148: "model results" can be compared with the observations, not the model itself.

Reply: We corrected as suggested.

p. 5, l. 161: "winds specific conditions" should read "wind-specific conditions"

Reply: We corrected as suggested.

p. 6, l. 206: "contains a low value of WSS" – not sure what this could possibly mean. "a low value of WSS" should anyway read "a low WSS" as the last 'S' stands for 'score,' which is a value.

Reply: We corrected as suggested L224-225:
"The model does not catch the two aforementioned maxima of chlorophyll, and it contains a low WSS and a strong bias (0.37 and +0.22 mg m$^{-3}$, respectively - Tab. 2)."

p. 6, l. 214: "calculates a WSS value of 0.69" better had to read "yields a WSS of 0.69"

Reply: We modified this sentence as following L234-236:
"The statistical analysis calculated a mean bias of +23 µatm, and a WSS of 0.69 (Tab. 2)."

p. 6, l. 223: "seasonal dynamic" should read "seasonal dynamics"

Reply: We corrected as suggested.

p. 7, ll. 257–262: I am quite surprised about this. I would expect that upwelling events not only bring up cold water, but also nutrient, DIC and TA rich waters. Unfortunately, the model description does not explain how the upwelling events are represented. Could you please elaborate on this.

Reply: In this study we did not experiments the impact of an upwelling events (with a decrease of temperature). We evaluate the impact of different forçings: the temperature, wind, intrusion of nutrients, alkalinity, and atmospheric CO2.
As we did not experiments upwelling events with model, we did not take into account the surface water enrichments by nutrients and TA.

p. 8, l. 281: "diatoms" should read " diatoms' " (genitive)

Reply: We corrected as suggested.

p. 8, l. 295: "in-gassing" should read "absorption" or "uptake"

Reply: We corrected as following L317-319:
*"Depending on the gradient of CO$_2$ between seawater and the atmosphere, strong wind speeds will favor either the emission or uptake of CO$_2$ (Figs. 6B & E)."*

p. 8, l. 295: "variability" at which time scales?

Reply: We modified as following LXX:
*"The seasonal variability of atmospheric CO$_2$ concentrations at the urban site …"*..

p. 9, l. 329: "weaker" should read "lower"

Reply: We corrected as suggested.

p. 9, l. 354: I think that "counteracting" is more appropriate than "counterbalanced" at this point

Reply: We corrected as suggested.

p. 12, l. 424: "Environnenemnts" should read "Environnements"

Reply: We corrected as suggested.

p. 12, l. 432: "takes part" should read "is part

Reply: We corrected as suggested.

p. 12, l. 434: "Agence" should read "Agency"

Reply: We corrected as suggested.

p. 12, l. 434: "from European" should read "from the European"

Reply: We corrected as suggested.

p. 12, l. 434: "used in this paper" should read "presented in this paper"

Reply: We corrected as suggested.

**Appendix**: **Details of resolution of carbonate system module**

Details of resolution of carbonate system module

    1. **Calculation of carbonate systems constants:**

The constant are calculate in total pH scale

- Conversion of $DIC$ and $TA$ in mol kg$^{-1}$

$DIC = DIC.\rho_{eau}$ and $TA = TA.\rho_{eau}$

- $TF$ from Riley (1965) in mol.kg$^{-1}$:

$$TF = \frac{0.000067}{18.998}.\frac{S}{1.80655}$$

- $TS$ from Morris and Riley (1966) in mol.kg$^{-1}$:

$$TS = \frac{0.14}{96.062}.\frac{S}{1.80655}$$

- Concentration I from the DOE handbook, Chapter 5, p. 13/22, eq. 7.2.4:

$$IonS = \frac{19.924 * S}{1000 - 1.005 * S}$$

- Concentration Total borate from Uppström (1974) in mol.kg$^{-1}$:

$$TB = \frac{0.000416.S}{35}$$

- $KS$ constant of $HSO_4$ dissolution from Dickson (1990a) in mol.kg$^{-1}$:

$$KS = \frac{-4276.1}{T_{(K)}} + 141.328 - 23.093 * \log(T_{(K)}) + \left(324.57 - 47.986 * \log(T_{(K)}) - \frac{13856}{T_{(K)}}\right) * Ions^2-$$

$$KS = KS + \left(-771.54 + 114.723 * \log(T_{(K)}) + \frac{35474}{T_{(K)}}\right) * Ions + \frac{-2698}{T_{(K)}} * Ions^{1.5} + \frac{1776}{T_{(K)}} * Ions^2--$$

$$KS = e^{KS} * (1 - 0.001005 * S)$$

- $KF$ constant of HF formation from Dickson and Riley (1979) in mol.kg$^{-1}$:

$$KF = e^{\frac{1590.2}{T_{(K)}} - 12.641 + 1.525 * Ions^{1.5}} * (1 - 0.001005 * S)$$

- pH scale conversion factors:

$$SWStoTOT = \frac{1 + \frac{TS}{KS}}{1 + \frac{TS}{KS} + \frac{TF}{KF}} \text{ and } FREEtoTOT = 1 + \frac{TS}{KS}$$

- $KB$ constant from Dickson (1990b) this is in total pH scale in mol.kg$^{-1}$

$$KB = (-8966.9 - 2890.53 * S^{\frac{1}{2}} - 77.942 * S + 1.728 * S^{\frac{3}{2}} - 0.0996 * S^2)/T_{(K)}$$

$$KB = KB + 148.0248 + 137.1942 * S^{\frac{1}{2}} + 1.62142 * S + (-24.4344 - 25.085 * S^{\frac{1}{2}} - 0.2474 * S)$$
$$* \log(T) + 0.053105 * S^{\frac{1}{2}} * T$$

- $K0$ constant of $CO_2$ solubility from Weiss (1974) in mol.kg$^{-1}$atm$^{-1}$:

$$K0 = \exp\left(-60.2409 + 93.4517 * \frac{100}{T_{(K)}} + 23.3585 * \log\left(\frac{T_{(K)}}{100}\right) + S * \left(0.023517 - 0.023656 * \right.\right.$$

$$\left.\left.\frac{T_{(K)}}{100} + 0.0047036 * \left(\frac{T_{(K)}}{100}\right)^2\right)\right)$$

- $Ke$: produit ionique de l'eau from Millero (1995), this is in SWS pH scale $(mol.kg^{-1})^2$ :

$$Ke = \exp\left(\frac{-13847.26}{T_{(K)}} + 148.9802 - 23.6521 * \log(T_{(K)}) + \left(-5.977 + \frac{118.67}{T_{(K)}} + 1.0495 * \log(T_{(K)})\right) * S^{1/2} - 0.01615 * S\right)$$

$Ke = Ke * SWStoTOT$, in total pH scale in $mol.kg^{-1}$

- $K1$ and $K2$ from Lueker et al., (2000) these are in total pH scale in $mol.kg^{-1}$:

$$K1 = 10^{\frac{-3633.86}{T_{(K)}} + 61.2172 - 9.6777 * \log(T_{(K)}) + 0.011555 * S - 0.0001152 * S^2}$$

$$K2 = 10^{\frac{-471.78}{T_{(K)}} + 251.929 - 3.16967 * \log(T_{(K)}) + 0.01781 * S - 0.0001122 * S^2}$$

- $Kca$ for calcite from Mucci (1983) this is in $(mol.kg^{-1})^2$:

$$Kca = 10^{-171.9065 - 0.077993 * T_{(K)} + \frac{2839.319}{T_{(K)}} + 71.595 * \log10(T_{(K)}) + \left(-0.77712 + 0.0028426 * T_{(K)} + \frac{178.34}{T_{(K)}}\right) * S^{\frac{1}{2}} - 0.07711 * S + 0.0041249 * S^{1.5}}$$

- $Ca^{2+}$ concentration Riley and Tongudai (1967) in $mol.kg^{-1}$:

$$Ca = \frac{0.02128}{40.087} * \frac{S}{1.80655}$$

- The constant are corrected by pressure:

$R = 83.1451$ in $ml.bar^{-1}K^{-1}mol^{-1}$ and $P_{bar} = 1\ bar$

$$lnK1fac = \frac{\left(25.5 - 0.1271 * T_{(°C)} + 0.5 * \left(\frac{-3.08 + 0.0877 * T_{(°C)}}{1000}\right) * P_{bar}\right) * P_{bar}}{R * T_{(K)}}; K1 = K1 * e^{lnK1fac}$$

$$lnK2fac = \frac{\left(15.82 - 0.0219 * T_{(°C)} + 0.5 * \left(\frac{1.13 + 0.1475 * T_{(°C)}}{1000}\right) * P_{bar}\right) * P_{bar}}{R * T_{(K)}}: K2 = K2 * e^{lnK2fac}$$

$$lnKBfac = \frac{\left(29.48 - 0.1622 * T_{(°C)} + 0.002608 * T_{(°C)}^2 + 0.5 * \left(\frac{-2.84}{1000}\right) * P_{bar}\right) * P_{bar}}{R * T_{(K)}}; KB = KB * e^{lnKBfac}$$

$$lnKefac = \frac{\left(20.02 - 0.1119 * T_{(°C)} + 0.001409 * T_{(°C)}^2 + 0.5 * \left(\frac{-5.13 + 0.0794 * T_{(°C)}}{1000}\right) * P_{bar}\right) * P_{bar}}{R * T_{(K)}}; Ke = Ke * e^{lnKefac}$$

$$lnKFfac = \frac{\left(9.78 - 0.009 * T_{(°C)} + 0.0009429 * T_{(°C)}^2 + 0.5 * \left(\frac{-3.91 + 0.054 * T_{(°C)}}{1000}\right) * P_{bar}\right) * P_{bar}}{R * T_{(K)}}; KF = KF * e^{lnKFfac}$$

$$lnKSfac = \frac{\left(18.03 - 0.0466 * T_{(°C)} + 0.000316 * T_{(°C)}^2 + 0.5 * \left(\frac{-4.53 + 0.009 * T_{(°C)}}{1000}\right) * P_{bar}\right) * P_{bar}}{R * T_{(K)}}; KS = KS * e^{lnKSfac}$$

$$lnKcafac = \frac{\left(48.76 - 0.5304 * T_{(°C)} + 0.5 * \left(\frac{-11.76 + 0.3692 * T_{(°C)}}{1000}\right) * P_{bar}\right) * P_{bar}}{R * T_{(K)}}; Kca = Kca * e^{lnKcafac}$$

- Calculation of Fugacity factor:

We suppose that the pressure is at one atmosphere or close to it (Weiss, 1974):

$Patm = 1.01325\ bar$

$delta = 57.7 - 0.118 * T$ in $cm^3 mol^{-1}$

$b = -1636.75 + 12.0408 * T - 0.0327957 * T^2 + 3.16528 * 0.00001 * T^3$ in $cm^3 mol^{-1}$

$$FugFac = \exp^{\frac{(b + 2 * delta) * Patm}{R * T}}$$

**2. Resolution of carbonate system**

To resolve the carbonate system we calculate the $deltapH$, which is the difference of $pH$ between two iterations. We initialize the run imposing a $pH$ value of 8. Bellow the code details of pH and pCO2 determination:

```
if (nbiter < 1.) pH = 8

pHtol = 0.001 !tolerance for iterations end

deltapH = pHtol + 1

do while (abs(deltapH) > 0.0001)
```

$$H = 10^{-pH}$$
$$Denom = H^2 + K_1.H + K_1.K_2$$
$$CAlk = DIC.K_1.\left(\frac{H + 2.K_2}{Denom}\right)$$
$$BAlk = \frac{TB.KB}{KB + H}$$
$$OH = \frac{Ke}{H}$$
$$FreetoTot = 1 + \frac{TS}{KS}$$
$$Hfree = \frac{H}{FreetoTot}$$
$$HSO_4 = \frac{TS}{1 + \frac{KS}{Hfree}}$$
$$HF = \frac{TF}{1 + \frac{KF}{Hfree}}$$
$$Residual = TA - CAlk - BAlk - OH + Hfree + HSO_4 + HF$$
$$Slope = DIC.H.K_1.(H^2 + K_1.K_2 + 4.H.K_2)$$
$$Slope = \frac{Slope}{Denom^2} + OH + H + \frac{BAlk * H}{KB + H}$$
$$Slope = log10 * Slope$$

$deltapH = Residual/Slope$ ! *this is Newton's method*

**do while** $(abs(deltapH) > 1)$ $deltapH = \frac{deltapH}{2}$ ! *to keep the jump from being too big*

```
enddo
```

$pH = pH + deltapH$ !Is on the same scale as K1 and K2 were calculated, i.e. total pH scale

$pCO_2 = \left(\frac{DIC*H^2}{H^2+K1*H+K1*K2}\right) * \frac{10^6}{K0*FugFac}$ ! in µatm

$CO_2 = \frac{DIC*10^6}{1+\frac{K1}{H}+\frac{K1*K2}{H^2}}$

$HCO_3 = \frac{K1 * CO_2}{H}$

$CO_3 = \frac{K2 * HCO_3}{H}$

$Omega = \frac{Ca * CO_3 * 10^{-6}}{Kca}$

---

## Author Response (AR1)

Sebastiaan van de Velde (Referee)
sebastiv@ucr.edu

Lajaunie-Salla and coworkers present an extension of an existing food-web model with a carbonate chemistry balance. They subsequently use this model to look at the carbonate dynamics in the Bay of Marseille, and use sensitivity test to find under what circumstances the coastal waters of the BoM could be a source or sink for $CO_2$. Overall I found this an interesting study, even though its focus is very local. Depending on the flexibility, this model could likely also be used on different coastal sites.

I do have a few issues with the manuscript in its current state that I feel need to be resolved before it can be published. First and foremost, I had troubles to understand the model set-up. Given that this is a model development journal, the model should be clearly articulated in the main text, and this is not the case.

Secondly, I am not convinced by the authors discussion of the disagreement between model and data. Because they are using the model to look at dynamics in carbonate chemistry, they should either be able to reproduce the data in a better way, or at the very least discuss in more detail why there is a disagreement, and why that is not a problem. I do however think these issues can be dealt with in a revised version, and find the study itself valuable.

Reply: We thank the reviewer for the positive evaluation of our work to be published in GMD journal. We thank too him for the detailed and useful comments that contributed to greatly improve the manuscript.

This study is focused on the Bay of Marseille (BoM) that harbors the big metropolis of the aforementioned town hosting more than 800 000 inhabitants, with in summer an increasing of tourism activities. Moreover, the BoM is impacted by many harbor activities. In this context, we think so that the main results of our study could be transposed to other coastal sites that are also impacted by urban and anthropic pressures.

We modify the conclusion of manuscript L439-450 as following:

*"The BoM biogeochemical functioning is mainly forced by wind-driven hydrodynamics (upwelling, downwelling), urban rivers, wastewater treatment plants, and atmospheric deposition (Fraysse et al., 2013). In addition, Northern Current and Rhone River plume intrusions frequently occurred (Fraysse et al., 2014; Ross et al., 2016). Moreover, the BoM harbors the second bigger metropolis of France (Marseille) that is impacted by many harbor activities. The next step of this study will be to couple the Eco3M-CarbOx biogeochemical model to a 3D hydrodynamic model that will mirror the complexity of the BoM functioning. In this way, the contributions of hydrodynamic, atmospheric, anthropic, and biogeochemical processes to the DIC variability will be able to be determined with higher refinement and realism, and an overview of the air-sea $CO_2$ exchange could be made at the scale of the Bay of Marseille. The main results of our study could be transposed to other coastal sites that are also impacted by urban and anthropic pressures. Moreover, in this paper we highlighted that fast and strong variations of $pCO_2$ values occur, so thus it is essential to acquire more in situ values at high frequency (at least with an hourly resolution) to understand the rapid variations of the marine carbon system at these short spatial and temporal scales."*

As suggested by the reviewer we give more details about the model set-up and the modifications made from the previous version. Moreover, we propose to change the figure 3 of the previous manuscript by the figure that compares model and in situ values without any mean. This figure shows better the good reproduction of the in situ data by the model.

**Comments:**

1. Model Description:

I do not think the model development or set-up has been well described in the text. All equations and parameters are to be found in a number of tables of the appendix, and the readers are expected to either know the plankton model used, or go to other papers to find it. This might be fine if it was an established model and in a different journal, but I do not think it is good for GMD. The reasoning behind model set-up and parameterization is not explaining in the text, so it is difficult to understand why the model was set-up as it was. After reading the methods I still had a number of basic questions;

Reply: The Eco3m-CarbOx model is based on a pre-existing model of the plankton ecosystem developed by Fraysse et al. (2013). The model presented in our study includes a set of new developments and improvements in the realism of the plankton web structure and process formulations. In order to improve the representation of chlorophyll concentration in the Bay of Marseille, two types of phytoplankton were added: the Synechococcus cyanobacteria, which is one of the major constitutive members of picophytoplankton in Mediterranean Sea (Mella-Flores et al., 2011), and the large diatoms, which are generally observed during spring blooms at mid-latitudes (Margalef, 1978). The functional response of primary production was modified using another formulation of temperature limitation function which takes into account the optimal temperature of growth for each phytoplankton (diatoms and picoplankton). The exudation of phytoplankton was modified taking into account the intracellular phytoplankton ratio of C, N and P. For the assimilation of matter (inorganic and organic) by bacteria and the remineralization processes the dependence on intracellular bacteria ratio was added. A temperature dependence of all biogeochemical processes was added. Also certain parameters in some formulations were modified. The Eco3m-CarbOx model also hosts now a carbonate system module. For this three processes were added (i) the precipitation, (ii) dissolution of carbonate and (iii) the gas exchange with the atmosphere. We agree with the reviewer on this point and we added some important elements on the approach of alteration of the pre-existing model. However, we think that, for a sake of clarity, it is important to keep in the appendix section the tables including source-sink equations and the long lists of model parameters.

We modify a part of the "Numerical model description" section from L103-126 as following:
*"The model presented in this study includes a set of new developments and improvements in the realism of the plankton web structure and process formulations compared to the model of Fraysse et al. (2013). In order to improve the representation of chlorophyll concentration in the Bay of Marseille the phytoplankton is divided in two groups: one with some ecological and physiological traits of the Synechococcus cyanobacteria, which is one of the major constitutive members of pico-autotrophs in Mediterranean Sea (Mella-Flores et al., 2011), and another with traits of large diatoms, which are generally observed during spring blooms at mid-latitudes (Margalef, 1978). For both of the phytoplankton, there is a diagnostic chlorophyll-a variable related to the phytoplankton C-biomass, the phytoplankton N-to-C ratio, and the limiting internal ratio $f_Q^N$ (Faure et al., 2010; Smith and Tett, 2000; Tab. B2). The functional response of primary production was modified using another formulation of temperature limitation function which takes into account the optimal temperature of growth*

*for each phytoplankton group. The exudation of phytoplankton was modified taking into account the intracellular phytoplankton ratio. For the uptake of matter by bacteria and the remineralization processes the dependence on intracellular bacteria ratio was modified. A temperature dependence of all biogeochemical processes was added to take into account the effects of rapid and strong variations of seawater temperature on plankton during episodes of upwelling for instance that are usually observed in the BoM. Also certain parameters in some formulations were modified owing to the alterations of some formulations.*
*Additionally, a carbonate system module was developed and three state variables were added: dissolved inorganic carbon (DIC), total alkalinity (TA) and the calcium carbonate (CaCO₃) implicitly representing calcifying organisms. The knowledge of DIC and TA allows the calculation of the $pCO_2$ and pH (total pH scale) diagnostic variables, necessary for resolving all the equations of the carbonate system. These equations use apparent equilibrium constants, which depend on temperature, pressure, and salinity (Dickson, 1990a, 1990b; Dickson and Riley, 1979; Lueker et al., 2000; Millero, 1995; Morris and Riley, 1966; Mucci, 1983; Riley, 1965; Riley and Tongudai, 1967; Uppström, 1974; Weiss, 1974). The details of the resolution of carbonate system module are given in the Appendix A. For this module three processes were also added: the precipitation and dissolution of calcium carbonate and the gas exchange of $pCO_2$ with the atmosphere."*

(i) what are the dimensions of the model (1 box, 3D, : : :)?

Reply: The spatial dimension of the model is 0 in this study. The state variables of the model only change along time. We add this information in L100-102, as following:
*"In this study, the state variables of the Eco3M-CarbOx model only change along time (i.e. usually termed "model 0D"), they are representative of the time evolution of a sea surface water cell but this biogeochemical model is not coupled with a hydrodynamic model."*

(ii) It is stated at L153 that the variables were initialized at winter conditions and forced with measured temperature etc. Does it not require a spin-up for the circulation – that is, presuming it has circulation?

Reply: We agree that this information was not clear in our previous version of the MS. As in this work the biogeochemical model is not coupled with a hydrodynamic model, the circulation is not taking into account. Please refer to the previous comment.

(iii) Why do you choose two three-day periods of wind speed? Why 7 m s$^{-1}$? What are the boundary conditions? How does the BoM connect to the rest of the Mediterranean?

Reply: As we can see in the figure below, which represents the time series of wind speed at the SOLEMIO station located in the BoM, the studied area is highly impacted by windy periods with speed values above 20 m s$^{-1}$. In 2017 the average wind speed was 7 m s$^{-1}$. In order to analyze the impact of these windy periods on the carbonate system we run the model with wind speed of 20 m s$^{-1}$ during three days. We choose three-day periods because the BoM is often impacted by short bursts of Mistral. This wind usually blows for a few days with high speeds ranging from 14 to 28 m s$^{-1}$.

[Figure]

*Figure A: Time series of wind velocity (m s$^{-1}$) at SOMLIT station between July 1st and August 1st, 2017 (gray line) and the average of wind velocity (red line).*

We modified the manuscript L181-184 as following:

*"Impact of wind events: a first simulation S2 was run with a constant wind intensity of 7 m s$^{-1}$ (2017 annual average wind speed) throughout the year and a second one (S3) with two three-day periods of strong wind speed (20 m s$^{-1}$) representative of short bursts of Mistral (data not shown) starting on May 15$^{th}$ and August 15$^{th}$, and a constant value of 7 m s$^{-1}$ the rest of the year."*

Furthermore, there seems to be a number of inconsistencies between tables and between tables and text, for example;

(i) In Table A3 you use 'POM' for the bacterial grazing term, but that does not show up in your state variable list (it is likely detritus, but then it should be called that, otherwise it causes confusion).

Reply: We agree with the reviewers to modify in the manuscript the term "detritus" by "DPOM" to define the detrital particulate organic matter. In same way we modify the term dissolved organic matter "DOM", by the labile and semi labile dissolved organic carbon "LDOM". In the manuscript these changes were made in the tables B1-B2 and B3 and in L18, L98 and the Fig. 1.

(ii) (O/N)nit in Table A2 -> (O/C)nit in Table A5, (O/N)uptNO3 in Table A2 -> (O/C)uptNO3 in Table A5 :

Reply: We agree with the Reviewer that there are several mistakes on the tables B2 and B5. In this study, the ratio $\frac{O}{C}$ and $\frac{O}{N}$ are equal to 1 and 2, respectively. There is no need to differentiate the name of this ratio for each process. We choose then to use only the ratio $\frac{O}{C}$ and $\frac{O}{N}$ without sub-index, we then modify the tables B2 and B5 considering these changes:

$$\left(\frac{O}{C}\right) = \left(\frac{O}{C}\right)_{PP} = \left(\frac{O}{C}\right)_{resp} = \left(\frac{O}{C}\right)_{respZ} = \left(\frac{O}{C}\right)_{respBa} = 1$$

$$\left(\frac{O}{N}\right) = \left(\frac{O}{N}\right)_{uptNO_3} = \left(\frac{O}{N}\right)_{nit} = 2$$

It is stated at L138 there is closure term, but I did not find where the grazing term closes the balance?

Reply: In the model, all the matter grazed by the zooplankton, (i.e. bacteria, phytoplankton, and DPOM) return as either organic or inorganic matter by excretion, egestion and mortality processes (Fig. B).

- The excretion of zooplankton returns dissolved inorganic matter with a flux of :

$$R_{excr}^{DIM} = \varepsilon_{DIM} \cdot d_X \cdot \left(1 - k_{X,zoo}\right) \cdot \left(R_{Gr}^{Phy} + R_{Gr}^{DPOM} + R_{Gr}^{Ba}\right)$$

- The excretion of zooplankton returns labile dissolved organic matter with a flux of :

$$R_{excr}^{LDOM} = (1 - \varepsilon_{DIM}) \cdot d_X \cdot \left(1 - k_{X,zoo}\right) \cdot \left(R_{Gr}^{Phy} + R_{Gr}^{DPOM} + R_{Gr}^{Ba}\right)$$

- The egestion of zooplankton returns detrital particulate organic matter with a flux of :

$$R_{pf} = (1 - d_X) \cdot \left(R_{Gr}^{Phy} + R_{Gr}^{DPOM} + R_{Gr}^{Ba}\right)$$

- The mortality of zooplankton returns detrital particulate organic matter with a flux of :

$$R_m = d_X \cdot k_{X,zoo} \cdot \left(R_{Gr}^{Phy} + R_{Gr}^{DPOM} + R_{Gr}^{Ba}\right)$$

[Figure]

*Figure B: Repartition of matter grazed by zooplankton*

The sum of these four processes is equal to the flux of matter grazed by zooplankton:

$$R_{excr}^{DIM} + R_{excr}^{LDOM} + R_{pf} + R_m = R_{Gr}^{Phy} + R_{Gr}^{DPOM} + R_{Gr}^{Ba}$$

This is the way whom the biogeochemical model is closed in this study.

2. Model-data agreement:

You discuss the model-data comparison in section 3.1, but seem to brush over some of the misfits quite easily. For example, the alkalinity in Fig.3
- you say the model results are within the range of the data, but they are only barely within the range, and most of the data plots above the model values.

Reply: We agree with the Reviewer and modify the sentence L245-247 as following:
"The biogeochemical model provides almost constant values around 2570 µmol kg$^{-1}$ all along the year, which is lower than *in situ* data."

- At L208 you say that the model shows the same trends for pH and pCO$_2$, but you have a consistent offset in the first half of your pCO$_2$ graph? And the trends seem to be inverse in the first half (blue dots going up and orange down) and last part (orange up and blue down) of the graph. Same for the pH (which is what you would expect as they are coupled) I would think that if you want to use the model to investigate carbonate dynamics under future climate change scenarios, you would want to be able to reproduce (or explain) what happens with the alkalinity and pH? Those model-data comparison does not give much confidence to the model (or parametrization) if you specifically want to address carbonate related questions.

Reply: We agree with the Reviewer that in the figure 3 we cannot see well the variability of $pCO_2$ and $pH$, because the plotted data from model is a mean value at ±5 days around the sampling date. Looking at the figure C below, that represents the simulated data and in situ data, we can see that the model reproduces the dynamic of seasonal variations of seawater $pCO_2$ and $pH$.

[Figure]

*Figure C: Comparison of model results (red) and in situ data (blue) at the SOLEMIO station. (a) pCO₂ (µatm) and (b) pH*

From January to February, the model reproduces the observed decrease in $pCO_2$ and from February to March the increase in $pCO_2$. In mid-April the observed drop in $pCO_2$ and increase in $pH$ are also spotted in the model (Fig. C green box). During summer, the dynamics linked to the temperature variations are also well reproduced by the model. We propose to change the Figure 3 of the previous MS by the figure that compares model and in situ values without any mean.

We modify the manuscript L222-232 as following:

*"On the whole, the seasonal variations of the seawater pCO₂ are correctly simulated by the biogeochemical model (Fig. 3B), even if the values are rather overestimated during MWC period. From January to February, the model reproduces the slight decrease in the observed pCO₂ and from February to March the increase in pCO₂ even if the latter modelled remains smaller. In mid-April, during the simulated spring bloom period, the observed drop in pCO₂ and increase in pH are also spotted in the model (Fig. 3B & 3C). The model especially succeeds in reproducing the observed increase in relation to high temperatures during the SWC period. The reduction of the CO₂ solubility due to thermal effects mostly explains the increase in pCO₂ during the SWC period. The strong standard deviation of modeled values during the SWC period can be explained by the rapid changes in temperature probably due to upwelling usually occurring at this time of the year (Millot, 1990). The range of modeled pCO₂ values (345 - 503 µatm) encompasses the range of observed values (358 - 471 µatm; Tab. 2). The statistical analysis provides a mean bias of +23 µatm, and a WSS of 0.69 (Tab. 2)."*

3. General readability:

In general, the text reads a bit awkward, and seems to have a strong French influence (I mean no offense, but that is just the way it felt when I was reading it). In particular, oddly placed articles, and plural forms where it should be singular. The manuscript could probably benefit

from proofreading by a native speaker. Then again, I might be wrong as I am also not a native speaker, and this is merely a suggestion.

Reply: Before submitting the new revised version, we will give it to a native speaker for proofreading and English improvements.

**Minor comments:**

L29: 'strong atmospheric $CO_2$' -> do you mean high concentrations?

Reply: We agree with the reviewer and changed "strong" by "high".

L41: are you considering the biological pump to be a physical process?

Reply: In fact, the sentence no was clear, we modified as following L46-48:
*"In the Ocean, the main processes regulating $CO_2$ exchanges between the atmosphere and sea are the solubility pump and the biological pump at different time scales"*

L47: 'organic matters' -> organic matter

Reply: We corrected as suggested.

L143: just 'dataset' suffices

Reply: We corrected as suggested.

L149: Why did you not plot the temporal trend versus the datapoints?

Reply: As the in situ data are recorded every 15 days, and in order to do a statistical analysis and have the same number of data, we used the mean value at ±5 days around the sampling date. In the figure D below, we plot the temporal trend versus the datapoints without any mean.

[Figure]

*Figure D: Comparison of model results (orange) and in situ data (blue) at the SOLEMIO station. (a) Chlorophyll-a concentrations (mg m$^{-3}$), (b) pCO$_2$ (µatm), (c) pH, (d) DIC (µmol kg$^{-1}$) and (e) TA (µmol kg$^{-1}$)*

As mentioned before, we have changed the Figure 3 of the previous MS by this figure that compares model and in situ values without any mean.

L236: Why is the flux not expressed in a mol per unit area value?

Reply: In our study as the model run only along time there is not an area dimension, then the flux is expressed in mmol m$^{-3}$ d$^{-1}$. All biogeochemical processes are expressed in mmol m$^{-3}$ d$^{-1}$.

L248: There is not really a decrease of seawater pCO$_2$, it just becomes much more variable

Reply: Here we wanted to highlight the impact of primary production on the seawater $p$CO$_2$. Looking at the figure E below (Fig. E panel C, black line), we can see net decreases of DIC (-10 mmol kg$^{-1}$) and seawater $p$CO$_2$ (-20 µatm) correlated with the increase in chlorophyll concentration. Then we do not prefer to change the sentence and to keep as written in the previous version of the ms.

[Figure]

*Figure E: Temporal focus between April 1$^{st}$ and July 1$^{st}$, 2017. In situ daily average of (a) temperature (°C, black line) and salinity (blue line) at the SOLEMIO station. Modeled daily average, (b) chlorophyll concentrations (mg m$^{-3}$, black line), (c) DIC (μmol kg$^{-1}$, black line) and pH (orange line), (d) seawater pCO$_2$ (μatm, black line) and atmosphere pCO$_2$ from OHP (μatm, red line), (e) air-sea CO$_2$ fluxes mmol m$^{-3}$ d$^{-1}$), (f) Gross Primary Production (mmol m$^{-3}$ d$^{-1}$, green line), total respiration (mmol m$^{-3}$ d$^{-1}$, red line) and Net Ecosystem Production (mmol m$^{-3}$ d$^{-1}$, black line).*

For sake of clarity, the figure 4 has been modified and the focus on the right panel goes from April 1$^{st}$ to July, 31$^{th}$ 2017.

L282: 'farer' -> further?

Reply: Yes, we agree with the reviewer we modified the sentence as suggested.

L304: how can it affect the spring bloom before the nutrients are supplied?

Reply: In fact, this sentence needed to be corrected. Here we wanted to highlight, that the input of nutrients from river favors the primary production, and then the bloom of phytoplankton will occur earlier than that in the reference simulation, which does not take into account rivers inputs. We modified the sentence as following L323-324

"It can be noted that with the strongest river supply at mid-March (Figs. 7A & 7B), the occurrence of the spring bloom is earlier (Fig. 7C) than that occurring in the reference simulation (S0)."

L359: It sounds contradictory to say the 1.5°C rise affects the carbonate system little, but at L350 that the system is mostly driven by temperature variations

Reply: In winter the seawater temperature is around 13°C and in summer is around 20°C. However, in summer the upwelling events drop temperature by more than 5°C which has a strong impact on the carbonate system. In case of the scenario of a temperature rise of 1.5°C, the seawater temperature is increased by 1.5°C all along the year. This increase being smaller than the variations of temperature that occur during upwelling events, the impact of +1.5°C over the year on carbonate system is less significant.

L363: double set of citations

Reply: We corrected this mistake.

Figures: give the legend in the figure panels instead of in the caption

Reply: We think that to insert the legend in the caption will overload the figure and will be unreadable. We prefer then to keep the legend figures in the caption.

Fig. 6: what are those weird spikes in the air-sea gas exchange curve?

In figure 6, the two spikes in the air-sea gas exchange occur when the wind reaches the maximum velocity ($20$ m s$^{-1}$). Between these two spikes, the $p$CO$_2$ of seawater and air are balanced then the air-sea gas exchange is null.
In this paper, K. Lajaunie-Salla and co-authors present Eco3M-CarbOx, a biogeochemical model of the carbonate system and the plankton food web. It is integrated into the Ecological, Mechanistic and Modular Modelling framework, Eco3M (Baklouti et al., 2006b). In the paper, it is applied to the Bay of Marseille (Mediterranean Sea) for which there are comprehensive data time series available. I found the study interesting, but the paper has, unfortunately, a number of weaknesses. The paper is well-readable, but there remain numerous English errors. This manuscript has been submitted as a "model description paper." The model description in the main text reduces to thirty-two lines of text only (plus a schematic and a few equations in the appendix), of which fifteen deal with the new carbonate system module. These fifteen lines contain only a few commonplace statements followed by a sequence of eleven reference citations. This is far from what I expect to read in a "model description paper" in Geoscientific Model Development. It is also far from what is expected for that type of publication (see http://www.geoscientific-model-development.net/about/manuscript_types.html#item1). Accordingly, I do not even think that this manuscript fits the scope of the journal in its current form. Details about the approximations, numerical methods adopted and algorithms used, their applicability and their limitations are completely missing. Furthermore, some of the model experiments also have critical shortcomings: it is not realistic to assume that river water intrusions only impact the Total Alkalinity, TA, budget but not that of Dissolved Inorganic Carbon, DIC (riverine TA is mainly carried as $HCO_3$ which impacts the DIC and the TA budgets alike). For upwelling events, only a temperature effect is considered. However, upwelling events also bring nutrients, DIC and TA to the surface. The effects of these latter are not considered in the paper, possible effects not even discussed. I think the authors will first have to make up their mind whether they want to consider their manuscript as a "model description paper" in Geoscientific Model Development, or whether they would prefer to focus on the data analysis and interpretation. In both cases, they will have to extend the model description and revise the experimental design; in the second case, it would be recommended to submit this paper in other journal, such as, e.g., the sister journal Biogeosciences.

I am nevertheless convinced that this paper could make a valuable contribution to Geoscientific Model Development: from what I have been able to grasp from the paper and the code, the model approach looks solid and it could certainly be applied to other regions of the World as well. I therefore encourage the authors to go for the first option and prepare a **major revision** of their manuscript that includes a comprehensive model description and a sound experimental design.

Reply: We thank the reviewer for the serious evaluation of our work to be published in GMD journal. We thank too him for the detailed and useful comments that, we hope, contributed to greatly improve the manuscript. We consider most of his comments to improve the manuscript.

To answer to the major weakness of the submitted manuscript (hesitation between a technical or analysis article) we have changed the revised version at many places (*e.g.* abstract, goals of paper at end of introduction, rearrangement of materials and methods, discussion, conclusion) to give a more technical outline to this paper as suggested by the Reviewer.

We agree that there was a lack information about the carbonate system new module. In this way, we added all the details on the module of the carbonate system, integration methods, and approximations in a new appendix A (commented in English).

Concerning the numerical experiments, we know that there are unrealistic especially for river supply and upwelling but we initially wanted to test the responses of the new carbonate module on simple cases. Concerning the riverine inputs scenario for instance, we decided to focus on the nitrate and alkalinity supply. And we can see that the model simulates the DIC increases, as observed, which highlights that the carbonate system module is well resolved by the model. However, as suggested by the Reviewer and to improve our manuscript we now discuss the limits of the numerical experiments and associated outcomes in the discussion (*e.g.* L.390-395; L408-414).

**2 Specific comments**

2.1 Abstract

The abstract is not well focused and the hesitation between a model description and a data analysis approach is strongly visible.

Reply: We agree with this remark. We have quite strongly changed the abstract to give it a more technical outline.

2.2 Model description

The model description is completely insufficient. For the underlying model, upon which Eco3M-CarbOx is built, only the ecological structure is summarized. Nothing is said about the spatial extension adopted: is it a point model? Does it have some spatial extensions? 1D, 2D, 3D? In the code, one can see that state variable arrays are three dimensional, but it is not clear if the three spatial dimensions are actually used: the applied pressure, *e.g.*, is set to 1 bar throughout, as if the model was applied for a water depth of about 10m only. If the model has some spatial extension, how are the lateral and bottom boundaries treated? The physical processes, although mentioned from time to time, are not at all dealt with here. How are they (e.g., transport processes) represented? This lack of description is rather incomprehensible as the authors themselves emphasize in their description of the study area that the biogeochemistry in the Bay of Marseille is "highly driven by hydrodynamics" (p. 3, l. 99).

Reply: We agree that this information was not clear in our previous version: The spatial dimension of the model is 0 in this study. The state variables of the model only change along time. As in this work the biogeochemical model is not coupled with a hydrodynamic model, the hydrodynamic circulation is not taking into account. We add this information in L100-L102, as following:

"*In this study, the state variables of the Eco3M-CarbOx model only change along time (i.e. usually termed "model 0D"),* are representative of the time evolution of a sea surface water cell *but this biogeochemical plankton model is not coupled with a hydrodynamic model.*"

Any carbonate system speciation calculation procedure rests upon a TA approximation and a pH solver. Here, we do not read anything about these two elements:
• What TA approximation is adopted, i.e., which acid-base systems are taken into account?
• What pH-scale is adopted?
• Which numerical method is used to solve the resulting $p$H equation? What are the limitations of the adopted method (some methods fail to converge for low salinity water samples, e.g.)?
• Which parametrisations have been used for the stoichiometric constants?
I have been able to find answers to some of these questions by browsing through the code (although I am still not sure which pH scale is actually used in the end—probably pHtot). These informations must nevertheless be given in the main paper. It should not be necessary to inspect the code to find such basic informations.
As the model description stands, there is no way to reproduce the model results, a main requirement of model descriptions in Geoscientific Model Development.

Reply: We agree with the Reviewer that information about the carbonate system speciation was not clearly specified. To resolve the carbonate system, we use the values of DIC and TA to determine the values of $p$H and $p$CO$_2$. DIC and TA are two state variables of the Eco3M-CarbOx model.
The biogeochemical processes that drive the DIC dynamics are (i) photosynthesis and respiration of phytoplankton, (ii) respiration of bacteria and zooplankton, (iii) precipitation and dissolution of $CaCO_3$, and (iv) the $CO_2$ exchange with the atmosphere.
The biogeochemical processes that drive TA dynamics are (i) uptake of nutrients by phytoplankton, (ii) mineralization of nitrogen organic matter by bacteria, (iii) nitrification and (iv) precipitation and dissolution of $CaCO_3$.
Taking into account the comments of the reviewer we added a new Appendix (Appendix A) that gives all details about the equations used, resolution of the carbonate system. This appendix is now at the end of the revised manuscript.

2.3 Experimental design

As mentioned above, I find that there are inconsistencies in the design of the model experiments. In the model upwelling events, only a temperature effect is taken into account. However, as stated on p. 3, l. 100, such events also bring nutrient rich waters to the surface. Accordingly, they should also perturb the nitrate, DIC and TA balances. This is not what the model results reflect: they witness of cooling events only. Similarly, only part of the effects of Rhône river plume intrusions on the carbonate system are taken into account: the experiments only consider the resulting TA perturbation, but not the DIC perturbation. To my best knowledge, rivers mainly carry TA in the form of HCO3 which impacts the DIC balance as strongly as the TA budget. I am even wondering—but could not find any decent data—if the River Rhône water does not also have high pCO2, in which case it would even carry more DIC than TA.

Reply: We agree with the Reviewer on this set of inconsistencies. We have changed in some places the text of manuscript to indicate that upwelling events (and Rhône intrusions) usually occur in the BoM but in our numerical study we do not per se experiment the impact of an upwelling or intrusion events under a realistic way. We now underline that we only evaluate the impact of different forcing on the marine carbonate dynamics as variations of seawater temperature or wind speed, supplies of nutrients or alkalinity, and levels of atmospheric $CO_2$. We also added comments in the discussion on the limits of our numerical approach and indicate that the present study will be extended in a next step in a 3D modelling context (see discussion at L393 and conclusion at L443).

Concerning the riverine inputs scenarii, it is true, that the Rhone River supplies only nitrate and TA in our study and not DIC. Furthermore, we agree with the Reviewer that DIC is probably supplied in high concentrations by the Rhône River and in turn, the DIC concentration provides by the model is likely underestimated and, *in fine*, this involves an inaccurate view of the carbonate balance. We have added a comment on this point in the discussion (L409-L411). We initially decided to focus on nitrate and alkalinity supply only because (i) neither the DIC concentrations nor $p$CO$_2$ levels have been yet measured in the Rhône River, to our knowledge and (ii) we wanted to test the model with as simple as possible perturbations. In fact, we can see that the carbonate system balance is well resolved by the model since, during intrusions (low salinity periods, see Fig. 4), TA supply and low salinity involve a DIC increase, owing to a $p$CO$_2$ decrease which is an expected response of the carbonate balance with such perturbation (Middelburg, 2019).

*2.4 Code*

The code is provided on Zenodo and is easy to download. No reference to this manuscript is given on the model's entry page on Zenodo though. I have not tried to compile the code but only browsed through it as I was interested in getting at least some basic information about the new carbonate system model announced in the title. The code is commented, but most of the comments are unfortunately in French. This is especially annoying for the Makefile and the initialization file BIO/config.ini which cannot be understood without a good proficiency in French. No user manual is provided, neither on the Zenodo page nor as a Supplement to the paper.

Reply:As suggested by the Reviewer, we added a new Appendix C in the paper including a short user manual as on the Zenodo dedicated page. Furthermore, the reference to the present study is now included on the Zenodo page. The Makefiles and the config.ini file were commented in English as requested.

**3 Technical comments**

Throughout the paper: please always specify which pH scale is used for reporting the data and model results (in tables, on graphs, etc.)

Reply: As suggested by the reviewer at the section of "numerical model description" we add the information that we use the total $p$H scale (L120).

p. 1, l. 5: is co-author "Irène Remy-Xueref" not actually "Irène Xueref-Remy"? (in the reference section at least the name is spelled that way and that is also the name registered in the submission system)

Reply: We corrected this mistake.

p. 1, l. 18: "22 states variables" should read "22 state variables"

Reply: We corrected as suggested.

p. 1, l. 33: "2018, May to 2019, May" should read "May 2018 to May 2019"

Reply: We corrected as suggested.

p. 2, ll. 42–43: this is unclear. I would not range the biological pump among the physical ones. But it is not sure what is meant here by "physical" pumps.

*Reply: We agree with the reviewer we corrected as following L46-48*
*"In the Ocean, the main processes regulating CO$_2$ exchanges between the atmosphere and sea are the solubility pump and the biological pump":*

p. 2, l. 54: "dynamic" should read "dynamics"

Reply: We corrected as suggested.

p. 2, l. 54: Why only "amplify"? A priori, the forcings could just as well attenuate or reduce acidification.

Reply: We agree with the reviewer we corrected as following L58-59.
*"Moreover, these forcings could affect the carbonate chemistry dynamics and amplify or attenuate the acidification in coastal zones"*

p. 2, l. 56: "At a global scale" should read "At the global scale"

Reply: We corrected as suggested.

p. 2, l. 58: "as a net sink or source" may be more appropriate.

Reply: We corrected as suggested.

p. 2, l. 63: 'MacKenzie" should read "Mackenzie" (also misspelled in the bibliography)

Reply: We corrected as suggested.

p.2, ll. 75–76: "strong winds events" should read "strong wind events"

Reply: We corrected as suggested.

p. 3, l. 86: "implemented within" should read "implemented into"

Reply: We corrected as suggested.

p. 3, l. 95: Please delete "inhabitants" ("a population of ca. 1 million" is sufficient).

Reply: We corrected as suggested.

p. 3, l. 97: "winds conditions" should read "wind conditions"

Reply: We corrected as suggested.

p. 3, l. 101: "intrudes in the BoM" should read "intrudes the BoM" or "intrudes into the BoM"

Reply: We corrected as suggested.

p. 3, ll. 103–104: "diverse anthropogenic forcing" should read "diverse anthropogenic forcings"

Reply: We corrected as suggested.

p. 3, l. 106: delete "city"

Reply: We corrected as suggested.

p. 3, l. 113: "modeling platform" should read "platform" (there is no need to repeat the "modeling")

Reply: We corrected as suggested.

p. 3, l. 113: The paper cited (Baklouti et al., 2006a) is not adequate, as far as I can see. The companion paper Baklouti et al. (2006b), which describes the platform would be more appropriate. Baklouti et al. (2006a) review mechanistic formulations for key processes that control phytoplankton dynamics and present a generic model, less so the platform. Please check this.

Reply: We corrected as suggested.

p. 4, ll. 129–134: The TA production and consumption rates are stated in a very imprecise way here. As such these statement do not make much sense. It should be specified what are the references for the stated TA changes (*e.g.* TA decreases by two moles for each mole of $CaCO_3$ precipitated, and by x moles for each mole of XY assimilated by phytoplankton, etc.)

Reply: We corrected as following L126-132:
*"Based on the review of Middelburg (2019), it is considered that: (i) TA decreases by 2 moles for each mole of $CaCO_3$ precipitated, by 1 mole for each mole of ammonium nitrified, by 1 mole for each mole of ammonium assimilated by phytoplankton, and TA increases by 2 moles for each mole of $CaCO_3$ dissolved, and by 1 mole for each mole of organic matter mineralized by bacteria in ammonium (See Appendix B Tab. B2)".*

p. 4, l. 132: "when bacteria mineralized" should read "when bacteria mineralize"

Reply: the sentence has been changed.

p. 4, l. 147: please delete "However" which does not make sense here.

Reply: We removed as suggested.

p. 4, l. 148: "model results" can be compared with the observations, not the model itself.

Reply: deleted sentence.

p. 5, l. 161: "winds specific conditions" should read "wind-specific conditions"

Reply: We corrected as suggested.

p. 6, l. 206: "contains a low value of WSS" – not sure what this could possibly mean. "a low value of WSS" should anyway read "a low WSS" as the last 'S' stands for 'score,' which is a value.

Reply: We corrected as suggested L220-221:
*"The model does not catch the two aforementioned maxima of chlorophyll, and it contains a low WSS and a strong bias (0.37 and +0.22 mg m$^{-3}$, respectively - Tab. 2)."*

p. 6, l. 214: "calculates a WSS value of 0.69" better had to read "yields a WSS of 0.69"

Reply: We modified this sentence as following L231-232:
*"The statistical analysis calculated a mean bias of +23 µatm, and a WSS of 0.69 (Tab. 2)."*

p. 6, l. 223: "seasonal dynamic" should read "seasonal dynamics"

Reply: We corrected as suggested.

p. 7, ll. 257–262: I am quite surprised about this. I would expect that upwelling events not only bring up cold water, but also nutrient, DIC and TA rich waters. Unfortunately, the model description does not explain how the upwelling events are represented. Could you please elaborate on this?

Reply: As aforementioned, in this study, we did not experiment the impact of an upwelling event per se but only the consequence of decrease in seawater temperature possibly either an upwelling or a latent heat loss by wind burst. We have now clarified this point (see our answer in the comments on experimental design).

p. 8, l. 281: "diatoms" should read "diatoms' " (genitive)

Reply: We corrected as suggested.

p. 8, l. 295: "in-gassing" should read "absorption" or "uptake"

Reply: We corrected as following L311-312:

*"Depending on the gradient of $CO_2$ between seawater and the atmosphere, strong wind speeds will favor either the emission or uptake of $CO_2$ (Figs. 6B & E)."*

p. 8, l. 295: "variability" at which time scales?

Reply: We modified as following L335:
*"The seasonal variability of atmospheric $CO_2$ concentrations at the urban site ...".*

p. 9, l. 329: "weaker" should read "lower"

Reply: We corrected as suggested.

p. 9, l. 354: I think that "counteracting" is more appropriate than "counterbalanced" at this point

Reply: We corrected as suggested.

p. 12, l. 424: "Environnenemnts" should read "Environnements"

Reply: We corrected as suggested.

p. 12, l. 432: "takes part" should read "is part

Reply: We corrected as suggested.

p. 12, l. 434: "Agence" should read "Agency"

Reply: We corrected as suggested.

p. 12, l. 434: "from European" should read "from the European"

Reply: We corrected as suggested.

p. 12, l. 434: "used in this paper" should read "presented in this paper"

Reply: We corrected as suggested.

---

## Referee Report (RR1)

Review of the first revised version of

**"Implementation and assessment of a carbonate system model (Eco3M-CarbOxv1.1) in a highly-dynamic Mediterranean coastal site (Bay of Marseille, France)"**

submitted for publication to *Geoscientific Model Development*
by K. Lajaunie-Salla and co-authors

**1 General comments**

The authors' reply the revised manuscript are not very "reviewer-friendly." It is nowadays standard practice to provide a "track change" (or a `LATEXDIFF`) version of the manuscript clearly identifying the changes made to the text, right in the text. The equivalent information is seemingly given in the reply, except that the line numbers provided there do not match, so that one has to search manually for the exact location of the changes.
The preparation of the manuscript would also have benefited from some extra care. Page numbers restart at 1 after page 23 without any apparent reason.

**1.1 Appreciation of the replies to reviewers**

The authors have all in all well responded to the referees' comments, with one exception though. In the response to my comment 2.3, regarding the missing effect of river intrusions on DIC — TA perturbations are taken into account, but as these are carried mainly by $HCO_3^-$, they also generate DIC perturbations of the same magnitude, which are neglected — I read at the top of the fourth page (page numbers in the response would have been helpful) that

> "Concerning the riverine inputs scenarii, we decide to focus on nitrate and alkalinity supply. In fact the model simulates the DIC increases, as is observed, which highlight that the carbonate system module is well resolved."

The reply to the comments is somewhat ambiguous as suggests that DIC changes are taken into account, while they are actually not, as stated in the manuscript at lines 413–414

> "[...] the experimental design on the Rhone River supply only considers the TA perturbation on the carbonate system but not that due to the DIC supply."

So, even if the model reproduces the observed DIC increases (as stated in the reply), this must obviously be for the wrong reasons, as only one half of the effects of the perturbation due to river intrusions is taken into account. By the way, no one argued that the carbonate system module was not well resolved.

**1.2 Appreciation of the revised manuscript**

The model description has been improved and the rationale behind the carbonate speciation calculations is now presented in a new appendix. Unfortunately, the layout of that appendix is rather chaotic which makes it difficult to read.

**2 Specific comments**

**2.1 River intrusion experiments: poor justification**

The justification added at lines 414–417 for taking the effect of river intrusions into account only in terms of the resulting TA but not DIC perturbations is rather cavalier and scientifically untenable. This is a completely unrealistic assumption that makes the outcome of the experiment meaningless and thus essentially invalidates any conclusion drawn from it.
I only see two options to address this shortcoming:

1. the river intrusion experiments are repeated with the effect on DIC included (which should be rather straightforward to correct) and the discussion of the results adapted;

2. these experiments are simply taken out of the paper as the current results are essentially unfounded.

Even in preliminary experiments, one must not chose to disregard one of two effects of a perturbation if these are of the same order of magnitude. Such arbitrary choices lead to arbitrary results.

**3 Technical comments**

**Page 1, line 22**: "the year 2017 that is a period for which" should read "the year 2017 for which"

**Page 1, line 25**: "of most of variables of carbonate system except Total Alkalinity." should read "of most of the variables of the carbonate system except for Total Alkalinity."

**Page 1, line 26**: "experiments were also conducted" should read "experiments were conducted"

**Page 1, line 26**: "to (i) seawater" should read "to (i) a seawater"

**Page 1, line 27**: "Rhône River plume intrusion" should read "Rhône River plume intrusions"; by the way: the name of that river is sometimes spelled "Rhône", more often "Rhone" — please use the same spelling consistently throughout

**Page 1, line 35**: "external forcing have" should read "external forcings have"

**Page 5, line 188**: "a salinity threshold of 37 has been chosen" – is this correct? A threshold of 37 looks rather high to me.

**Page 6, line 226**: "during MWC period" should read "during the MWC period"

**Page 7, line 271**: "15 March and 6 May" should either read "15th March and 6th May" or "March 15th and May 6th" (as on line 277, p. 8)

**Page 11, line 390**: "Moreover, previous study" should read "Moreover, a previous study", or even better reformulate the sentence to read "Moreover, Fraysse et al. (2013) highlight that ..." and discard the citation in brackets.

**Page 11, line 402**: "a longer period *ca.* 15 days" should read "a longer period of ca. 15 days"

**Page 11, lines 402–403**: "high atmospheric $p$CO$_2$ value and wind speed" better had to read "high atmospheric $p$CO$_2$ and high wind speeds"

**Page 11, line 411**: "due to some two" should read "due to two"

**29th and 30th pages (pages nr. 6 and 7)**, throughout: "in the $p$H scale" should read "on the $p$H scale"

**29th page (page nr. 6)**: "Concentration in Total Fluoride (*TF*) ions" should read "Total Fluoride concentration" (without "ions," as *TF* includes the non ionic HF)

**29th page (page nr. 6)**: "Concentration in Total Sulphate (*TS*) ions" should read "Total Sulphate concentration" ("ions" is superfluous)

**29th page (page nr. 6)**: "Concentration in Total Boron (*TB*)" should read "total boron concentration" (without "ions," as *TB* includes the non ionic B(OH)$_3$)

**29th page (page nr. 6)**: $K_F$ is the dissociation constant of hydrogen fluoride (or of hydrofluoric acid), not of fluoride ions

**29th page (page nr. 6)**: in the expression for $K_F$, the exponent for *Ions* should be 0.5 (or $\frac{1}{2}$) and not 1.5

**30th page (page nr. 7)**: "Every constant are corrected by the hydrostatic pressure" should read "All the constants are corrected for the effect of hydrostatic pressure"

Guy Munhoven
Liège, 9th October 2020

---

## Author Response (AR2)

**INSTITUT MEDITERRANEAN D'OCEANOLOGIE (M.I.O)**
**UMR 7294, UR 235**

AIX MARSEILLE UNIVERSITE
OBSERVATOIRE DES SCIENCES DE L'UNIVERS (OSU) – INSTITUT PYTHEAS

Marseille, November 17, 2020

To the Editor,
*Geoscientific Model Development*

Please find enclosed the second revised version of our research article entitled **"*Implementation and assessment of a carbonate system model (Eco3M-CarbOxv1.1) in a highly-dynamic Mediterranean coastal site (Bay of Marseille, France)*"**. We considered all the valuable comments of the Dr. Munhoven and especially, we addressed the point on the poor scientific justification of the River intrusion experiment. We hope that our manuscript may now be accepted for final publication.
All authors approved the final version of the paper for submission.

Sincerely,

Frédéric DIAZ/Katixa LAJAUNIE-SALLA

Review of the first revised version of

"Implementation and assessment of a carbonate system model (Eco3M-CarbOxv1.1) in a highly-dynamic Mediterranean coastal site (Bay of Marseille, France)"

submitted for publication to Geoscientific Model Development
by K. Lajaunie-Salla and co-authors

*General reply: We thank again the Reviewer Dr Munhoven for his second evaluation of our work. We thank too him for the detailed and useful comments that contributed to greatly improve the manuscript. We consider all of his comments to improve the manuscript.*

**1. General comments**

The authors' reply the revised manuscript are not very "reviewer-friendly." It is nowadays standard practice to provide a "track change" (or a LATEXDIFF) version of the manuscript clearly identifying the changes made to the text, right in the text. The equivalent information is seemingly given in the reply, except that the line numbers provided there do not match, so that one has to search manually for the exact location of the changes.

The preparation of the manuscript would also have benefited from some extra care. Page numbers restart at 1 after page 23 without any apparent reason.

*Reply : The preparation of the revised ms has been reviewed with care and page numbers have been reprocessed. A new check of the different sections has been carefully performed.*

**1.1 Appreciation of the replies to reviewers**

The authors have all in all well responded to the referees' comments, with one exception though. In the response to my comment 2.3, regarding the missing effect of river intrusions on DIC — TA perturbations are taken into account, but as these are carried mainly by $HCO_3^-$ , they also generate DIC perturbations of the same magnitude, which are neglected — I read at the top of the fourth page (page numbers in the response would have been helpful) that

"Concerning the riverine inputs scenarii, we decide to focus on nitrate and alkalinity supply. In fact the model simulates the DIC increases, as is observed, which highlight that the carbonate system module is well resolved."

The reply to the comments is somewhat ambiguous as suggests that DIC changes are taken into account, while they are actually not, as stated in the manuscript at lines 413–414 :

" [. . . ] the experimental design on the Rhône River supply only considers the TA perturbation on the carbonate system but not that due to the DIC supply."

So, even if the model reproduces the observed DIC increases (as stated in the reply), this must obviously be for the wrong reasons, as only one half of the effects of the perturbation due to river intrusions is taken into account. By the way, no one argued that the carbonate system module was not well resolved.

*Reply : We decided to remove the scenario of AT supply by river and all the text refered to it in the methods and results/discussion sections. See our detailed comment on this point hereafter.*

**1.2 Appreciation of the revised manuscript**

The model description has been improved and the rationale behind the carbonate speciation calculations is now presented in a new appendix. Unfortunately, the layout of that appendix is rather chaotic which makes it difficult to read.

**2. Specific comments**

**2.1 River intrusion experiments: poor justification**

The justification added at lines 414–417 for taking the effect of river intrusions into account only in terms of the resulting TA but not DIC perturbations is rather cavalier and scientifically untenable. This is a completely unrealistic assumption that makes the outcome of the experiment meaningless and thus essentially invalidates any conclusion drawn from it.

I only see two options to address this shortcoming:
1. the river intrusion experiments are repeated with the effect on DIC included (which should be rather straightforward to correct) and the discussion of the results adapted;
2. these experiments are simply taken out of the paper as the current results are essentially unfounded.
Even in preliminary experiments, one must not chose to disregard one of two effects of a perturbation if these are of the same order of magnitude. Such arbitrary choices lead to arbitrary results.

*Reply : We have chosen the second option proposed by the Reviewer. Mainly because it was not straightforward to consider DIC supply by River as considered for AT. In fact DIC is not as conservative as AT is regard to salinity. It is then impossible to use a relationship between DIC and salinity as that used between AT and salinity in the previous version of our manuscript. In the context of a 0D modelling developed here, this kind of relationship would have been the only mean to take into account the DIC supply by the Rhône River. In the revised version, we stressed (l 398-401) that rivers also supply TA and DIC and a consideration for these supplies in future works may sensibly modify the modeled carbonate balance in the BoM compared to that presented in this study. Concomitantly we slightly rewrote the end of section 4.1 on the model performance in Discussion.*

*As noted in the conclusion (l434), a coupling of the Eco3M-CarbOx model to a 3D hydrodynamics model is planed to better represent the complexity of functioning of the BOM. This implementation will enable, for example, to take into account actual DIC and AT supplies from the Rhône River by considering forcing values measured in the River.*

**3. Technical comments**

*Reply : All these technical comments have been taken into account in the new revised version of ms.*

Page 1, line 22: "the year 2017 that is a period for which" should read "the year 2017 for which"
Page 1, line 25: "of most of variables of carbonate system except Total Alkalinity." should read "of most of the variables of the carbonate system except for Total Alkalinity."
Page 1, line 26: "experiments were also conducted" should read "experiments were conducted"
Page 1, line 26: "to (i) seawater" should read "to (i) a seawater"
Page 1, line 27: "Rhône River plume intrusion" should read "Rhône River plume intrusions"; by the way: the name of that river is sometimes spelled "Rhône", more often "Rhone" — please use the same spelling consistently throughout
Page 1, line 35: "external forcing have" should read "external forcings have"
Page 5, line 188: "a salinity threshold of 37 has been chosen" – is this correct? A threshold of 37 looks rather high to me.
Page 6, line 226: "during MWC period" should read "during the MWC period"
Page 7, line 271: "15 March and 6 May" should either read "15th March and 6th May" or "March 15th and May 6th" (as on line 277, p. 8)

Page 11, line 390: "Moreover, previous study" should read "Moreover, a previous study", or even better reformulate the sentence to read "Moreover, Fraysse et al. (2013) highlight that . . . " and discard the citation in brackets.

Page 11, line 402: "a longer period ca. 15 days" should read "a longer period of ca. 15 days"

Page 11, lines 402–403: "high atmospheric pCO2 value and wind speed" better had to read "high atmospheric pCO2 and high wind speeds"

Page 11, line 411: "due to some two" should read "due to two"

29th and 30th pages (pages nr. 6 and 7), throughout: "in the pH scale" should read "on the pH scale"

29th page (page nr. 6): "Concentration in Total Fluoride (TF ) ions" should read "Total Fluoride concentration" (without "ions," as TF includes the non ionic HF)

29th page (page nr. 6): "Concentration in Total Sulphate (TS) ions" should read "Total Sulphate concentration" ("ions" is superfluous)

29th page (page nr. 6): "Concentration in Total Boron (TB)" should read "total boron concentration" (without "ions," as TB includes the non ionic B(OH)3)

29th page (page nr. 6): KF is the dissociation constant of hydrogen fluoride (or of hydrofluoric acid), not of fluoride ions

29th page (page nr. 6): in the expression for KF, the exponent for Ions should be 0.5 (or 1/2) and not 1.5.

30th page (page nr. 7): "Every constant are corrected by the hydrostatic pressure" should read "All the constants are corrected for the effect of hydrostatic pressure"

Guy Munhoven
Liège, 9th October 2020

[revised manuscript text omitted]


 **Table B3: Biogeochemical processes simulated by the Eco3M-CarbOx model**

| Notation | Biogeochemical processes | Unit | Formulation | |
|---|---|---|---|---|
| $R_{PP}^{Phy}$ | Primary production | molC m$^{-3}$ s$^{-1}$ | $R_{PP}^{PhyC} = P_{max} \cdot f_T^{PP} \cdot f_I \cdot PhyC$ | $f_Q = min[f_Q^N, f_Q^P]; f_Q^X = \frac{Q_C^X - Q_{C,min}^X}{Q_C^X - Q_{C,min}^X + \beta_X}$ $f_T^{PP} = max\left(\frac{2 \cdot (1-b) \cdot \frac{(T-T_{let})}{(T_{opt}-T_{let})}}{\left(\frac{(T-T_{let})}{(T_{opt}-T_{let})}\right)^2 - 2 \cdot b \cdot \frac{(T-T_{let})}{(T_{opt}-T_{let})} + 1} ; 0\right)$ $f_I = \left[1 - exp\left(\frac{-\alpha_{Chla} \cdot E_{PAR} \cdot Q_C^{Chla}}{P_{max} \cdot f_Q \cdot f_T^{PP}}\right)\right]$ |
| $R_{resp}^{Phy}$ | Phytoplankton respiration | molC m$^{-3}$ s$^{-1}$ | $R_{resp}^{PhyC} = k_r^{PhyC} \cdot PhyC$ | |
| $R_{uptPhy}^{NH_4}$ | NH$_4$ uptake by phytoplankton | molX m$^{-3}$ s$^{-1}$ | $R_{uptPhyN}^{NH_4} = V_{N,max} \cdot \frac{NH_4}{NH_4 + K_{NH_4}}$ | $V_{N,max} = Q_{C,max}^N \cdot R_{PP}^{Phy}$ |
| $R_{uptPhy}^{NO_3}$ | NO$_3$ uptake by phytoplankton | molN m$^{-3}$ s$^{-1}$ | $R_{uptPhyN}^{NO_3} = V_{N,max} \cdot \frac{NO_3}{NO_3 + K_{NO_3}} \cdot \left(1 - \frac{I_{in} \cdot NH_4}{NH_4 + K_{in}}\right)$ | |
| $R_{uptPhy}^{PO_4}$ | PO$_4$ uptake by phytoplankton | molP m$^{-3}$ s$^{-1}$ | $R_{uptPhyP}^{PO_4} = V_{P,max} \cdot \frac{PO_4}{PO_4 + K_{PO_4}} \cdot$ | $V_{P,max} = Q_{C,max}^P \cdot R_{PP}^{Phy}$ |
| $R_{exu}^{PhyC}$ | Phytoplankton exudation as LDOC | molC m$^{-3}$ s$^{-1}$ | $R_{exu}^{PhyC} = (1 - f_Q) \cdot R_{PP}^{Phy}$ | |
| $R_{exr}^{PhyX}$ | Phytoplankton exudation as LDON or LDOP | molX m$^{-3}$ s$^{-1}$ | $R_{exu}^{PhyX} = (1 - h_Q^X) \cdot R_{uptX}^{Phy}$ | $h_Q^X = \frac{Q_{C,max}^X - Q_C^X}{Q_{C,max}^X - Q_{C,min}^X}$ |
| $R_{BP}$ | Bacterial production | cell m$^{-3}$ s$^{-1}$ | $R_{BP} = \mu_{max}^{Ba} \cdot f_Q^{Ba} \cdot f_T^{Ba} \cdot NBA$ | $f_T^{Ba} = Q_{10}^{\frac{(T-T_{rem})}{10}}$ ; $f_Q^{BA} = min\left[1 - \frac{Q_{C,min}^{BA}}{Q_C^{BA}}, 1 - \frac{Q_{N,min}^{BA}}{Q_N^{BA}}, 1 - \frac{Q_{P,min}^{BA}}{Q_P^{BA}}\right]$ |
| $R_{BR}$ | Bacterial respiration | molC m$^{-3}$ s$^{-1}$ | $R_{BR} = \rho_g^{Ba} \cdot \frac{Q_C^{Ba} \cdot R_{BP}}{} \cdot Q_C^{Ba} \cdot R_{BP} + \rho_r^{Ba} \cdot \frac{(Q_C^{Ba} - Q_{C,min}^{Ba})}{} \cdot (Q_C^{Ba} - Q_{C,min}^{Ba}) \
[revised manuscript text omitted]

| Q$_{10}$ | Temperature coefficient | 2.0 | $\neq$ | $\neq$ |
| T$_{rem}$ | Reference temperature for mineralization | 20.0 | °C | $\neq$ |
| $k_{nit}$ | Nitrification rate | 0.05 | d$^{-1}$ | Lacroix and Grégoire (2002) |
| T$_{nit}$ | Reference temperature for nitrification | 10.0 | °C | $\neq$ |

| | | | | | |
|---|---|---|---|---|---|
| $K_{DO}$ | Half-saturation constant DO | 30.0 | $mmolO_2\,m^{-3}$ | Tett (1990) | |
| $k_{diss}$ | Dissolution rate | 10.9 | $d^{-1}$ | Gehlen et al. (2007) | |
| $k_{precip}$ | Fraction of PIC to LPOC | 0.02 | ‡ | Marty et al. (2002) | |
| $K_C$ | Half-saturation constant of $CaCO_3$ precipitation | 0.40 | $(\mu mol\,kg^{-1})^2$ | | |
| $\left(\dfrac{O}{C}\right)$ | Ratio O:C | 1.0 | ‡ | ‡ | |
| $\left(\dfrac{O}{N}\right)$ | Ratio O:N | 2.0 | ‡ | ‡ | |

**Appendix C: Short User Manual**

After uploading the whole archive on the zenodo site (ref. doi: 10.5281/zenodo.3757677), the exact version of the Eco3M-CarbOx
code used in this study can be  run as following:

make !two executable files will be created : eco3M_ini.exe and eco3M.exe

- the file config.ini allows to define: the time, time step, and save time of simulation variables biogeochemical process
- Results files are stocked in "SORTIES" directory
- Boundary conditions and forcings data are stocked in "DATA" directory
- All subroutines of biogeochemical processes are stocked in "F_PROCESS" directory

For further information, please contact Dr. Frédéric DIAZ (frederic.diaz@univ-amu.fr) or Dr. Christel PINAZO
(christel.pinazo@univ-amu.fr)

"Implementation and assessment of a carbonate system model (Eco3M-CarbOxv1.1) in a highly-dynamic Mediterranean coastal site
(Bay of Marseille, France)"
submitted for publication to Geoscientific Model Development
by K. Lajaunie-Salla and co-authors

*General reply: We thank again the Reviewer Dr Munhoven for his second evaluation of our work. We thank too him for the detailed and useful comments that contributed to greatly improve the manuscript. We consider all of his comments to improve the manuscript.*

**1. General comments**

The authors' reply the revised manuscript are not very "reviewer-friendly." It is nowadays standard practice to provide a "track change" (or a LATEXDIFF) version of the manuscript clearly identifying the changes made to the text, right in the text. The equivalent information is seemingly given in the reply, except that the line numbers provided there do not match, so that one has to search manually for the exact location of the changes.

The preparation of the manuscript would also have benefited from some extra care. Page numbers restart at 1 after page 23 without any apparent reason.

*Reply: The preparation of the revised ms has been reviewed with care and page numbers have been reprocessed. A new check of the different sections has been carefully performed.*

**1.1 Appreciation of the replies to reviewers**

The authors have all in all well responded to the referees' comments, with one exception though. In the response to my comment 2.3, regarding the missing effect of river intrusions on DIC — TA perturbations are taken into account, but as these are carried mainly by $HCO_3^-$ , they also generate DIC perturbations of the same magnitude, which are neglected — I read at the top of the fourth page (page numbers in the response would have been helpful) that

"Concerning the riverine inputs scenarii, we decide to focus on nitrate and alkalinity supply. In fact the model simulates the DIC
increases, as is observed, which highlight that the carbonate system module is well resolved."

The reply to the comments is somewhat ambiguous as suggests that DIC changes are taken into account, while they are actually not, as stated in the manuscript at lines 413–414 :

" [. . . ] the experimental design on the Rhône River supply only considers the TA perturbation on the carbonate system but not that
due to the DIC supply."

So, even if the model reproduces the observed DIC increases (as stated in the reply), this must obviously be for the wrong reasons, as only one half of the effects of the perturbation due to river intrusions is taken into account. By the way, no one argued that the carbonate system module was not well resolved.

*Reply: We decided to remove the scenario of AT supply by river and all the text referred to it in the methods and results/discussion sections. See our detailed comment on this point hereafter.*

**1.2 Appreciation of the revised manuscript**

The model description has been improved and the rationale behind the carbonate speciation calculations is now presented in a new appendix. Unfortunately, the layout of that appendix is rather chaotic which makes it difficult to read.

**2. Specific comments**

**2.1 River intrusion experiments: poor justification**

The justification added at lines 414–417 for taking the effect of river intrusions into account only in terms of the resulting TA but not

DIC perturbations is rather cavalier and scientifically untenable. This is a completely unrealistic assumption that makes the outcome of the experiment meaningless and thus essentially invalidates any conclusion drawn from it.

I only see two options to address this shortcoming:

1. the river intrusion experiments are repeated with the effect on DIC included (which should be rather straightforward to correct) and the discussion of the results adapted;

2. these experiments are simply taken out of the paper as the current results are essentially unfounded.

Even in preliminary experiments, one must not choose to disregard one of two effects of a perturbation if these are of the same order of magnitude. Such arbitrary choices lead to arbitrary results.

Reply: *We have chosen the second option proposed by the Reviewer. Mainly because it was not straightforward to consider DIC*

*supply by River as considered for AT. In fact, DIC is not as conservative as AT is regard to salinity. It is then impossible to use a relationship between DIC and salinity as that used between AT and salinity in the previous version of our manuscript. In the context of a 0D modelling developed here, this kind of relationship would have been the only mean to take into account the DIC supply by the Rhône River. In the revised version, we stressed (l 398-401) that rivers also supply TA and DIC and a consideration for these supplies in future works may sensibly modify the modeled carbonate balance in the BoM compared to that presented in this study.*

*Concomitantly we slightly rewrote the end of section 4.1 on the model performance in Discussion.*

*As noted in the conclusion (l434), a coupling of the Eco3M-CarbOx model to a 3D hydrodynamics model is planned to better represent the complexity of functioning of the BOM. This implementation will enable, for example, to take into account actual DIC and AT supplies from the Rhône River by considering forcing values measured in the River.*

**3. Technical comments**

Reply: *All these technical comments have been taken into account in the new revised version of ms.*

Page 1, line 22: "the year 2017 that is a period for which" should read "the year 2017 for which"

Page 1, line 25: "of most of variables of carbonate system except Total Alkalinity." should read "of most of the variables of the carbonate system except for Total Alkalinity."

Page 1, line 26: "experiments were also conducted" should read "experiments were conducted"

Page 1, line 26: "to (i) seawater" should read "to (i) a seawater"

Page 1, line 27: "Rhône River plume intrusion" should read "Rhône River plume intrusions"; by the way: the name of that river is sometimes spelled "Rhône", more often "Rhone" — please use the same spelling consistently throughout

Page 1, line 35: "external forcing have" should read "external forcings have"

Page 5, line 188: "a salinity threshold of 37 has been chosen" – is this correct? A threshold of 37 looks rather high to me.

Page 6, line 226: "during MWC period" should read "during the MWC period"

Page 7, line 271: "15 March and 6 May" should either read "15th March and 6th May" or "March 15th and May 6th" (as on line 277, p. 8)

Page 11, line 390: "Moreover, previous study" should read "Moreover, a previous study", or even better reformulate the sentence to read "Moreover, Fraysse et al. (2013) highlight that . . . " and discard the citation in brackets.

Page 11, line 402: "a longer period ca. 15 days" should read "a longer period of ca. 15 days"

Page 11, lines 402–403: "high atmospheric pCO2 value and wind speed" better had to read "high atmospheric pCO2 and high wind speeds"

Page 11, line 411: "due to some two" should read "due to two"

29th and 30th pages (pages nr. 6 and 7), throughout: "in the pH scale" should read "on the pH scale"

29th page (page nr. 6): "Concentration in Total Fluoride (TF ) ions" should read "Total Fluoride concentration" (without "ions," as TF includes the non ionic HF)

29th page (page nr. 6): "Concentration in Total Sulphate (TS) ions" should read "Total Sulphate concentration" ("ions" is superfluous)

29th page (page nr. 6): "Concentration in Total Boron (TB)" should read "total boron concentration" (without "ions," as TB includes the non ionic B(OH)3)

29th page (page nr. 6): KF is the dissociation constant of hydrogen fluoride (or of hydrofluoric acid), not of fluoride ions

29th page (page nr. 6): in the expression for KF, the exponent for Ions should be 0.5 (or 1/2) and not 1.5.

30th page (page nr. 7): "Every constant are corrected by the hydrostatic pressure" should read "All the constants are corrected for the effect of hydrostatic pressure"

Guy Munhoven
Liège, 9th October 2020

---

## Author Response (AR3)

Review of the first revised version of

"Implementation and assessment of a carbonate system model (Eco3M-CarbOxv1.1) in a highly-dynamic Mediterranean coastal site (Bay of Marseille, France)"

submitted for publication to Geoscientific Model Development
by K. Lajaunie-Salla and co-authors

*General reply: We thank again the Reviewer Dr Munhoven for his second evaluation of our work. We thank too him for the detailed and useful comments that contributed to greatly improve the manuscript. We consider all of his comments to improve the manuscript.*

**1. General comments**

The authors' reply the revised manuscript are not very "reviewer-friendly." It is nowadays standard practice to provide a "track change" (or a LATEXDIFF) version of the manuscript clearly identifying the changes made to the text, right in the text. The equivalent information is seemingly given in the reply, except that the line numbers provided there do not match, so that one has to search manually for the exact location of the changes.

The preparation of the manuscript would also have benefited from some extra care. Page numbers restart at 1 after page 23 without any apparent reason.

*Reply : The preparation of the revised ms has been reviewed with care and page numbers have been reprocessed. A new check of the different sections has been carefully performed.*

**1.1 Appreciation of the replies to reviewers**

The authors have all in all well responded to the referees' comments, with one exception though. In the response to my comment 2.3, regarding the missing effect of river intrusions on DIC — TA perturbations are taken into account, but as these are carried mainly by $HCO_3^-$ , they also generate DIC perturbations of the same magnitude, which are neglected — I read at the top of the fourth page (page numbers in the response would have been helpful) that

> "Concerning the riverine inputs scenarii, we decide to focus on nitrate and alkalinity supply. In fact the model simulates the DIC increases, as is observed, which highlight that the carbonate system module is well resolved."

The reply to the comments is somewhat ambiguous as suggests that DIC changes are taken into account, while they are actually not, as stated in the manuscript at lines 413–414 :

> " [. . . ] the experimental design on the Rhône River supply only considers the TA perturbation on the carbonate system but not that due to the DIC supply."

So, even if the model reproduces the observed DIC increases (as stated in the reply), this must obviously be for the wrong reasons, as only one half of the effects of the perturbation due to river intrusions is taken into account. By the way, no one argued that the carbonate system module was not well resolved.

*Reply : We decided to remove the scenario of AT supply by river and all the text refered to it in the methods and results/discussion sections. See our detailed comment on this point hereafter.*

**1.2 Appreciation of the revised manuscript**

The model description has been improved and the rationale behind the carbonate speciation calculations is now presented in a new appendix. Unfortunately, the layout of that appendix is rather chaotic which makes it difficult to read.

**2. Specific comments**
**2.1 River intrusion experiments: poor justification**
The justification added at lines 414–417 for taking the effect of river intrusions into account only in terms of the resulting TA but not DIC perturbations is rather cavalier and scientifically untenable. This is a completely unrealistic assumption that makes the outcome of the experiment meaningless and thus essentially invalidates any conclusion drawn from it.

I only see two options to address this shortcoming:
1. the river intrusion experiments are repeated with the effect on DIC included (which should be rather straightforward to correct) and the discussion of the results adapted;
2. these experiments are simply taken out of the paper as the current results are essentially unfounded.
Even in preliminary experiments, one must not chose to disregard one of two effects of a perturbation if these are of the same order of magnitude. Such arbitrary choices lead to arbitrary results.

*Reply : We have chosen the second option proposed by the Reviewer. Mainly because it was not straightforward to consider DIC supply by River as considered for AT. In fact DIC is not as conservative as AT is regard to salinity. It is then impossible to use a relationship between DIC and salinity as that used between AT and salinity in the previous version of our manuscript. In the context of a 0D modelling developed here, this kind of relationship would have been the only mean to take into account the DIC supply by the Rhône River. In the revised version, we stressed (l 398-401) that rivers also supply TA and DIC and a consideration for these supplies in future works may sensibly modify the modeled carbonate balance in the BoM compared to that presented in this study. Concomitantly we slightly rewrote the end of section 4.1 on the model performance in Discussion.*

*As noted in the conclusion (l434), a coupling of the Eco3M-CarbOx model to a 3D hydrodynamics model is planed to better represent the complexity of functioning of the BOM. This implementation will enable, for example, to take into account actual DIC and AT supplies from the Rhône River by considering forcing values measured in the River.*

**3. Technical comments**
*Reply : All these technical comments have been taken into account in the new revised version of ms.*

Page 1, line 22: "the year 2017 that is a period for which" should read "the year 2017 for which"
Page 1, line 25: "of most of variables of carbonate system except Total Alkalinity." should read "of most of the variables of the carbonate system except for Total Alkalinity."
Page 1, line 26: "experiments were also conducted" should read "experiments were conducted"
Page 1, line 26: "to (i) seawater" should read "to (i) a seawater"
Page 1, line 27: "Rhône River plume intrusion" should read "Rhône River plume intrusions"; by the way: the name of that river is sometimes spelled "Rhône", more often "Rhone" — please use the same spelling consistently throughout
Page 1, line 35: "external forcing have" should read "external forcings have"
Page 5, line 188: "a salinity threshold of 37 has been chosen" – is this correct? A threshold of 37 looks rather high to me.
Page 6, line 226: "during MWC period" should read "during the MWC period"
Page 7, line 271: "15 March and 6 May" should either read "15th March and 6th May" or "March 15th and May 6th" (as on line 277, p. 8)

Page 11, line 390: "Moreover, previous study" should read "Moreover, a previous study", or even better reformulate the sentence to read "Moreover, Fraysse et al. (2013) highlight that . . . " and discard the citation in brackets.

Page 11, line 402: "a longer period ca. 15 days" should read "a longer period of ca. 15 days"

Page 11, lines 402–403: "high atmospheric pCO2 value and wind speed" better had to read "high atmospheric pCO2 and high wind speeds"

Page 11, line 411: "due to some two" should read "due to two"

29th and 30th pages (pages nr. 6 and 7), throughout: "in the pH scale" should read "on the pH scale"

29th page (page nr. 6): "Concentration in Total Fluoride (TF ) ions" should read "Total Fluoride concentration" (without "ions," as TF includes the non ionic HF)

29th page (page nr. 6): "Concentration in Total Sulphate (TS) ions" should read "Total Sulphate concentration" ("ions" is superfluous)

29th page (page nr. 6): "Concentration in Total Boron (TB)" should read "total boron concentration" (without "ions," as TB includes the non ionic B(OH)3)

29th page (page nr. 6): KF is the dissociation constant of hydrogen fluoride (or of hydrofluoric acid), not of fluoride ions

29th page (page nr. 6): in the expression for KF, the exponent for Ions should be 0.5 (or 1/2) and not 1.5.

30th page (page nr. 7): "Every constant are corrected by the hydrostatic pressure" should read "All the constants are corrected for the effect of hydrostatic pressure"

Guy Munhoven
Liège, 9th October 2020

[revised manuscript text omitted]

To resolve the carbonate system, we calculate the $deltapH$, which is the difference of $pH$ between two iterations of the model. We initialize the run by imposing a $pH$ value of 8.

$if\ (nbiter < 1)\ pH = 8$

$pHtol = 0.001$ ! tolerance for iterations end

$deltapH = pHtol + 1$

$do\ while\ (abs(deltapH) > 0.0001)$

$$H = 10^{-pH}$$
$$Denom = H^2 + K_1 \cdot H + K_1 \cdot K_2$$
$$CAlk = DIC \cdot K_1 \cdot \left(\frac{H + 2 \cdot K_2}{Denom}\right)$$
$$BAlk = \frac{TB \cdot K_B}{K_B + H}$$
$$OH = \frac{K_e}{H}$$

$$FreetoTot = 1 + \frac{TS}{K_S}$$
$$Hfree = \frac{H}{FreetoTot}$$
$$HSO_4 = \frac{TS}{1 + \frac{K_S}{Hfree}}$$
$$HF = \frac{TF}{1 + \frac{K_F}{Hfree}}$$
$$Residual = TA - CAlk - BAlk - OH + Hfree + HSO_4 + HF$$

$$Slope = DIC \cdot H \cdot K_1 \cdot (H^2 + K_1 \cdot K_2 + 4 \cdot H \cdot K_2)$$
$$Slope = \frac{Slope}{Denom^2} + OH + H + \frac{BAlk \cdot H}{K_B + H}$$
$$Slope = \log 10 \cdot Slope$$

$deltapH = Residual/Slope$ ! this is Newton's method

$do\ while\ (abs(deltapH) > 1)\ deltapH = \frac{deltapH}{2}$ ! to keep the jump from being too big

$enddo$

$pH = pH + deltapH$ ! Is on the same scale as K$_1$ and K$_2$ were calculated, $i.e.$ total $p$H scale

$$pCO_2 = \left(\frac{DIC \cdot H^2}{H^2 + K_1 \cdot H + K_1 \cdot K_2}\right) \cdot \frac{10^6}{K_0 \cdot FugFac}$$ ! in µatm

$$CO_2 = \frac{DIC \cdot 10^6}{1 + \frac{K_1}{H} + \frac{K_1 \cdot K_2}{H^2}}$$

$$HCO_3 = \frac{K_1 \cdot CO_2}{H}$$

$$CO_3 = \frac{K_2 \cdot HCO_3}{H}$$

$$Omega = \frac{Ca \cdot CO_3 \cdot 10^{-6}}{K_{ca}}$$

**Appendix B: Biogeochemical model variables and parameters**

[revised manuscript text omitted]


[revised manuscript text omitted]

submitted for publication to Geoscientific Model Development by K. Lajaunie-Salla and co-authors

*General reply: We thank again the Reviewer Dr Munhoven for his second evaluation of our work. We thank too him for the detailed and useful comments that contributed to greatly improve the manuscript. We consider all of his comments to improve the manuscript.*

**1. General comments**

The authors' reply the revised manuscript are not very "reviewer-friendly." It is nowadays standard practice to provide a "track change" (or a LATEXDIFF) version of the manuscript clearly identifying the changes made to the text, right in the text. The equivalent information is seemingly given in the reply, except that the line numbers provided there do not match, so that one has to search manually for the exact location of the changes.

The preparation of the manuscript would also have benefited from some extra care. Page numbers restart at 1 after page 23 without any apparent reason.

*Reply: The preparation of the revised ms has been reviewed with care and page numbers have been reprocessed. A new check of the different sections has been carefully performed.*

**1.1 Appreciation of the replies to reviewers**

The authors have all in all well responded to the referees' comments, with one exception though. In the response to my comment 2.3, regarding the missing effect of river intrusions on DIC — TA perturbations are taken into account, but as these are carried mainly by $HCO_3^-$, they also generate DIC perturbations of the same magnitude, which are neglected — I read at the top of the fourth page (page numbers in the response would have been helpful) that

"Concerning the riverine inputs scenarii, we decide to focus on nitrate and alkalinity supply. In fact the model simulates the DIC
increases, as is observed, which highlight that the carbonate system module is well resolved."

The reply to the comments is somewhat ambiguous as suggests that DIC changes are taken into account, while they are actually not, as stated in the manuscript at lines 413–414 :

" [. . . ] the experimental design on the Rhône River supply only considers the TA perturbation on the carbonate system but not that
due to the DIC supply."

So, even if the model reproduces the observed DIC increases (as stated in the reply), this must obviously be for the wrong reasons, as only one half of the effects of the perturbation due to river intrusions is taken into account. By the way, no one argued that the carbonate system module was not well resolved.

*Reply: We decided to remove the scenario of AT supply by river and all the text referred to it in the methods and results/discussion sections. See our detailed comment on this point hereafter.*

**1.2 Appreciation of the revised manuscript**

The model description has been improved and the rationale behind the carbonate speciation calculations is now presented in a new appendix. Unfortunately, the layout of that appendix is rather chaotic which makes it difficult to read.

**2. Specific comments**

**2.1 River intrusion experiments: poor justification**

The justification added at lines 414–417 for taking the effect of river intrusions into account only in terms of the resulting TA but not

DIC perturbations is rather cavalier and scientifically untenable. This is a completely unrealistic assumption that makes the outcome of the experiment meaningless and thus essentially invalidates any conclusion drawn from it.

I only see two options to address this shortcoming:

1. the river intrusion experiments are repeated with the effect on DIC included (which should be rather straightforward to correct) and the discussion of the results adapted;

2. these experiments are simply taken out of the paper as the current results are essentially unfounded.

Even in preliminary experiments, one must not choose to disregard one of two effects of a perturbation if these are of the same order of magnitude. Such arbitrary choices lead to arbitrary results.

*Reply: We have chosen the second option proposed by the Reviewer. Mainly because it was not straightforward to consider DIC*

*supply by River as considered for AT. In fact, DIC is not as conservative as AT is regard to salinity. It is then impossible to use a relationship between DIC and salinity as that used between AT and salinity in the previous version of our manuscript. In the context of a 0D modelling developed here, this kind of relationship would have been the only mean to take into account the DIC supply by the Rhône River. In the revised version, we stressed (l 398-401) that rivers also supply TA and DIC and a consideration for these supplies in future works may sensibly modify the modeled carbonate balance in the BoM compared to that presented in this study.*

*Concomitantly we slightly rewrote the end of section 4.1 on the model performance in Discussion.*

*As noted in the conclusion (l434), a coupling of the Eco3M-CarbOx model to a 3D hydrodynamics model is planned to better represent the complexity of functioning of the BOM. This implementation will enable, for example, to take into account actual DIC and AT supplies from the Rhône River by considering forcing values measured in the River.*

**3. Technical comments**

*Reply: All these technical comments have been taken into account in the new revised version of ms.*

Page 1, line 22: "the year 2017 that is a period for which" should read "the year 2017 for which"

Page 1, line 25: "of most of variables of carbonate system except Total Alkalinity." should read "of most of the variables of the carbonate system except for Total Alkalinity."

Page 1, line 26: "experiments were also conducted" should read "experiments were conducted"

Page 1, line 26: "to (i) seawater" should read "to (i) a seawater"

Page 1, line 27: "Rhône River plume intrusion" should read "Rhône River plume intrusions"; by the way: the name of that river is sometimes spelled "Rhône", more often "Rhone" — please use the same spelling consistently throughout

Page 1, line 35: "external forcing have" should read "external forcings have"

Page 5, line 188: "a salinity threshold of 37 has been chosen" – is this correct? A threshold of 37 looks rather high to me.

Page 6, line 226: "during MWC period" should read "during the MWC period"

Page 7, line 271: "15 March and 6 May" should either read "15th March and 6th May" or "March 15th and May 6th" (as on line 277, p. 8)

Page 11, line 390: "Moreover, previous study" should read "Moreover, a previous study", or even better reformulate the sentence to read "Moreover, Fraysse et al. (2013) highlight that . . . " and discard the citation in brackets.

Page 11, line 402: "a longer period ca. 15 days" should read "a longer period of ca. 15 days"

Page 11, lines 402–403: "high atmospheric pCO2 value and wind speed" better had to read "high atmospheric pCO2 and high wind speeds"

Page 11, line 411: "due to some two" should read "due to two"

29th and 30th pages (pages nr. 6 and 7), throughout: "in the pH scale" should read "on the pH scale"

29th page (page nr. 6): "Concentration in Total Fluoride (TF ) ions" should read "Total Fluoride concentration" (without "ions," as TF includes the non ionic HF)

29th page (page nr. 6): "Concentration in Total Sulphate (TS) ions" should read "Total Sulphate concentration" ("ions" is superfluous)

29th page (page nr. 6): "Concentration in Total Boron (TB)" should read "total boron concentration" (without "ions," as TB includes the non ionic B(OH)3)

29th page (page nr. 6): KF is the dissociation constant of hydrogen fluoride (or of hydrofluoric acid), not of fluoride ions

29th page (page nr. 6): in the expression for KF, the exponent for Ions should be 0.5 (or 1/2) and not 1.5.

30th page (page nr. 7): "Every constant are corrected by the hydrostatic pressure" should read "All the constants are corrected for the effect of hydrostatic pressure"

Guy Munhoven

Liège, 9th October 2020